



# Observations of high time-resolution and size-resolved aerosol chemical composition and microphysics in the central Arctic: implications for climate-relevant particle properties

Benjamin Heutte[1], Nora Bergner[1], Hélène Angot[1,a], Jakob B. Pernov[1], Lubna Dada[1,2], Jessica A. Mirrielees[3], Ivo Beck[1], Andrea Baccarini[1,b], Matthew Boyer[4], Jessie M. Creamean[5], Kaspar R. Daellenbach[2], Imad El Haddad[2], Markus M. Frey[6], Silvia Henning[7], Tiia Laurila[4], Vaios Moschos[2], Tuukka Petäjä[4], Kerri A. Pratt[3,8], Lauriane L. J. Quéléver[4], Matthew D. Shupe[9,10], Paul Zieger[11,12], Tuija Jokinen[4,13], Julia Schmale[1,*]

[1]Extreme Environments Research Laboratory, Ecole Polytechnique Fédérale de Lausanne (EPFL) Valais Wallis, Sion, Switzerland
[2]Laboratory of Atmospheric Chemistry, Paul Scherrer Institute, Villigen, Switzerland
[3]Department of Chemistry, University of Michigan, Ann Arbor, MI, USA
[4]Institute for Atmospheric and Earth System Research, INAR/Physics, Faculty of Science, University of Helsinki, Helsinki, Finland
[5]Department of Atmospheric Science, Colorado State University, Fort Collins, CO, USA
[6]Natural Environment Research Council, British Antarctic Survey, Cambridge, UK
[7]Leibniz Institute for Tropospheric Research, Leipzig, Germany
[8]Department of Earth & Environmental Sciences, University of Michigan, Ann Arbor, MI USA
[9]Cooperative Institute for Research in Environmental Sciences, University of Colorado, Boulder, CO, USA
[10]Physical Sciences Laboratory, National Oceanic and Atmospheric Administration, Boulder, CO, USA
[11]Department of Environmental Science, Stockholm University, Stockholm, Sweden
[12]Bolin Centre for Climate Research, Stockholm, Sweden
[13]Climate and Atmosphere Research Centre (CARE-C), The Cyprus Institute, Nicosia, Cyprus

[a] Now at: Univ. Grenoble Alpes, CNRS, INRAE, IRD, Grenoble INP, IGE, Grenoble, France
[b] Now at: Laboratory of Atmospheric Processes and their Impacts, Ecole Polytechnique Fédérale de Lausanne (EPFL), Lausanne, Switzerland

*Correspondence to*: Julia Schmale (julia.schmale@epfl.ch)

**Abstract.**

Aerosols play a critical role in the Arctic's radiative balance, influencing solar radiation and cloud formation based on their physicochemical properties (e.g., size, abundance, and chemical composition). Limited observations in the central Arctic leave gaps in understanding aerosol dynamics year-round, affecting model predictions of climate-relevant properties. Here, we present the first annual high-time resolution observations of submicron aerosol chemical composition in the central Arctic during the Arctic Ocean 2018 (AO2018) and the 2019-2020 Multidisciplinary drifting Observatory for the Study of Arctic Climate (MOSAiC) expeditions. Seasonal variations in aerosol mass concentrations and chemical composition were found to be driven by typical Arctic seasonal regimes. Organic aerosols dominated the pristine summer, while anthropogenic sulfate prevailed in autumn and spring under Arctic haze conditions. Ammonium, which impacts aerosol acidity, was consistently



less abundant, relative to sulfate, in the central Arctic compared to lower latitudes of the Arctic. Cyclonic (storm) activity was found to have a significant influence on aerosol variability by enhancing both emission from local sources and transport of remote aerosol, with locally wind-generated particles contributing up to 80% (20%) of the cloud condensation nuclei population in autumn (spring). While the analysis presented herein provides the current central Arctic aerosol baseline, which will serve to improve climate model predictions in the region, it also underscores the importance of integrating short-timescale processes, such as seasonal wind-driven aerosol sources from blowing snow and open leads/ocean in model simulations, especially in light of the declining mid-latitude anthropogenic emissions influence and the increasing local anthropogenic emissions.

## 1 Introduction

Under the influence of climate change, surface temperatures in the Arctic have increased at a rate nearly fourfold compared to that of the global average, with the highest warming rates in the dark autumn and winter months (Rantanen et al., 2022). This phenomenon, referred to as Arctic amplification, is associated with a rapidly changing Arctic environment (Serreze and Barry, 2011), including a substantial loss of sea ice in the central Arctic (Jahn et al., 2024; Stroeve and Notz, 2018). The resulting decrease in the surface albedo is only one of the many feedback mechanisms contributing to the amplified warming (Pithan and Mauritsen, 2014; Serreze and Barry, 2011; Wendisch et al., 2023). Aerosols, acting as short-lived climate forcers, have long been recognized to be important components of the Arctic radiative balance (Barrie, 1986; Schmale et al., 2021; Shaw and Stamnes, 1980; Shindell and Faluvegi, 2009). First, through aerosol-radiation interactions (ARIs), aerosols can directly scatter (cooling effect) or absorb (warming effect, at the altitude of the absorbing layer) the incoming shortwave solar radiation. Second, through aerosol-cloud interactions (ACIs), a subset of aerosols can act as cloud condensation nuclei (CCN) or ice nucleating particles (INPs) which, depending on their physicochemical properties and abundance, can modulate cloud formation, lifetime (Albrecht, 1989), and radiative properties (Twomey, 1977).

Regarding ARIs, although the overall effect in the Arctic remains a net cooling (Li et al., 2022; Quinn et al., 2008; von Salzen et al., 2022; Sand et al., 2015), the past decades' reduction in anthropogenic emissions of sulfur dioxide (a precursor to sulfate), related to emission regulation policies and the fall of the Soviet Union in the 1990s, have contributed to the observed Arctic warming, because of a diminished dimming effect (Acosta Navarro et al., 2016; Breider et al., 2017; Gong et al., 2010; von Salzen et al., 2022; Shindell and Faluvegi, 2009). Regarding ACIs, in the central Arctic, clouds have a net warming effect throughout most of the year, due to the re-emission of terrestrial longwave radiation by low-level clouds, especially during the dark autumn and winter months (Curry and Ebert, 1992; Shupe and Intrieri, 2004). Clouds exert a net negative forcing for a brief period in summer over Arctic sea ice (Intrieri et al., 2002; Shupe and Intrieri, 2004). Present-day models struggle to represent the sign and magnitude of the seasonally varying cloud radiative effects in the Arctic (Tjernström et al., 2008; Taylor



et al., 2019; Wei et al., 2021; Yeo et al., 2022), amongst others owing to poorly simulated CCN and INPs that define the cloud
phase.

Overall, ARI and ACI are heavily dependent on the particles' size, chemical composition, and abundance, which all follow a
significant seasonality in the Arctic (e.g., Croft et al., 2016b; Freud et al., 2017; Karlsson et al., 2021; Nguyen et al., 2016;
Platt et al., 2022; Quinn et al., 2002; Schmale et al., 2022; Sharma et al., 2013; Tunved et al., 2013; Zieger et al., 2023). This
pattern is driven by seasonally varying local and regional environmental and meteorological conditions, including air
temperature, shortwave radiation, sea ice extent, atmospheric stratification (and boundary layer dynamics), and the strength of
the Arctic polar vortex (Willis et al., 2018, and references therein).

In winter (December-February) and spring (March-May), aerosol mass concentrations in most locations are impacted by long-
range transported anthropogenic (and natural) aerosols from lower latitudes, associated with an expansion of the Arctic front
further south to as low as 40°N (Barrie and Hoff, 1984; Quinn et al., 2007). This implies that pollution emitted within the polar
dome, particularly from Eurasia (Willis et al., 2018), is exposed to thermodynamically facilitated poleward isentropic transport
into the high Arctic boundary layer (Stohl, 2006). In addition, the prevalent dry and stratified atmospheric conditions at this
time of the year, which minimize aerosol removal processes, lead to the observed accumulation of atmospheric pollutants
during the Arctic haze (Croft et al., 2016b; Mitchell, 1957; Quinn et al., 2007; Rahn et al., 1977; Rahn and McCaffrey, 1979;
Shaw, 1995). Haze is primarily composed of accumulation mode particles, comprising a mixture of aged sulfate, organics,
black carbon, ammonium, and nitrate (Lange et al., 2018; Moschos et al., 2022b; Quinn et al., 2007), with the potential to
strongly affect atmospheric radiative properties (e.g., Quinn et al., 2002, 2008; Schmale et al., 2022; Schmeisser et al., 2018;
Shaw and Stamnes, 1980). Sulfate has been found to be the major component of Arctic haze (Quinn et al., 2007, and references
therein), generally making for very acidic aerosols that are only partly neutralized by low concentrations of ammonium at the
surface (Fisher et al., 2011), but ultimately depending on the particles' mixing state (Kirpes et al., 2018). Atmospheric aging
during air mass transport, through condensation of low-volatility gases on existing particles, coagulation processes, cloud
processing, and/or photooxidation reactions, is a key mechanism that controls particle activation potential, especially for black
carbon and organic species (Ervens et al., 2010; Jimenez et al., 2009; Liu et al., 2011). Several studies from across the Arctic
have reported annual cycles of haze tracers (mainly sulfate and black carbon), spanning from January to April, with a maximum
typically in March and April (e.g., Croft et al., 2016b; Platt et al., 2022; Quinn et al., 2007; Schmale et al., 2022; Sharma et
al., 2006). Local emissions of sea salt from wind-driven mechanisms, including sea spray and blowing snow, are also an
important source of aerosol loading in the Arctic in winter and spring, particularly when some of the highest yearly wind
speeds occur (Chen et al., 2022; Gong et al., 2023; Huang and Jaeglé, 2017; Kirpes et al., 2019; Lapere et al., 2024; Marelle
et al., 2021; May et al., 2016; Radke et al., 1976).



During the transition from spring to summer, the Arctic front retracts northward, thus limiting the long-range transport of emissions from lower latitudes (Bozem et al., 2019). This, associated with more frequent precipitation and a weaker atmospheric stratification, both locally and along the trajectory of transported air masses, leads to the summertime Arctic

(June-August) being characterized by relatively low aerosol mass concentrations from more local/regional emissions (Stohl, 2006). The aerosol population in summer is characterized by a dominance of Aitken mode and nucleation mode particles (Boyer et al., 2023; Freud et al., 2017; Pernov et al., 2022; Tunved et al., 2013) originating from local biogenic sources, primary marine and terrestrial aerosols, or secondary particles formed via new particle formation (Baccarini et al., 2020; Beck et al., 2020; Brean et al., 2023; Schmale and Baccarini, 2021) or condensation of precursor gases onto pre-existing particles. Organic

aerosols from different sources contribute significantly to the submicron aerosol mass concentrations in summer (e.g., Chang et al., 2011; Croft et al., 2019; Fu et al., 2009, 2013; Leaitch et al., 2018; Moschos et al., 2022b; Nielsen et al., 2019; Siegel et al., 2021). One important organic compound in summer is methanesulfonic acid (MSA), an oxidation product of marine-sourced dimethylsulfide (DMS), while part of the sulfate mass present in the summer Arctic boundary layer also originates from DMS oxidation (Barnes et al., 2006; Leaitch et al., 2013; Leck and Persson, 1996).


Finally, the autumn season (September-November) marks a minimum in total particle number and mass concentration with a dominant accumulation mode, owing to limited transport from lower latitudes, less frequent new-particle formation events, and efficient wet removal of particles (Croft et al., 2016b). However, little is known about the aerosol chemical composition and sources during this season, especially in the central Arctic.


Present-day knowledge on the seasonally varying chemical composition in the Arctic, and the processes related to it, has predominantly been obtained from observations at land-based stations (AMAP, 2006; Moschos et al., 2022a; Platt et al., 2022; Schmale et al., 2022; Sharma et al., 2019; Ström et al., 2003). Whether the observations from these lower latitude stations can be extrapolated throughout the Arctic, particularly to the central Arctic, remains an open question (Freud et al., 2017; Schmale

et al., 2021). Direct observations of aerosol physicochemical properties in the central Arctic have historically been limited to short ship-based and aircraft summertime campaigns. Among those, a series of expeditions onboard the Swedish icebreaker (I/B) *Oden* significantly contributed to our understanding of aerosol processes in the summertime central Arctic Ocean. Such expeditions (and example literature references) include the International Arctic Ocean Expeditions of 1991 (Leck et al., 1996; Leck and Persson, 1996), 1996 (Hillamo et al., 2001; Leck et al., 2001; Zhou et al., 2001), 2001 (Leck et al., 2004; Tjernström,

2005), and 2018 (Karlsson et al., 2022; Lawler et al., 2021; Siegel et al., 2021), as well as the Arctic Summer Cloud Ocean Study (ASCOS) expedition in 2008 (Chang et al., 2011; Hamacher-Barth et al., 2016; Mauritsen et al., 2011; Tjernström et al., 2014). Despite the year-round observations at the land-based stations, there are still severe knowledge gaps on aerosol sources, sinks, chemical composition, and associated processes in the Arctic (Schmale et al., 2021; Willis et al., 2018), which stem from a general lack of organic aerosol and speciation measurements and a central Arctic summer observation bias. The

available information appears to be insufficient for models to satisfactorily (i.e., without a large model spread) reproduce the



seasonality and abundance of anthropogenic and natural aerosol species throughout the Arctic (AMAP, 2011, 2015; Eckhardt et al., 2015; Lapere et al., 2023; Shindell et al., 2008).

In particular, measurements of bulk chemical composition in the Arctic have often been limited to off-line techniques, through
analysis of aerosols collected on filter samples (e.g., Hillamo et al., 2001; Moschos et al., 2022a; Schmale et al., 2022).
Although such techniques offer a good quantitative and qualitative assessment of aerosol bulk chemical composition, the time resolution over which they are performed (days to weeks) is evidently insufficient to resolve processes happening on shorter time scales (e.g., in-cloud aerosol processing, wind-driven aerosolization processes, and intense pollution transport events). Detailed chemical composition and mixing state have been obtained from single-particle microscopy measurements (e.g.,
Adachi et al., 2022; Bigg and Leck, 2008; Hamacher-Barth et al., 2016; Kirpes et al., 2022). The development of on-line aerosol mass spectrometry techniques over the last decades has provided the ability to study aerosol chemical composition at much higher temporal and spectral resolutions, shedding light on the sources and processes controlling the Arctic aerosol populations (e.g., Chang et al., 2011; Gunsch et al., 2020; Karlsson et al., 2022; Köllner et al., 2021; Nielsen et al., 2019; Ovadnevaite et al., 2011a; Willis et al., 2016). Yet, studies reporting on-line measurements from aerosol mass spectrometers
in the Arctic remain scarce due to the technical complexities associated with the operation of such instruments in remote environments.

Furthermore, in the rapidly changing Arctic, it is expected that local and remote emission sources and processes of anthropogenic and natural aerosols will change (Schmale et al., 2021), associated with socio-economical changes within the
Arctic and sub-Arctic regions, changing atmospheric transport patterns (Heslin-Rees et al., 2020; Pernov et al., 2022), sea ice retreat, increased liquid precipitations (Bintanja and Andry, 2017), and ecosystem shifts (Lannuzel et al., 2020). The summer melt season will likely further lengthen at the expense of a shortened winter sea ice growth (Markus et al., 2009; Stroeve et al., 2014), having direct consequences on the coupled ocean-sea ice-atmosphere processes (Willis et al., 2023), and the role that transition seasons (i.e., spring and autumn) will play for this changing seasonality is yet to be elucidated with present-day
measurements. Importantly, the frequency and intensity of extreme synoptic-scale circulation events, including cyclones and warm and moist air mass intrusions into the Arctic have increased over the last decades (Graham et al., 2017; Overland, 2021; Zhang et al., 2023). While such events have been shown to be associated with high levels of pollutants transported to the Arctic in spring, profoundly affecting aerosol chemical composition and CCN populations (Dada et al., 2022a; Stohl et al., 2007), less is known on the aerosol-driven impact of such extreme events in the pristine autumn transition season, when the ocean
freeze-up happens.

In this study, we investigate the annual cycle of aerosol chemical composition in the central Arctic, based on unique year-long aerosol physicochemical measurements, collected with a high-resolution aerosol mass spectrometer during two ship-based expeditions to the central Arctic between 2018 and 2020, namely the Arctic Ocean 2018 (AO2018) expedition (Baccarini et





al., 2020; Karlsson et al., 2022) and the 2019-2020 Multidisciplinary drifting Observatory for the Study of Arctic Climate (MOSAiC) expedition (Shupe et al., 2022). We assess the relevance of land-based pan-Arctic station observations for the central Arctic conditions through a comparison of our measurements with those from various stations. Finally, we infer the processes governing the central Arctic aerosol populations using high-time resolution and size-resolved chemical measurements. We investigate through case studies and a clustering of particle number size distributions the contribution of

local and remote aerosol sources to the overall aerosol and CCN number concentration in the dark and pristine autumn season as well as during the spring haze season, with specific emphasis on storm-induced high concentration events.

## 2 Experimental and methods

### 2.1. The MOCCHA and MOSAiC expeditions

Data used in this study were collected during the Microbiology Ocean Cloud Coupling in the High Arctic (MOCCHA)

campaign as part of the Arctic Ocean 2018 (AO2018) expedition, as well as during the 2019-2020 Multidisciplinary drifting Observatory for the Study of Arctic Climate (MOSAiC) expedition, both taking place in the central Arctic Ocean (Fig. 1). These two expeditions set out to gather observational data, using state-of-the-art instrumentation, to close knowledge gaps on the coupled atmospheric-ice-ocean-ecosystem processes driving, or influenced by, the changing Arctic climate. This work primarily focuses on surface observations made during MOSAiC. Measurements from MOCCHA close the summer data gap

when no chemical composition measurements were available during MOSAiC (see Sect. 2.2).

During MOCCHA, the Swedish I/B *Oden* was moored to an ice floe and drifted with the central Arctic sea ice, at latitudes higher than 88° N, between August 14th and September 14th, 2018. Detailed descriptions of the campaign, the aerosol instrumentation, and sampling conditions can be found elsewhere (Baccarini et al., 2020; Karlsson et al., 2022; Lawler et al.,

2021; Siegel et al., 2021).

During MOSAiC, the German research vessel (R/V) *Polarstern* (Knust, 2017) drifted in the central Arctic, whilst moored to an ice flow, from October 4th, 2019 to September 20th, 2020, at latitudes mostly above 80° N. To provide context into the sea ice extent during that year, we show in Fig.1 the minimum and maximum sea ice extent, respectively reached on September

15th and March 5th, 2020. Except for the drift period during leg 4 in mid-summer (i.e., between June 19th and July 31st), *Polarstern* was in general far away from the marginal ice zone and the open ocean. This means that the results presented in this study are mostly representative of the ice-covered central Arctic Ocean region. Most of the measurements relevant to this work were carried out in the *Swiss* container onboard *Polarstern*, combining aerosol physicochemical properties (Heutte et al., 2023b) and trace gas measurements (Angot et al., 2022b). For further information on the expedition conditions and technical

descriptions of the observations performed by the "atmosphere", "oceanography", and "snow and sea ice" teams, refer to Shupe et al. (2022), Rabe et al. (2022), and Nicolaus et al. (2022), respectively.





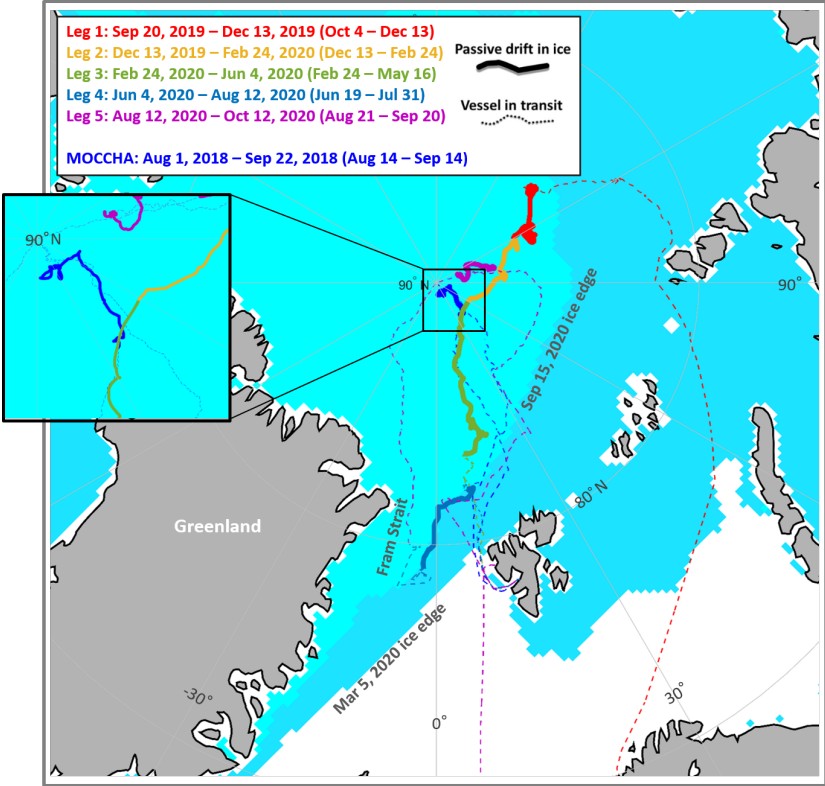

**Figure 1: Expedition tracks during MOSAiC and MOCCHA.** The MOSAiC track is adapted from Shupe et al. (2022) and colored by the leg of the expedition. Periods of passive drift by *Polarstern* and *Oden* (solid) and periods of transit when the vessel was underway (dotted) are distinguished. The inclusive dates for each of the 5 legs during MOSAiC and the whole MOCCHA campaign are given in the legend, with the second set of dates in parentheses being the dates spent in passive drift. A zoom-in above 88°N is provided for the MOCCHA campaign. The approximate sea ice edge at the annual maximum (Mar 5, 2020) and minimum (Sep 15, 2020), from NIMBUS-7 and DMSP SSM/I-SSMIS passive microwave data (National Snow and Ice Data Center; Cavalieri et al., 1996), is also provided for the MOSAiC year.

Importantly, for both MOCCHA and MOSAiC, aerosols were sampled through heated inlet lines, ensuring that sampling occurred at relative humidities below 40% (see Heutte et al. (2023c) for measurements of temperature and relative humidity inside the inlet, and Heutte et al. (2023b) for a description of the inlet system and flow rates), resulting in measurements of

dried particles, following the Global Atmosphere Watch standards for aerosol sampling (WMO, 2016).

## 2.2 Aerosol chemical composition measurements

The bulk chemical composition of non-refractory submicron aerosols (NR-PM$_1$) was measured using an Aerodyne high-resolution time-of-flight aerosol mass spectrometer (HR-ToF-AMS, hereafter referred to as AMS) during MOCCHA and MOSAiC. Detailed technical descriptions of the AMS functioning can be found in DeCarlo et al. (2006) and Canagaratna et

al. (2007). The readers are referred to Heutte et al. (2023b) for the description of the AMS operation, including calibrations, during MOSAiC and to Karlsson et al. (2022) during MOCCHA. Importantly, the same instrument was used on both expeditions. We refer to "non-refractory" species as the species that are flash-vaporized in the AMS at a temperature below



~600 °C, that of the resistively heated tungsten vaporizer. In practice, such species include sulfate ($SO_4^{2-}$), nitrate ($NO_3^-$), ammonium ($NH_4^+$), chloride (Chl), and organics (Org). Refractory species, including black carbon, sea salt, crustal materials,

and metal oxides, are hence not quantitatively detected by the AMS. However, a small fraction of refractory species can undergo slow vaporization and surface ionization at 600 °C (Drewnick et al., 2015; Ovadnevaite et al., 2012). For example, sea salt, which has a boiling point temperature of 1465 °C, can still be partly vaporized at 600 °C and contribute to the Chl signal (Zorn et al., 2008). Furthermore, the AMS can efficiently measure non-refractory species that are internally mixed with refractory ones (e.g., an organic coating on sea salt) (Salcedo et al., 2006). During MOSAiC (MOCCHA), the AMS was

operated with an effective time resolution of 90 sec (60 sec). In this work, the data were averaged (arithmetic mean: A.M.) to either 10 min or 1 h depending on the analysis purposes. The detection limits for the five main chemical species, $SO_4^{2-}$, $NO_3^-$, $NH_4^+$, Chl, and Org, are reported in Table S1 at 10 min and 1 h time resolutions. Values below detection limit were not removed when calculating monthly statistics (from the hourly-averaged data) reported in Sect. 3.1. In the cases where the fraction of data below the detection limit was high (i.e., > 50%; see Table S1), we clearly stated it and the data were not further analyzed.

Instrumental failures caused the AMS to cease functioning during MOSAiC between December 5[th], 2019, and February 29[th], 2020, between May 30[th] and June 6[th], 2020, and after July 10[th], 2020. Issues with the AMS turbo pumps rendered the $NH_4^+$ measurements very noisy in June and July, 2020; thus the $NH_4^+$ data were discarded for that period (Heutte et al., 2023b).

Scaling factors, derived from a mass closure analysis between the AMS and a scanning mobility particle sizer (SMPS, see

Sect. 2.3), were applied to the species' mass concentrations during MOSAiC (Heutte et al., 2023b). These scaling factors were derived and applied independently for the measurements periods in-between the non-operational periods mentioned above, and the scaled concentrations are expected to be upper estimates (Heutte et al., 2023b). For the MOCCHA data, such a mass closure was performed by Karlsson et al. (2022), between the AMS and a custom-made differential mobility particle sizer (DMPS), and no scaling factors were applied to the AMS in this case given the closure agreement.


As previously mentioned, sea salt cannot be quantified using the AMS. However, following the approach suggested by Ovadnevaite et al. (2012), we estimated the particulate sodium chloride mass concentrations with the AMS using the signal of the $^{23}Na^{35}Cl^+$ fragment at the mass-to-charge ratio (*m/z*) of 58 (see Fig. S1). Based on a calibration of the AMS with sea salt, Ovadnevaite et al. (2012) found that the $^{23}Na^{35}Cl^+$ sea salt surrogate should be multiplied by a calibration factor of 51.

Therefore, we multiplied our $^{23}Na^{35}Cl^+$ signal by 51 to estimate sea salt mass concentrations, averaged (A.M.) to 10 min time resolution. In the absence of any AMS calibration for sea salt during MOSAiC, and because the calibration factor reported by Ovadnevaite et al. (2012) is likely only valid for the AMS they used (with its particular tuning), sea salt mass concentrations reported in this study can only be considered as estimates and should be looked at qualitatively rather than quantitatively. Hence, concentrations are reported in arbitrary units (a.u.) rather than µg/m³.




The particle time-of-flight (PToF) feature of the AMS also enables retrieval of the size-resolved chemical composition (Jimenez et al., 2003; Salcedo et al., 2006; Zhang et al., 2005). The particle vacuum aerodynamic diameter is inferred from the time it takes for the particle to travel from the mechanical chopper (determining time zero of flight) to the detector. Velocity calibrations, used to convert the particle time of flight to diameter, were regularly performed during MOSAiC, using size-
selected monodisperse ammonium nitrate ($NH_4NO_3$) and ammonium sulfate (($NH_4)_2SO_4$) particles. Importantly, it should be mentioned that the calibration factor used to convert the measured particle time of flight to diameter relied on a comparison with SMPS data and a conversion from vacuum aerodynamic diameter ($D_{va}$) to mobility diameter ($D_m$). This relation is linearly dependent on the particle density ($\rho$), if we assume that particles are spherical ($D_{va} = D_m * \rho$), such that an uncertainty in the density estimated from the particle chemical composition (see calculation details in Heutte et al. (2023b)) would propagate
into an uncertainty of the same magnitude for the mass size distribution. Given the relatively low signal-to-noise ratio of the measured species in the pristine atmosphere of the central Arctic, size distributions in this work are only reported as monthly medians for sulfate and organics, for a set of months during MOSAiC. A monomodal log-normal distribution was fitted to each monthly sulfate mass size distribution to retrieve the mode diameter, using the "Multipeak fitting" package within IGOR Pro v9.02. Such fitting was not done for the organics' size distributions, as these were too noisy, and the locations of the modal
diameters were estimated manually (i.e., by eye). The fitting modal parameters (location and amplitude of the modes) are given in Table S3.

Equivalent black carbon (eBC) mass concentrations were obtained from the measurement of light attenuation at 880 nm on a filter tape, using an aethalometer model AE33 (Magee Scientific, Berkeley, USA). Description of the AE33 operation during
MOSAiC and of the data processing can be found in Heutte et al. (2023b) and Boyer et al. (2023). The aethalometer and AMS were sampling air through the same inlet. The original 1 Hz eBC data were averaged (A.M.) to 10 min and 1 h time resolutions, complementing the chemical composition obtained from the AMS. The same instrument was used during MOCCHA, with the same data processing procedure. A comparison was performed between the AE33 and a multi-angle absorption photometer (MAAP) for the MOCCHA eBC data and both instruments agreed well, within 20% of each other ($R^2 = 0.77$, not shown).

**2.3 Aerosol number concentration and size distribution measurements**

During MOCCHA, the particle number size distributions (PNSD) of aerosols between 18 and 660 nm ($D_m$) were measured with a custom-made SMPS at a time resolution of 3 min (time for a complete scan through all size bins). Further information on the acquisition and processing of the data is provided by Baccarini and Schmale (2020) and Baccarini et al. (2020). During MOSAiC, a commercial SMPS (TSI Inc., USA) was used to measure the PNSD between 10 and 500 nm (Boyer et al., 2023),
at a time resolution of 5 min (scan time). The instrument was located in the Aerosol Observing System (AOS) container (Uin et al., 2019), operated as part of the United States Department of Energy Atmospheric Radiation Measurement (ARM) facility, 1.5 m away from the *Swiss* container. In this work, PNSD data were used to retrieve the total aerosol volume (in the common size range from 18 to 500 nm), assuming particles are spherical, where data were prior averaged (A.M.) to 1 h time resolution.



When only the MOSAiC data are used, in Sect. 3.2 and 3.3, particle number concentrations (PNC) are reported using the size
range of the MOSAiC SMPS (i.e., between 10 and 500 nm).

## 2.4  Ancillary measurements

The description of the following ancillary measurements only refers to observations made during MOSAiC and the results are presented and discussed in Sect. 3.2 and 3.3.

### 2.4.1 Carbon dioxide measurements

Hourly-averaged (A.M.) carbon dioxide ($CO_2$) dry air mole fractions used in this study result from the merging of several cross-evaluated measurements with cavity ring-down spectroscopes during MOSAiC. Measurements were performed in the University of Colorado (CU) container using a commercial Picarro instrument model G2311-f, on the sea ice at Met City (a few hundred meters away from *Polarstern*) also using a Picarro model G2311-f, and in the *Swiss* container using a Picarro model G2401. Additional discrete whole air samples were collected for post-cruise analysis at the National Oceanic and
Atmospheric Administration (NOAA) Global Monitoring Laboratory (GML) and included in the data merging procedure. Details regarding the instruments' operation, calibrations, data processing, and the creation of the merged dataset can be found in Angot et al. (2022).

### 2.4.2 CCN measurements

Measurements of CCN number concentrations were performed during MOSAiC using a Cloud Condensation Nuclei Counter
(CCNC) model CCN-100 (Droplet Measurements Technologies, Boulder, USA), collocated with the aethalometer and the AMS in the *Swiss* container. The supersaturation (SS) in the instrument's chamber was set to 0.15, 0.2, 0.3, 0.5, and 1% SS, throughout 1 h cycles (Heutte et al., 2023b). CCN number concentrations were averaged (A.M.), for each SS level, to 10 min time resolution (resulting in one 10 min averaged data point for each SS level per hour).

### 2.4.3 Aerosol light scattering measurements

The aerosol total light scattering coefficients at the blue (450 nm), green (550 nm), and red (700 nm) wavelengths were measured during MOSAiC using an integrating nephelometer (TSI model 3563), located in the AOS container. Scattering coefficients were measured at a 1 min time resolution, and corrected to account for size-dependent truncation (incomplete collection) of strongly forward or backward scattered light (Koontz et al., 2022). Using an impactor at the inlet of the external sampling system, the aerodynamic diameter cutoff of sampled particles was alternated between 1 and 10 μm. In this study, we
used the submicron measurements, averaged (A.M.) at 10 min time resolution.



### 2.4.4 Snowdrift density and blowing snow events identification

During MOSAiC, the particle number flux of airborne snow particles was measured, at 1 min time resolution, using two open-path Snow Particle Counters (SPC-95; Niigata Electric Co., Ltd) and used to compute the snowdrift density (Gong et al., 2023). Blowing snow periods were identified (Gong et al., 2023) as times when airborne snow particles were detected and the wind speed measured at 10 m above the snow surface exceeded a critical value, which was empirically estimated from the temperature-dependent parametrization proposed by Li and Pomeroy (1997). The two SPCs, Unit 1104 and Unit 1206, were located at 0.08 and 10 m above the snow surface, respectively. In this study, the SPC at 0.08 m was used to report snowdrift density and derive the blowing snow flag, except for October and November 2019, where the instrument was not operational and data from the SPC at 10 m were used instead.

### 2.4.5 Satellite-derived lead fraction

In our study, we use a published dataset of lead fractions (von Albedyll et al., 2023). In brief, lead fractions, within a 50 km radius from *Polarstern*, were derived from a divergence-based method using satellite synthetic-aperture radar (SAR) data with a spatial resolution of 700 m. The daily available divergence and convergence fields were accumulated for up to 10 subsequent dates to account for leads continuously opening or closing. In this work, we used the divergence-derived lead fractions with no accumulation ($LF_{no\ accu,\ div}$), which represent newly-opened leads, as well as the 5 times accumulated lead fractions ($LF_{5x\ accu,\ div}$), which account for leads opening, closing, or staying open within a 5 days period.

### 2.5 Identification and removal of pollution from ship emissions

Ship-based measurements of some atmospheric variables (including aerosol physicochemical properties) can be greatly influenced by local pollution from research activities (Beck et al., 2022). Exhausts from the ship's stack can be an important source of particles that need to be distinguished from the ambient aerosol signal. Other sources of local contamination, including snowmobiles, diesel generators, and helicopters, can also be important sources of local pollution, discretely affecting the ambient aerosol measurements. Not all instruments react the same way to fresh pollution from fossil fuel combustion. Hence, different pollution detection methods were applied to the various datasets, and are described in detail by Heutte et al. (2023b) for the *Swiss* container aerosol measurements during MOSAiC. In short, AMS measurements were cleaned from local pollution influence by identifying periods where the measured chemical spectrum resembled that of a chosen spectrum of fresh hydrocarbon emissions. This method was applied in analogy to the AMS data from MOCCHA. All the other MOSAiC datasets were cleaned using a multi-step pollution detection algorithm (PDA), developed by Beck et al. (2022). A similar method to the PDA was employed by Baccarini et al. (2020) to remove local pollution from the MOCCHA SMPS data used in this analysis.



### 2.6 Clustering of the particle number size distributions

The measured PNSDs from the SMPS, averaged (A.M.) to 10 min time resolution, were grouped using the Hartigan-Wong k-means clustering algorithm (Hartigan and Wong, 1979), as commonly done for clustering of PNSDs (e.g., Beddows et al., 2009; Boyer et al., 2023; Pernov et al., 2022). This analysis was performed separately for the data from October to November and from March to April, taking the PNSDs normalized to the vector length as input for the algorithm. The number of clusters for the solution was initially varied from 3 to 30 and it was concluded that the 8 clusters solution was best at describing the October-November aerosol size distributions while the 7 clusters solution was optimal for the March-April period. The resulting clusters were further manually merged into 4 clusters for the October-November period and 3 clusters for the March-April period, based on similarities in their potential dominating source (i.e., locally sourced, long-range transported, or low concentration background) and the shape of their median size distribution. Additional information on the criteria for choosing the number of clusters and the manual attributions to more comprehensive "potential-source" groups are provided in Sect. S4. A bimodal log-normal distribution was fitted to each cluster's median PNSD using the "Multipeak fitting" package within IGOR Pro v9.02, and the fitting modal parameters (location and amplitude of the modes) are given in Table S2. The results of the clustering analysis are presented and discussed in Sect. 3.3.1.

### 3 Results and discussion

### 3.1 Bulk submicron aerosol yearly chemical composition

Figure 2a shows the high-time resolution (1 h) annual cycle of bulk submicron aerosol mass concentration and composition measured during the MOCCHA and MOSAiC expeditions. Periods identified as being directly influenced by local pollution from research activities are indicated by grey-shaded areas and are excluded from any subsequent analysis. The unpolluted relative fractions of each species to the submicron aerosol ($PM_1$) mass during different measurement periods are given in Fig. 2b-e. The fractional contribution was derived by summing the mass of each species over the respective period and dividing by the total $PM_1$ mass of that period. In this work, total $PM_1$ is defined as the sum of AMS-based non-refractory $SO_4^{2-}$, Org, $NO_3^-$, Chl, and $NH_4^+$ mass concentrations, and aethalometer-based eBC mass concentrations. Therefore, this does not include submicron dust or fresh and aged sea salt, for example. Figure 3 shows the same species-specific annual cycle as in Fig. 2a but with monthly statistics (median and interquartile range). Note that although the few available AMS data points in December and July were included in the fractional mass contributions in Fig. 2c and Fig. 2e, respectively, these months were not considered in the monthly statistics reported in Fig. 3 due to low data availability. For completeness, the seasonality of total aerosol volume ($V_{tot}$) for particles between 18 and 500 nm in mobility diameter, calculated from the SMPS PNSD, is also shown and used as a proxy for $PM_1$ for months where AMS data are missing, minding that $V_{tot}$ does not include sizes > 500





**Figure 2: Bulk submicron aerosol mass composition measured with the AMS and aethalometer during the MOCCHA and MOSAiC expeditions.** The year-long timeseries of these species (a) is shown as the 1 h averaged (A.M.) total mass concentration from August to September 2018 (MOCCHA) and from October 2019 to July 2020 (MOSAiC). Periods identified as being affected by local pollution from research activities are indicated with vertical grey areas. The unpolluted relative contributions of the main aerosol species to the total summed mass concentration are shown during MOCCHA for the (b) Aug-Sep 2018 period, and during MOSAiC for the (c) Oct-Dec 2019 period, (d) Mar-May 2020 period and (e) Jun-Jul 2020 period. Ammonium was not considered for the Jun-Jul period due to instrumental issues. Note that, during MOSAiC, the AMS was not operational between December 2019 and March 2020.

nm. This annual cycle is segregated into five distinct periods: August to September 2018, October to December 2019, January

to February 2020, March to May 2020, and June to July 2020, which are analyzed and discussed in sections 3.1.1 - 3.1.5, respectively. The start and end dates of these periods were determined from (1) the different physicochemical processes associated with each period and the resulting contrasted aerosol mass concentrations and composition, and from (2) the data availability imposed by the expeditions' timing and instruments' functioning. Furthermore, we argue that the MOCCHA data from summer 2018 can be considered representative of the central Arctic Ocean summer conditions, and hence used to replace

the missing MOSAiC summer (2020) data, for the following reasons. First, long-term observations at coastal Arctic land-based stations have revealed minimal interannual variability in summer $SO_4^{2-}$ and BC mass concentration (Gong et al., 2010) or total



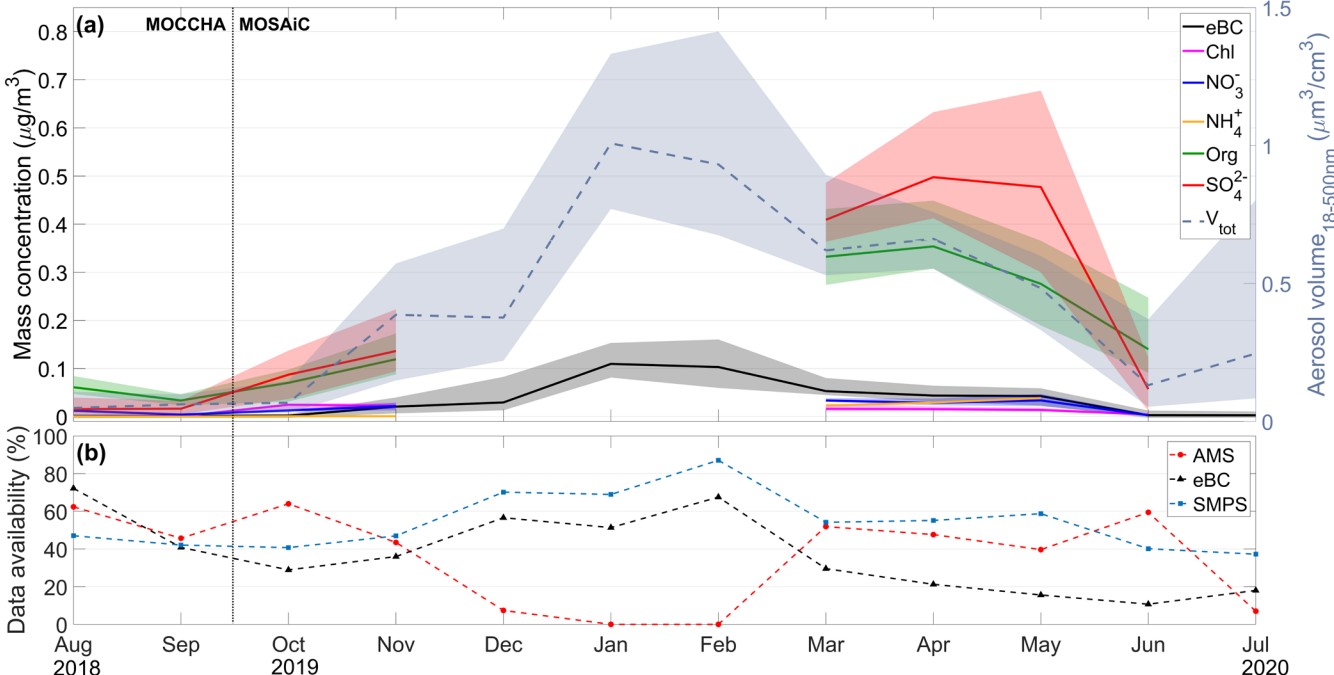

**Figure 3: Monthly seasonality of the bulk submicron aerosol mass composition measured with the AMS and aethalometer during the MOCCHA and MOSAiC expeditions.** Monthly medians are shown in (a), along with the 25th (75th) quantile as lower (upper) envelop boundary, for the different chemical species and for the total aerosol volume (18-500 nm) calculated from the SMPS' PNSD. The monthly statistics, calculated from the hourly-averaged concentrations, consider only unpolluted data and the percentages of available data per month and per instrument (i.e., data retained after the quality and pollution flags have been applied to the datasets) are shown in (b). August and September are from the MOCCHA expedition, other months are from MOSAiC. The dashed vertical line indicates the separation from the two distinct datasets. Median mass concentrations from the AMS are not reported for December 2019 and July 2020 due to low data availability (< 10%).

aerosol mass (Tunved et al., 2013). Second, as MOCCHA and MOSAiC summers were separated by two years only, the influence of long-term trends in species mass concentrations can be neglected, especially as there are not so many statistically significant trends in summer (Schmale et al., 2022).

### 3.1.1 Aug-Sep: late summer (MOCCHA)

In late summer (August 1st – September 15th, 2018, during MOCCHA), organics had the largest mass contribution to the total $PM_1$ (62%, Fig. 2b), followed by $SO_4^{2-}$ (28%), $NO_3^-$ (8%), and eBC (2%). $NH_4^+$ and Chl were mostly below the detection limits (Table S1) and will hence not be further discussed in this section. The highest median mass concentrations during this period were found in August (see Fig. 3a), with medians (25th quantile, 75th quantile) of 0.06 (0.05, 0.08) $\mu g/m^3$ and 0.016 (0.006, 0.040) $\mu g/m^3$ for Org and $SO_4^{2-}$, respectively. Overall, we observed very low mass concentrations for the different species at this time of the year, also reflected in the low aerosol volume (18-500 nm; median = 0.049 $\mu m^3/cm^3$ in August, and 0.062 $\mu m^3/cm^3$ in September). Such low aerosol mass loadings are typical and characteristic of the high Arctic during this season (Leaitch et al., 2018; Massling et al., 2015; Schmale et al., 2022; Ström et al., 2003). This is likely the combined result





of (1) limited long-range transport of aerosols from lower latitudes, as the extent of the Arctic dome is small (Stohl, 2006), (2)
efficient wet and dry removal of locally emitted and transported aerosols, as previously observed and modeled (Browse et al.,
2012; Freud et al., 2017; Pernov et al., 2022), and (3) the position of *Oden* high up in the pack ice (> 88° N, see Fig. 1), far
away from most open-ocean marine and terrestrial sources. Chang et al. (2011) reported similar mass concentrations of
organics (median Org = 0.055 µg/m$^3$) at such high latitudes in August - September 2008, although with a higher sulfate fraction
to the total mass of 45% (median SO$_4^{2-}$ = 0.051 µg/m$^3$). Previous studies have nonetheless shown the importance of
local/regional natural emission sources for the central Arctic at this time of the year (e.g., Chang et al., 2011; Hamacher-Barth
et al., 2016; Heintzenberg et al., 2015), in particular during MOCCHA (Baccarini et al., 2020; Siegel et al., 2021), with
relevance towards cloud formation (Bulatovic et al., 2021; Duplessis et al., 2024; Karlsson et al., 2022). With the present
observations, we further emphasize the major contribution of organics, which are likely naturally-sourced, for the central Arctic
submicron aerosol budget in summer.

### 3.1.2 Oct-Dec: autumn (MOSAiC)

October marked the beginning of the dark season in the central Arctic, and temperatures dropped significantly (Shupe et al.,
2022). In the transition from summer to autumn, we observed a drastic change in aerosol chemical composition, whereby SO$_4^{2-}$
became the dominant measured species by mass (47%, Fig. 2c), followed by Org (31%), Chl (8%), eBC (7%), and NO$_3^-$ (6%).
NH$_4^+$ was mostly below detection limit. The combined increase in the fraction of SO$_4^{2-}$ and eBC, and decrease in the fraction
of Org, compared to summer, is indicative of changes in the aerosol sources, sinks, and processing. The ongoing ocean freeze-
up, coupled with the dark conditions, gradually decrease the influence of local marine aerosol sources (Leck and Persson,
1996; Moschos et al., 2022b; Schmale et al., 2022), while long-range transport of anthropogenic pollutants from lower latitudes
occurs more readily towards the winter (Boyer et al., 2023; Moschos et al., 2022b; Quinn et al., 2009). During MOSAiC,
November and December experienced several storms (see the annual wind speed measurements from the 2D ultrasonic
anemometer onboard *Polarstern* (Schmithüsen, 2021a, b, c, d, e) in Fig. S2), which have been shown to greatly increase the
number of fine sea salt (NaCl) aerosols, associated with the sublimation of salty blowing snow and/or sea spray aerosol (SSA)
emissions from open leads (Chen et al., 2022; Gong et al., 2023). The elevated chloride fraction seen here may be related to
NaCl but the interpretation remains difficult because of the limited ability of the AMS to measure it (see Sect. 2.2). Another
possibility is non-refractory chloride that partitioned into the particles (Hara et al., 2002). Using instead the NaCl$^+$ fragment at
*m/z* 58 (Ovadnevaite et al., 2012), we discuss in Sect 3.2 and 3.3 the contribution of sea salt in the autumn aerosol budget. Due
to the logistical complexities associated with in situ measurements in the central Arctic Ocean at this time of the year, reports
of aerosol chemical composition are scarce, which challenges the comparison of our dataset with others. However,
observations of aerosol number concentrations and size distributions from lower latitude land-based stations commonly
reported October as the yearly minimum in PNC, as a result of enhanced wet removal and limited long-range transport of
pollutants (e.g., Croft et al., 2016b; Freud et al., 2017; Nguyen et al., 2016; Pernov et al., 2022; Tunved et al., 2013). In line
with these studies, the yearly total aerosol number concentration also reached a minimum in October during MOSAiC (Boyer





et al., 2023), likewise for $V_{tot}$ (median = 0.067 µm$^3$/cm$^3$, Fig. 3a). In November, the median mass concentrations of Org and SO$_4^{2-}$ reached their seasonal maximum (i.e., for autumn), with medians (25$^{th}$ quantile, 75$^{th}$ quantile) of 0.12 (0.09, 0.17) µg/m$^3$ and 0.136 (0.094, 0.223) µg/m$^3$, respectively. Although limited, we show in Sect. 3.2.1 based on our high-time resolution dataset, that long-range transport of anthropogenic pollutants is an important and climate-relevant contribution to the central Arctic aerosol budget in autumn likely representing the start of the well-known haze period.

### 3.1.3 Jan-Feb: winter (MOSAiC)

Wintertime during MOSAiC (here defined as January and February 2020) marked the peak of the year 2020 haze season, with the highest yearly median aerosol total volume and eBC mass concentration observed in January (1.008 µm$^3$/cm$^3$ and 0.11 µg/m$^3$, respectively). This unusually early timing for the occurrence and intensity of Arctic haze has been discussed by Boyer et al. (2023) and was attributed to a record-breaking positive phase of the Arctic Oscillation (AO) between January and March 2020 (Lawrence et al., 2020), leading to enhanced air mass transport from lower latitudes to the central Arctic. The authors demonstrated the importance of Russia/Siberia as a pollution source for eBC and accumulation mode aerosol number concentrations ($N_{100-500}$) during these two months, with both eBC and $N_{100-500}$ reaching their annual maxima in January. As already mentioned, the AMS was not measuring at this time of the year due to instrumental malfunctions. Hence, it was not possible to derive any chemical composition information except for BC; however, given the similar fractional mass chemical composition for the neighboring months (see Fig. 2c-d), it is likely that SO$_4^{2-}$ was the dominant non-refractory species by mass. Nonetheless, due to a low abundance of photochemically-produced oxidants in the dark winter conditions and the limited cloud liquid water for aqueous-phase reactions in the high Arctic, we could also expect that primary emissions such as BC or primary anthropogenic Org (Moschos et al., 2022b) dominated over secondary processes that would produce particulate sulfate (Schmale et al., 2022) and secondary organics. The two latter species have their peak contribution in March and April across Arctic observatories (Moschos et al., 2022a; Schmale et al., 2022). Wintertime oxidation pathways involving, for instance, metal-catalyzed in-cloud oxidation of SO$_2$ by O$_2$ (Alexander et al., 2009; McCabe et al., 2006) and poleward transport of SO$_4^{2-}$ formed at lower latitudes where sunlight is available for photo-oxidation, could have, however, still resulted in SO$_4^{2-}$ being a dominant species in the dark January and February months, in the context of anomalously high positive AO. This is also supported by recent findings from Boyer et al. (2024), who found a close agreement between measured high SO$_2$ mixing ratios and simulated SO$_4^{2-}$ mass concentrations using the ECLIPSE v6b emission inventory coupled with back-trajectories, in January and February 2020 at *Polarstern*'s location. Primary SO$_4^{2-}$ (i.e., emitted as fully oxidized from coal and oil-burning stacks), which was found by Moon et al. (2023) to be the dominant source of SO$_4^{2-}$ in a polluted city of the Alaskan sub-Arctic, could have also contributed to the central Arctic winter SO$_4^{2-}$ budget, since the process likely applies to other locations around the Arctic including Siberia. Without additional observational evidence, this will not be discussed further, and the focus will be turned towards spring Arctic haze chemical characterization (see Sect. 3.1.4).





### 3.1.4 Mar-May: spring (MOSAiC)

The spring season (March - May) was characterized by elevated background PM$_1$ concentrations, where SO$_4^{2-}$ contributed by
50 % to the measured mass, followed by Org (36%), eBC (6%), NH$_4^+$ (3%), NO$_3^-$ (3%), and Chl (2%). This pattern is
representative of the well-studied Arctic haze phenomenon (Nielsen et al., 2019; Quinn et al., 2007). Not only were the
background concentrations elevated, but a number of high mass concentration events were observed, such as on March 15[th]
when PM$_1$ mass concentration neared 2 μg/m$^3$, and during two intense episodes of warm and moist air mass intrusions from
northern Eurasia on April 15[th] and 16[th], where pollution levels were so high ([PM$_1$] ≥ 4 μg/m$^3$) that they became comparable
to central-European urban pollution levels (Dada et al., 2022a). Dada et al. (2022) showed that sudden direct transport of
pollution to the central Arctic can have important impacts on aerosol climate-relevant properties (i.e., acidity, oxidation state,
and hence hygroscopicity). The highest monthly median mass concentrations in spring were found in April, with medians (25[th]
quantile, 75[th] quantile) of 0.35 (0.31, 045) μg/m$^3$ and 0.50 (0.41, 0.63) μg/m$^3$ for Org and SO$_4^{2-}$, respectively. At this time of
the year, atmospheric conditions favored transport from lower latitudes compared to summer (Bozem et al., 2019), and Boyer
et al. (2023) found that the surface aerosol population was largely influenced by transport from Siberia in spring during
MOSAiC. The prevalence of SO$_4^{2-}$ observed here corroborates that Russia/Siberia is an important source of pollution to the
central Arctic haze burden (Hirdman et al., 2010; Petäjä et al., 2020), as industrial activities in these regions (mainly metal
smelters) are known to be important sources of atmospheric sulfur (Sipilä et al., 2021). We also measured relatively low NH$_4^+$
concentrations at the surface. Observational and modelling studies have shown strong vertical gradient of NH$_4^+$/SO$_4^{2-}$ ratio in
the springtime Arctic, with higher concentration of NH$_4^+$ in the upper troposphere/lower stratosphere than in the boundary
layer, resulting from a stronger contribution of East Asian anthropogenic (agricultural) NH$_4^+$ emissions at higher altitudes
(Fisher et al., 2011; Willis et al., 2019). Together, these observations suggest that submicron aerosols measured in the
springtime at the surface are very acidic, with potential implications for the partitioning of gaseous organic acids to particle
phase, as observed for MSA during MOSAiC (Dada et al., 2022a). In May, SO$_4^{2-}$ concentrations remained high, especially at
the beginning of the month, when large-scale vertical mixing associated with the collapse of the polar vortex could have
introduced large quantities of aged particles into the troposphere from aloft (Ansmann et al., 2023). Natural sources of sulfur
species from DMS oxidation had a growing contribution to gaseous sulfur compounds (MSA) during this month and towards
summer (Boyer et al., 2024), with the initiation of the summer sea ice melt in late May (Shupe et al., 2022).The detailed aerosol
chemical and geographical sources during haze, especially those of organics, will be presented in a follow-up source
apportionment study.

The collapse of the polar dome in late April – early May (Ansmann et al., 2023), along with a likely transition of mixed-phase
clouds containing a higher fraction of liquid droplets and a transition from less efficient ice-phase to more efficient liquid-
phase in-cloud particle scavenging for late spring warm clouds (Browse et al., 2012), marked the end of the Arctic haze season
and a sharp regime transition (within a few days) towards summertime.



### 3.1.5 Jun-Jul: summer (MOSAiC)

In June and July 2020, the aerosol chemical composition during MOSAiC resembled that from August - September during MOCCHA. Organics became the dominant species in terms of mass (63% of $PM_1$), followed by $SO_4^{2-}$ (32%), eBC (3%), and Chl (1%). $NH_4^+$ was excluded from the analysis during these months due to instrumental issues (see Sect. 2.2) and $NO_3^-$ was

mostly below detection limit. In June, median (25th quantile, 75th quantile) concentrations of Org and $SO_4^{2-}$ were 0.14 (0.09, 0.25) µg/m³ and 0.057 (0.012, 0.122) µg/m³, respectively. The sharp decrease in $SO_4^{2-}$ mass concentration and $V_{tot}$ (median = 0.131 µm³/cm³) in June is indicative of reduced air mass transport from anthropogenically polluted regions due to the poleward contraction of the polar dome. Enhanced wet scavenging from increased precipitation, with intermittent wash out events, can also partly explain the lower mass concentrations. The combination of these two processes (i.e., change in air mass transport

and increased wet scavenging) can also explain the observed very low eBC mass concentrations in June and July (median = 3 ng/m³, for both months). One important feature here is the large variability in $V_{tot}$ in July (interquartile range = 0.719 µm³/cm³), which could be indicative of intermittent events of transport or local release of organic material from melt ponds, the marginal ice zone, or nearby coastal and open ocean areas (Chang et al., 2011). In general, it is likely that organic aerosols at this time of the year are dominated by natural marine and terrestrial biogenic sources, through the emission of primary aerosols (e.g.,

bacteria, viruses, diatoms, and fragments of them) and formation of secondary aerosols from gaseous compounds (e.g., MSA, mono- and sesquiterpenes, and isoprene) (Moschos et al., 2022b). A follow-up source apportionment study will elucidate the sources associated with organic aerosols in the summertime central Arctic during MOSAiC.

### 3.1.6 Comparison of MOSAiC and MOCCHA observations to pan-Arctic land-based stations

To understand potential spatiotemporal variability, we compared our yearly chemical composition observations with six land-

based stations' measurements from around the Arctic (Fig. 4): Alert, Canada (ALT), Baranova, Russia (BAR), Gruvebadet, Svalbard/Norway (GRU), Pallas, Finland (PAL), Villum, Greenland (VRS), and Zeppelin, Svalbard/Norway (ZEP), with measurements from 2015 to 2019 depending on the station (see Fig. 4 caption for details). Further information on the location of these stations, sampling methods, and description of their yearly cycles of chemical composition were presented and discussed by Moschos et al. (2022a). The results of this comparison need nonetheless to be interpreted with caution, for several

reasons: (1) the sampling method differed substantially, offline analysis of weekly to bi-weekly filter samples by ion chromatography for inorganic ions and an OC/EC sunset analyzer for organics was employed for the pan-Arctic datasets while an AMS was used during MOCCHA and MOSAiC; (2) the sampling sites for the pan-Arctic datasets are located at lower latitudes than those at which MOSAiC and MOCCHA took place; (3) the cutoff size for the sampling inlets was different: 10 µm ($PM_{10}$) for the filter samples and 1 µm ($PM_1$) for the AMS; (4) finally, for the anomaly calculations, the yearly mean value

for MOCCHA and MOSAiC could be biased by the fact that data were missing in December, January, February, and July. Despite this, intercomparisons of the trends in the chemical species were still possible.




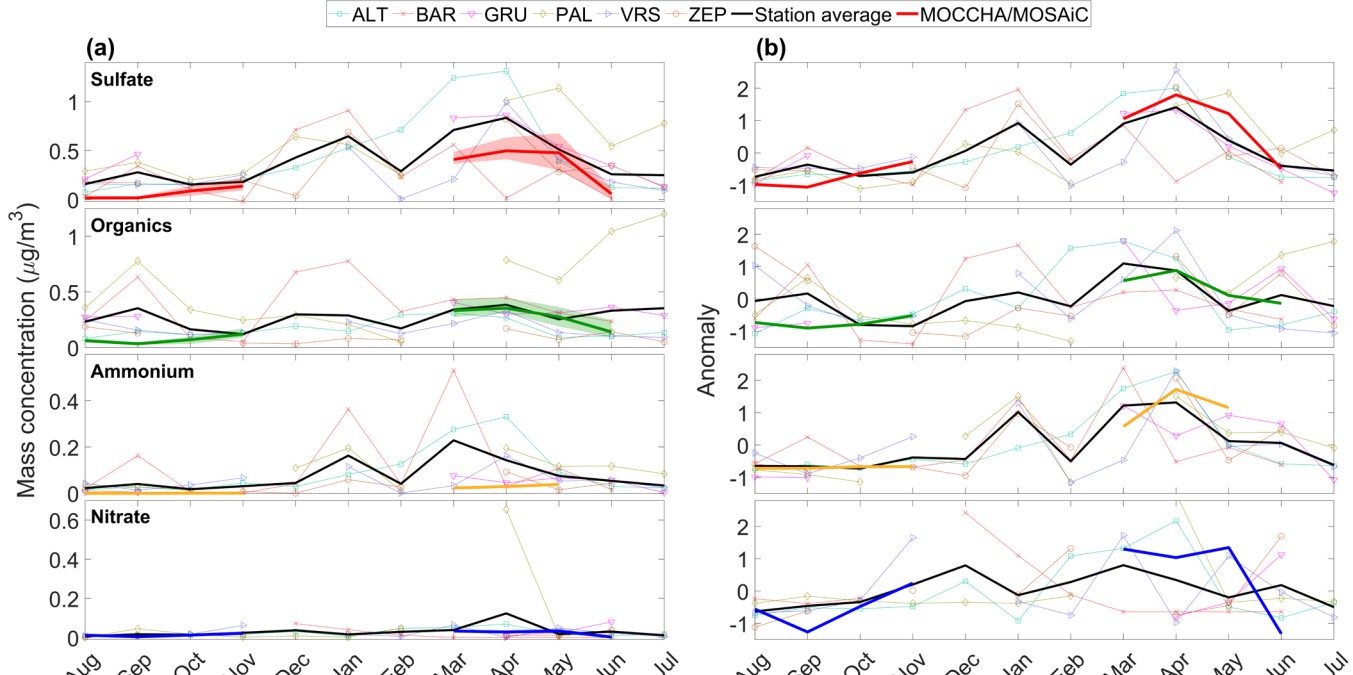

**Figure 4: Comparison of the seasonal cycles of absolute (a) and standardized (b) monthly mass concentration of submicron non-refractory sulfate, organics, ammonium, and nitrate from MOCCHA (Aug-Sep) and MOSAiC (Oct-Jun) to filter-based PM$_{10}$ measurements from six Arctic land-based stations (Moschos et al., 2022a).** The stations (and sampling periods) are the following: ALT (Apr 2015 – Dec 2018), BAR (Apr 2015 – Nov 2016), GRU (Mar 2017 – Aug 2018), PAL (Aug 2018 – Aug 2019), VRS (Dec 2017 – Dec 2018), and ZEP (Jan 2017 – Dec 2018). The anomalies in panel (b) were calculated using the absolute mass concentration values as: (monthly_mean_site – annual_mean_site) / annual_standard_deviation_site. The thin colored lines correspond to each stations' yearly cycle, and thick colored lines represent the MOCCHA and MOSAiC data, with the 25$^{th}$ and 75$^{th}$ quantiles as the shaded envelop for the mass concentration. The thick black lines correspond to the stations' average for each chemical species. For MOCCHA and MOSAiC, data identified as affected by local contamination were not considered in the computation of the monthly statistics.

Sulfate, organics, and nitrate were, on average, within the same range of absolute mass concentrations in the central Arctic as at the land-based stations, although geographical variability was evident (e.g., high SO$_4^{2-}$ concentration at ALT in spring or high Org concentrations at PAL in summer) and expected (Schmale et al., 2021). These similarities are remarkable when considering the differences in the sampling conditions described above between the two datasets. An exception were August and September during MOCCHA, where both SO$_4^{2-}$ and Org (median SO$_4^{2-}$ = 0.016 µg/m$^3$ and median Org = 0.050 µg/m$^3$) were consistently lower than at the various land-based stations (median SO$_4^{2-}$ = 0.218 µg/m$^3$ and median Org = 0.293 µg/m$^3$, for the August-September stations' average). As discussed in Sect. 3.1.1, *Oden* was close to the north pole and deep in the pack ice which partly isolated it from most remote natural and anthropogenic sources. SO$_4^{2-}$ during springtime also exhibited lower concentrations in the central Arctic, during MOSAiC (median = 0.446 µg/m$^3$, for March-May), compared to other land-based stations (median = 0.697 µg/m$^3$). Interestingly, at the same time, Org levels (median = 0.329 µg/m$^3$) were relatively similar to the stations' measurements (median = 0.334 µg/m$^3$). This could possibly suggest an enhanced sink term of SO$_4^{2-}$ during transport to the central Arctic, provided that SO$_4^{2-}$ and Org sources were the same (i.e., anthropogenic). Alternatively,



the fraction of sulfate in the coarse mode (PM$_{10}$) could have been larger than that of organics, which could explain the difference between the PM$_1$ MOSAiC observations and the PM$_{10}$ pan-Arctic observations. NO$_3^-$ was generally low, both in the central Arctic (yearly median = 0.017 µg/m$^3$) and at the stations (yearly median = 0.021 µg/m$^3$). In the case of MOCCHA and MOSAiC, this was potentially furthered by the PM$_1$ limitation, since a large fraction of NO$_3^-$ is expected to be found in supermicron-sized, and more alkaline, sea salt particles (Cavalli et al., 2004; Fenger et al., 2013; Mukherjee et al., 2021; Ricard

et al., 2002; Saltzman, 2009). A striking difference was observed for ammonium, which was found to be consistently less abundant throughout the year in the central Arctic (yearly median = 0.001 µg/m$^3$) compared to the land-based stations (yearly median = 0.043 µg/m$^3$), especially in spring. This results in generally more acidic aerosols in the central Arctic. Differences could be explained by a stronger contribution of ammonia emissions at Arctic coastal sites from migratory seabird colonies (Croft et al., 2016a), as well as different spatiotemporal NH$_4^+$ contributions from open biomass burning events in the Arctic or sub-Arctic regions (Gramlich et al., 2024). In light of the decreasing sulfate concentrations in the Arctic (Schmale et al., 2022),

efforts should be maintained to rigorously monitor aerosol chemical composition in the future, as a range of aerosol physicochemical processes depend on the particles' acidity (Pye et al., 2020), for example, the partitioning of nitrate into the particle phase (Sharma et al., 2019), which tends to increase as the sulfate-to-ammonium ratio decreases. Regarding the seasonality of the anomaly values (Fig. 4b), the haze signal peaking in March/April appeared to be similar for all species

between MOSAiC and the pan-Arctic station average. As stated above, the summer peak for Org was not observed during MOCCHA. This comparative study shows that long-term observations at Arctic land-based stations are relevant to the central Arctic seasonal cycle of chemical composition and mass loading. Differences are nonetheless noticeable, in particular for ammonium, which seems to be far less abundant in the central Arctic throughout the year, as well as sulfate and organics in summer.

**3.2 Case studies on storm-driven locally-emitted and long-range transported aerosols**

Compared to the relatively low-time resolution imposed by aerosol filter sampling, the present year-long MOCCHA/MOSAiC dataset also offers unique opportunities to study aerosol processes happening on short timescales which can elucidate important aspects other than the large-scale features of e.g., Arctic haze. In particular, the MOSAiC dataset covers seasons with high time resolution observations other than summer, where previous central Arctic measurements are already available (e.g., Chang

et al., 2011; Karlsson et al., 2022; Lawler et al., 2021). Our dataset allows us to answer several questions: Are there any significant changes in aerosol chemical composition on shorter timescales over the central Arctic Ocean? If so, what are the sources of the particles, and what are their contributions to the CCN population and direct radiative budget? How often do we observe significant and sudden increases in aerosol mass and number concentrations, and with how much deviation from the background conditions or monthly medians/means? How long do these episodes last, what drives them, and what is their

impact? In the following Sect. 3.2.1 and 3.2.2, we address these questions by means of case study analyses.



### 3.2.1 High-resolution case studies from autumn 2019

Several storms occurred in autumn 2019 during MOSAiC (Rinke et al., 2021), from which two major ones in November are presented in Fig. 5. Given the relatively low aerosol number concentrations (Boyer et al., 2023), the central Arctic climate system at this time is expected to be particularly sensitive to the aerosol population. The combination of high-time resolution
chemical composition (Fig. 5c, d), dynamical and physicochemical source markers (Fig. 5a, b, e, f), optical properties and CCN number concentrations (Fig. 5g, h) discussed below is important to uncover the sources that contributed to the aerosol and CCN populations in the overlooked dark autumn period. The measured chemical species shown in Fig. 5c, d (i.e., $SO_4^{2-}$, Org, NaCl and eBC) showed distinct temporal evolution during the two storms. While NaCl and eBC were correlated with the local wind speed, suggesting a wind-dependent aerosol generation as a source, $SO_4^{2-}$ and Org correlated more with $CO_2$,
indicating that these species were likely primarily long-range transported. The discussion hereafter will hence be separated in two, first addressing the contribution from local sources, then the contribution from remote sources. Note that all times reported are in UTC.

*Wind-dependent aerosol generation as a local source of aerosols:*

During the first storm, the wind speed, measured onboard *Polarstern*, started to increase on November 10[th], reaching a maximum on the 11[th] with values above 16 m/s. Blowing or drifting snow was detected (see Sect. 2.4.4) without interruption between the 11[th] at 1:30 and the 12[th] at 12:00. Within this period, a strong increase in supermicron PNC ($N_{>1000nm}$) was observed, following, with a 3-hour lag, the increase in wind speed (3-hour-lag Pearson correlation ($\rho_{pearson}$) = 0.87 p-value < 0.001). Similarly, sea salt mass concentrations followed closely the $N_{>1000nm}$ signal. Despite the $PM_1$ limitation of the AMS,
the fact that the submicron NaCl signal correlated greatly with $N_{>1000nm}$ ($\rho_{pearson}$ = 0.89, p-value < 0.001) is an indication that sea salt was likely present in the blowing snow, as expected (Frey et al., 2020; Gong et al., 2023). It is also likely that a fraction of the observed increase in submicron NaCl signal originated from wind-driven SSA emissions from neighboring open leads in the sea ice, as has been observed elsewhere (e.g., Chen et al., 2022; Kirpes et al., 2019; Myers et al., 2021; Nilsson et al., 2001; Radke et al., 1976), especially since storms are associated with mechanical deformation of the sea ice and leads opening
(von Albedyll et al., 2023). However, as shown in Fig. 5a, the lead fraction (spatial resolution of 700 m) within a 50 km radius of *Polarstern* was less than 1% during the storm. Hence, comparing the relative surface area of open leads to that of sea ice covered by salty snow (i.e., well above 95%) submicron NaCl emissions from salty blowing snow conceivably can dominate over SSA emissions from leads. A recent modelling study suggested an anti-phased seasonal contribution of leads and blowing snow to sea spray fluxes in the high Arctic, with leads being the dominant source of sea salt in terms of mass in summertime
and blowing snow being dominant in winter (Lapere et al., 2024). Furthermore, it cannot be entirely excluded that the observed wind-driven increase in $N_{>1000nm}$ and submicron NaCl came from longer range transported SSA from the ice-free Arctic Ocean. As a sea salt source apportionment is impossible here, the increase in its signal has to be seen as a mixed contribution from



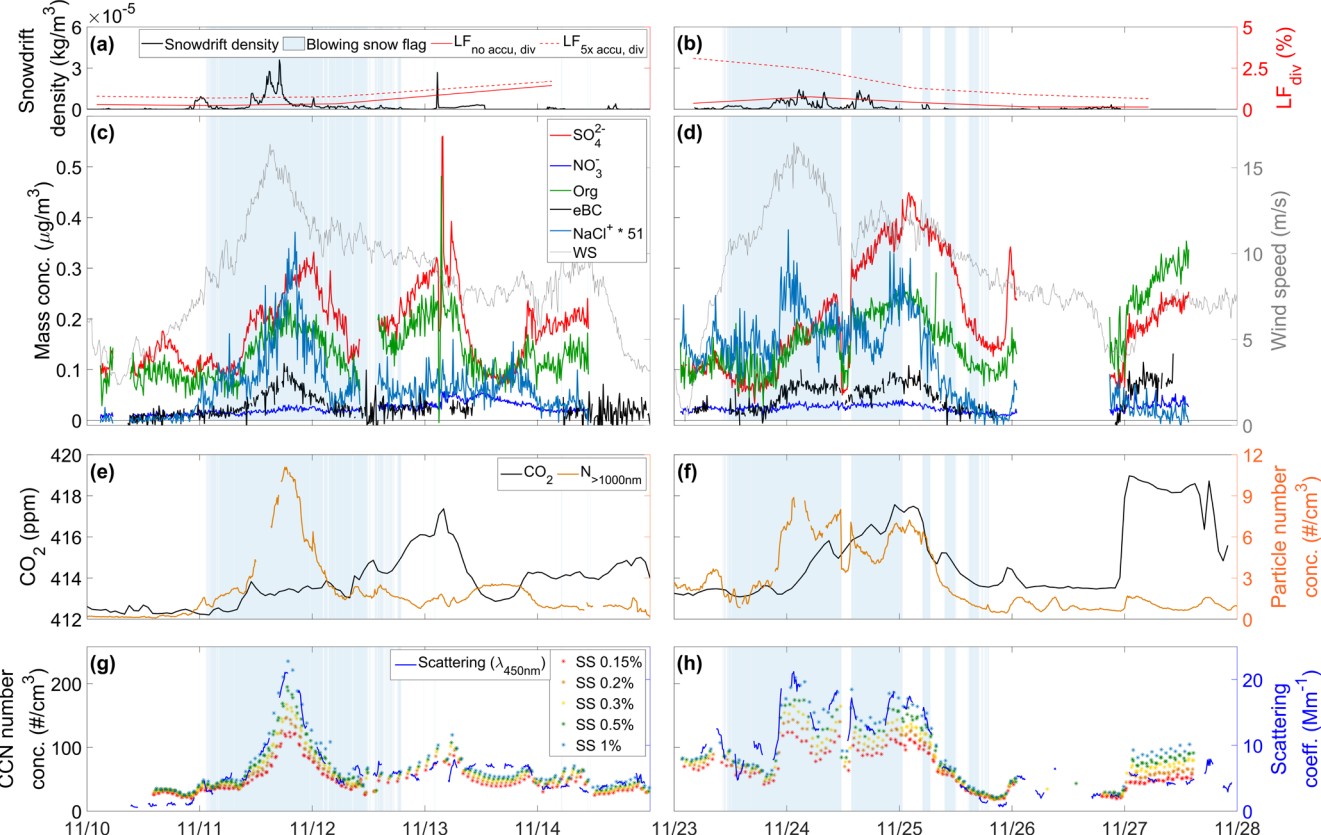

**Figure 5: High-time resolution case studies of two storms in November 2019 during MOSAiC.** The snowdrift density at 10 m above ground and the 50 km radius accumulated divergence-derived lead fractions with no accumulation ($LF_{no\ accu,\ div}$) and 5x accumulated ($LF_{5x\ accu,\ div}$) are shown in (a) for the first storm from November 10th to 15th, 2019, and in (b) for the second storm from November 23rd to 28th, 2019. The aerosol chemical composition, eBC mass concentrations, and wind speed are shown in (c) and (d) for the first and second storm, respectively. NaCl signals are in arbitrary units. $CO_2$ dry air mole fraction and coarse mode particle number concentrations are shown in (e) and (f) for the first and second storm, respectively. CCN number concentrations and total light scattering coefficient at the blue wavelength are shown in (g) and (h) for the first and second storm, respectively. For all panels, a blue shading indicates periods when blowing/drifting snow was detected. All measurements were averaged (A.M.) to 10 min time resolution, expect for $CO_2$ which is hourly and $LF_{div}$ which depends on satellite data availability. Data identified as affected by local contamination (pollution) were removed.

various wind-dependent emission sources, where blowing snow may be the dominant one. Hence, for any further references to blowing snow, we implicitly include wind-generated SSA as a potential additional contribution to our observations.


Compared to the background period prior to the event from 12:00 to 22:00 on the 10th, we observed an increase in $N_{>1000nm}$ by a factor of ~54 (from 0.2 to 10.7 cm$^{-3}$) and by a factor of ~87 (from 0.003 to 0.26 a.u.) for NaCl during the 2 hours peak of the storm (i.e., between 18:00 and 20:00 on the 11th). Since supermicron particles are mainly related to primary particles formed by mechanical processes (Seinfeld and Pandis, 2016), in this case wind-generated, the comparison of NaCl signal was made

with $N_{>1000nm}$ rather than with submicron PNC ($N_{10-500nm}$) from the SMPS, which would be influenced by other sources such as long-range transport. Nonetheless, we show in Fig. S3a a comparison between $N_{>1000nm}$ and $N_{10-500nm}$ during the storm, where



the two were highly covariant, especially during the blowing snow episode ($\rho_{pearson}$ = 0.99, p-value < 0.001). The wind-driven increase in $N_{>1000nm}$ and NaCl mass concentrations resulted in a proportional increase of the CCN number concentrations (shown for SS levels from 0.15 to 1% in Fig. 5g). We found correlations ($\rho_{pearson}$) between NaCl signal and CCN number concentrations between 0.84 and 0.88 depending on the SS level (all p-values < 0.001) during blowing snow, with increases in CCN number concentration during the storm peak compared to background by factors of ~4 (from 27.0 to 119.3 cm$^{-3}$), ~5 (from 30.2 to 144.5 cm$^{-3}$), ~5 (from 32.1 to 161.3 cm$^{-3}$), ~6 (from 32.4 to 186.0 cm$^{-3}$), and ~7 (from 33.1 to 228.3 cm$^{-3}$), at SS levels of 0.15, 0.2, 0.3, 0.5, and 1% respectively. The larger increase for higher SS levels is indicative of the presence of Aitken mode particles (as seen in Fig. S3a with the stronger increase in 10-80 nm particles compared to the 80-200 or 200-500 nm ones) generated from blowing snow and SSA, which are only activated in the instrument when the SS is high enough to overcome the high curvature of these small particles (Kelvin effect). In ambient autumn conditions (i.e., not in the artificial conditions of the CCNC), high values of maximum cloud supersaturation (> 1%) are likely to happen (Duplessis et al., 2024; Motos et al., 2023), making the Aitken mode fraction of blowing snow-related particles climate-relevant. The strong enhancement of CCN number concentrations from fine sea salt particles associated with blowing snow has been shown by Gong et al. (2023) for several blowing snow events during MOSAiC in autumn and winter. The authors further estimated, from model simulations including sea salt aerosol generation from blowing snow, that the increase in CCN number concentrations associated with blowing snow led to an increase of the downwelling longwave radiation of about +2.3 W m$^{-2}$ under cloudy sky conditions from November to April. We also observed a blowing snow-related increase in the total submicron aerosol light scattering coefficient (shown for the blue wavelength in Fig. 5g), tightly following the NaCl signal time series ($\rho_{pearson}$ = 0.90, p-value < 0.001). Compared to the November 10[th] background, the scattering coefficient increased by a factor of ~21 at the storm's peak (from 1.0 to 21.0 Mm$^{-1}$). The production of wavelength-dependent scattering particles during blowing snow episodes would be specifically relevant for radiative forcing at lower latitudes of the central Arctic and other times of the year, where sunlight is present.

Overall, the same strong relations between wind speed, $N_{>1000nm}$, NaCl signal, scattering coefficient, and CCN number concentrations were observed for the second storm case (Fig. 5b, d, f, h), where blowing snow was identified from November 23[rd] at 10:00 to the 25[th] at 1:00, with an intermittent break in the storm on the 24[th] from 11:30 to 13:30. Defining the background period from 1:00 to 10:00 on the 23[rd] (i.e., just before the blowing snow event) and the storm's peak from 1:00 to 3:00 on the 24[th], we find an increase in NaCl signal by a factor ~2 (from 0.14 to 0.25), for $N_{>1000nm}$ by a factor ~4 (from 2.4 to 8.5 cm$^{-3}$), for the scattering coefficient by a factor ~2 (from 10.7 to 19.4 Mm$^{-1}$), and for CCN number concentration by factors of ~2 at all SS levels (e.g., from 26.9 to 203.3 cm$^{-3}$, at 1% SS). The background concentration seemed nonetheless to be elevated already before the blowing snow event. If we consider the background period from 15:00 to 22:00 on the 25[th] (i.e., after the blowing snow event, when the influence of wind speed on the considered variables seemed minimized), we found relative increases by factors of ~25, 11, and 15, for the NaCl signal, $N_{>1000nm}$, and the scattering coefficient, respectively, and between ~5 and ~8 for CCN number concentrations (the increase being larger at higher SS levels). Although the relative increases



differed between the two storms, due to the different background conditions, the absolute values reached during the storms were very similar. In agreement with Gong et al. (2023), the large deviations from the relatively pristine background suggest that blowing snow episodes are an important, but intermittent, source of scattering particles and CCN in autumn in the central Arctic. Further analysis is needed to better quantify these impacts.

635

Another major observation during both storms was the strong correlation between NaCl and eBC during blowing snow ($\rho_{pearson}$ = 0.74, p-value < 0.001, and $\rho_{pearson}$ = 0.59, p-value < 0.001, for the first and second blowing snow events, respectively). This indicates that eBC was possibly contained in the sublimated particles from blowing snow. It should be noted here that the eBC measurements could be slightly overestimated due to enhancement of light absorption in the filter matrix under the presence of strongly scattering particles (associated with a high single-scattering albedo (SSAlb; Drinovec et al. (2022)). However, during both storms, the SSAlb was below 0.94 (not shown), which is below the 0.99 threshold where strong bias emerges, as experimentally determined by Drinovec et al. (2022). Compared to background conditions, the mass concentration of eBC increased by factors of ~12 (from 0.008 to 0.092 $\mu g/m^3$) and ~4 (from 0.016 to 0.066 $\mu g/m^3$) during the first and second storm, respectively, reaching levels comparable to Arctic haze conditions (see Sect. 3.1.3-3.1.4 and Fig. 3a). To our knowledge, this is the first time that such an observation has been made. The source of deposited eBC on the snowpack is uncertain but could be explained by one, or the combination, of the two following hypotheses. On the one hand, eBC could have been long-range transported from lower latitudes and subsequently dry or wet-deposited on the snowpack. On the other hand, it is also conceivable that eBC on the snowpack originated from *Polarstern*'s stack indirectly. Continuous in situ observations of eBC in snow, as well as measurements in collected snow, would be needed to further examine the hypotheses presented above. In any case, the re-emission of previously deposited eBC could represent an important and overlooked source of atmospheric eBC in the central Arctic, during a period when long-range transport is still limited by the extent of the polar dome. Due to its hydrophobic properties, eBC could influence the CCN activation potential of the sublimated blowing snow particles, depending on the particles' mixing state (Motos et al., 2019; Zieger et al., 2023). Additionally, eBC contributes to atmospheric warming and stratification through the absorption of incoming shortwave radiation (Flanner, 2013). The latter effect is irrelevant during the dark autumn months but could become important with the return of solar radiation in spring and at lower latitudes of the Arctic where sunlight is present longer during autumn. Overall, future studies and observational campaigns should focus on the characterization of this process, especially in the likely scenario where shipping becomes more important in the Arctic (Gilgen et al., 2018; Smith and Stephenson, 2013) and this potential eBC-cycling process becomes increasingly relevant.


*Long-range transport as a remote source of aerosols:*

Around November 11[th], a shift from anomalously low to high surface temperature, associated with 2 consecutive cyclones, triggered the storms presented above (Rinke et al., 2021). These synoptic-scale events were associated with air mass transport from lower latitudes, especially from northern Siberia (see Fig. S5). During the first storm, the gradual increase in $CO_2$ on



November 11th, peaking on the 13th (an increase of ~6 ppm), was evidence of the air mass change associated with the cyclonic conditions. Likewise, for the second storm, $CO_2$ started increasing on the 24th and peaked on the 25th (an increase of ~4 ppm). For both cyclones, the perturbed $CO_2$ signal was highly correlated with that of $SO_4^{2-}$ ($\rho_{pearson} = 0.69$, p-value < 0.001, and $\rho_{pearson} = 0.89$, p-value < 0.001, for the first and second storms, respectively), following a distinct temporal evolution from that of the wind speed-related variables discussed before. This decoupling is evident during the first storm, on the 13th, when $SO_4^{2-}$ and

$CO_2$ peaked when wind speed was continuously decreasing, and $N_{>1000nm}$ and NaCl signals were low. In contrast with the blowing snow period when $N_{>1000nm}$ and $N_{10-500nm}$ were highly correlated, the correlations dropped to $\rho_{pearson} = 0.42$ and $\rho_{pearson} = 0.40$ (p-values < 0.001) outside the blowing snow events for the first and second storms, respectively (see Fig. S3a, b), highlighting that the sources of these particles were different. The peak $SO_4^{2-}$ during the first (~0.32 μg/m³) and second (~0.44 μg/m³) storm were respectively ~2 and ~3 times larger than the monthly $SO_4^{2-}$ median concentration in November (see Sect.

3.1.2). Organics behave similarly to $SO_4^{2-}$, reaching about 0.26 μg/m³ during both storms, or ~2 times that of the November median Org concentration. The relative abundance of $SO_4^{2-}$, the increase in $CO_2$, and the related emission source region (i.e., Siberia) indicate that the pollution brought to the central Arctic under these cyclonic conditions was anthropogenic in origin.

We also observed a temporal co-variation between the mass concentrations of $SO_4^{2-}$ and Org and the number concentrations

of CCN, particularly when the influence of blowing snow was ruled out (i.e., outside the blowing snow flag). As such, the increase in CCN number concentrations on November 13th, reaching 87.2, 95.4, 101.9, 107.7, and 119.6 cm⁻³ at the respective SS levels 0.15, 0.2, 0.3, 0.5, and 1%, was likely related to the increase in $SO_4^{2-}$ and Org mass concentrations from long-range transport. The smaller spread in CCN number concentrations with increasing SS compared to that of the blowing snow-related increase, discussed before, indicates that the size distributions of the long-range transported material contained less Aitken

mode particles, as seen in Fig. S3a. For the second storm, the sharp increase in $CO_2$ mixing ratio (and $SO_4^{2-}$ and Org mass concentrations) on the 27th, was also associated with an increase in CCN number concentration at all SS levels, although with a larger spread. The larger CCN concentration with increasing SS (~95 cm⁻³ at 1% SS versus 45 cm⁻³ at 0.15%) could here be explained by the fact that organics became the dominant species and that, because of their lower hygroscopicity compared to $SO_4^{2-}$ (Siegel et al., 2022), required higher SS levels for droplet activation. Alternatively, a larger fraction of Aitken mode

particles could also explain this behavior. However, as seen in Fig. S3b, PNC in the size range 80-200 nm dominated over Aitken mode particles in the size range 10-80 nm.

Overall, under these cyclonic conditions, not only did wind-driven local aerosol production from blowing snow and SSA enhance CCN number concentrations, but long-range transported aerosols also played an important, yet smaller, role. Together,

these large increases in CCN number concentration can increase cloud emissivity and longevity, and could be in direct relation with the measured anomalously high downward longwave radiation in November during MOSAiC (Rinke et al., 2021).



### 3.2.2 High-resolution case study from spring 2020

We have seen in Sect. 3.2.1 how, under synoptic scale meteorological events, local and remote sources of aerosols can elevate aerosol mass and number concentrations from the low autumn background values, having climate-relevant implications for

the central Arctic cloud formation and radiative budget. The question logically arises whether these storm-induced high-concentration events were also observed in spring when the background particle concentration was much higher (haze), and if so, whether the implications are comparable to autumn. We present in Fig. 6 two spring storms in March 2020, where the same variables as in Fig. 5 are discussed.

*Wind-dependent aerosol generation as a local source of aerosols:*

On March 15$^{th}$, the wind speed increased rapidly reaching a maximum of 17 m/s. Blowing or drifting snow was detected from 7:00 on the 15$^{th}$ to 10:00 on the 17$^{th}$. Similarly to the storms presented in Sect. 3.2.1, the increasing wind speed was associated with increases in aerosol properties, where $N_{>1000nm}$ increased by a factor of ~3 (from 7.0 cm$^{-3}$ before the storm, from 13:00 on the 14$^{th}$ to 6:00 on the 15$^{th}$, to 21.6 cm$^{-3}$ at the highest, from 7:00 to 9:00 on the 15$^{th}$) and the NaCl signal increased by a factor

of ~4 (from 0.03 to 0.12 a.u.). Here as well, the NaCl and $N_{>1000nm}$ signals were highly correlated during the blowing snow period ($\rho_{pearson} = 0.83$, p-value < 0.001). Simultaneously, CCN number concentrations were increased by a factor of ~2 at all SS levels compared to background (from 121.7 to 229.9 cm$^{-3}$ at 0.15% SS, from 177.6 to 313.1 cm$^{-3}$ at 0.3% SS, and from 254.4 to 417.3 cm$^{-3}$ at 1% SS). It however appears that the CCN number concentrations were already quite high before the storm, especially at 1% SS, as the result of an important contribution of Aitken mode particles at that time (see Fig. S4a). The

correlation between CCN number concentrations and the NaCl signal was also lower at 1% SS ($\rho_{pearson} = 0.66$, p-value < 0.001) than at 0.15% SS ($\rho_{pearson} = 0.82$, p-value < 0.001). This indicates that the Aitken mode particle population (relevant for SS = 1%) is different from the one in fall, where CCN and NaCl correlated with $\rho_{pearson} = 0.88$. Given that an Aitken mode was already present prior to the storm, there were likely at least two particle populations within this size range. The scattering coefficient increased by a factor of ~3 (from 16.6 to 46.2 Mm$^{-1}$) during the storm and correlated greatly with the NaCl signal

($\rho_{pearson} = 0.85$, p-value < 0.001). Finally, eBC mass concentrations seemed to follow the NaCl signal, with a storm-peak increase by a factor of ~2 (from 0.10 to 0.21 µg/m$^3$). However, the low eBC data availability at that time was insufficient to draw robust conclusions on the source of eBC related to the increase, whether it was locally emitted through blowing snow or long-range transported. While the influence of local pollution from *Polarstern* reduced the amount of available data for the second storm from March 26$^{th}$ to March 29$^{th}$, especially for eBC mass and CCN number concentrations, the same conclusion

as for the first storm could be drawn regarding the relations between wind speed, NaCl signal, $N_{>1000nm}$, and the scattering coefficient. In particular, the peak in wind speed on the 27$^{th}$ (from around 7:00 to 11:00 pm), under blowing snow conditions, was associated with increases in $N_{>1000nm}$ from 4.8 to 16.5 cm$^{-3}$ (increase by a factor of ~3), in NaCl from 0.04 to 0.13 a.u. (~3), and in the scattering coefficient from 10.4 to 25.5 Mm$^{-1}$ (~3), compared to background conditions on the 26$^{th}$. In





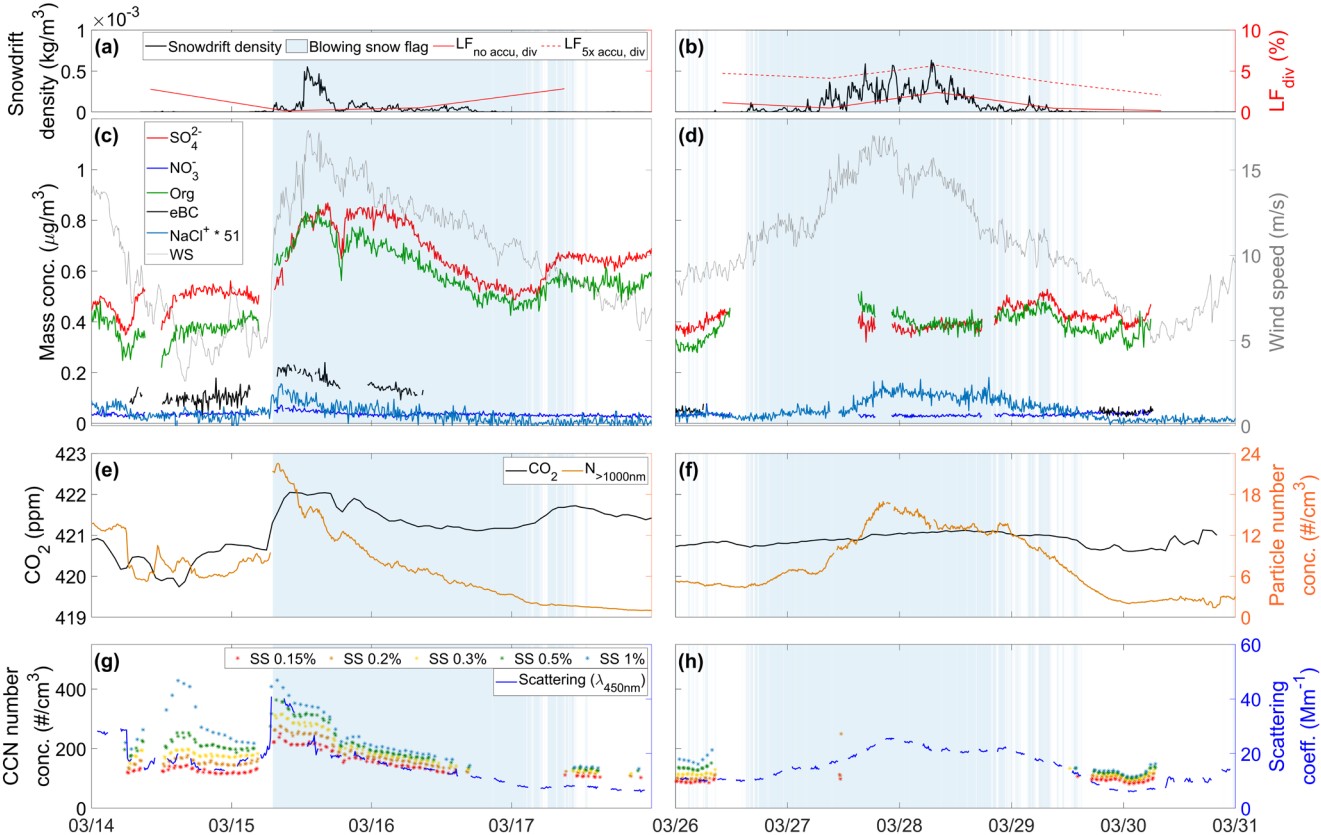

**Figure 6: High-time resolution case studies of two storms in March 2020 during MOSAiC.** Refer to Fig. 5 for a description of the variables included in all panels. The snowdrift density was derived from the snow particle counter mounted at 8 cm above ground. All measurements were averaged (A.M.) to 10 min time resolution, expect for $CO_2$ which is hourly and $LF_{div}$ which depends on satellite data availability. $LF_{5x\ accu,\ div}$ data were not available for the first storm (panel (a)). Data identified as affected by local contamination (pollution) were removed, except for NaCl which did not exhibit any reaction to local pollution during these periods. NaCl signals are in arbitrary units.

comparison to the November storms, the relative wind-dependent increases in the variables in March were however less

important.

Finally, as discussed in Sect. 3.2.1, wind-generated SSA from open leads and the ice-free ocean may have also contributed to the observed increase in the variables mentioned above. The co-occurrence in the observed drifting snow, NaCl signal, and $N_{>1000nm}$, coupled with the relatively small lead fraction (< 1 % and < 2.5 % for the first and second storm, respectively),

however, suggest that the influence of blowing snow may have been dominant over SSA from leads during the blowing snow periods. The exact strengths of these different SSA sources remain an open research question and cannot be fully answered here.

*Long-range transport as a remote source of aerosols:*



The increase in wind speed on March 15[th] was associated with a rising $CO_2$ mixing ratio (~ 2 ppm increase), covarying with $SO_4^{2-}$ mass concentrations ($\rho_{pearson}$ = 0.66 and p-value < 0.001 during the blowing snow period). Organics followed a similar temporal evolution. $SO_4^{2-}$ and Org mass concentrations peaked with a six to seven hours delay compared to the peak in NaCl signal, indicating that the source of these particles was different. This is also observed in the different behavior between the PNC in the size range 10-80 nm and 80-200 nm (see Fig. S4a). The observed increases in $SO_4^{2-}$ (from 0.51 to 0.85 $\mu g/m^3$) and

Org (from 0.34 to 0.78 $\mu g/m^3$) mass concentrations by a factor of ~2 were associated with long-range transport of pollution from eastern Siberia (see Fig. S6a) and a strong cyclone activity (Rinke et al., 2021). The impact that these long-range transported particles had on the CCN population here is difficult to quantify given the apparent co-occurrence of locally and remotely-emitted particles during the storm. Despite the low data availability during the March 26[th]-29[th] storm, the influence of long-range transported particles seemed to be limited, with little to no variations in $CO_2$ mixing ratio as well as $SO_4^{2-}$ and

Org mass concentrations. While back-trajectories at that time pointed towards Siberia being a potential source region (see Fig. S6b), it appeared that the transported air masses were relatively clean compared to e.g., the March 15[th] storm. The seasonal differences in aerosol populations (related with size distributions and number and mass concentrations) associated with local and remote aerosol sources are presented and discussed in Sect. 3.3.

### 3.3 Aerosol size distributions during transition seasons

**3.3.1 Contribution of local and remote sources to the submicron particle number size distributions**

Based on the above case studies, we found that local wind-dependent aerosol generation and long-range transport were important sources of aerosols during the autumn and spring storms during MOSAiC. We extended this approach to the entire periods of October to November 2019 and March to April 2020 by clustering the PNSDs into distinct clusters (see Sect. S4.1 for a description). Figure 7a shows the resulting median size distributions in October-November associated with the clusters,

where each SMPS timestep was uniquely assigned to time periods classified by: blowing snow-related (*BLSN*), long-range transport-related (*LRT*), long-range transport-related with larger but fewer particles (*LRT aged*), or low-concentration background conditions (*BG*). For March-April, the PNSDs were clustered into *BLSN*, haze-related (*Haze*), or bimodal haze-related (*Haze bimodal*), as shown in Fig. 7b. For a direct comparison of the shape of the clustered PNSDs, the normalized size distributions clusters are provided in Fig. S11. The fitting modal parameters (location and amplitude of the modes) are given

in Table S2. Median values (and 25[th] - 75[th] quantiles) of $N_{>1000nm}$, $N_{10-500nm}$, $SO_4^{2-}$, eBC, NaCl, and CCN concentrations associated with each cluster are given in Table 1. Importantly, the k-mean clustering algorithm is a statistical dimensionality reduction method and cannot be used to separate the contribution of various aerosol sources to each single PNSD measured. In other words, the names given to the clusters do not indicate that the PNSDs included in each cluster are the result of a single contributing emission source, as multiple sources contribute to every aerosol size distribution. Names hence indicate the likely

dominating process and source.

 

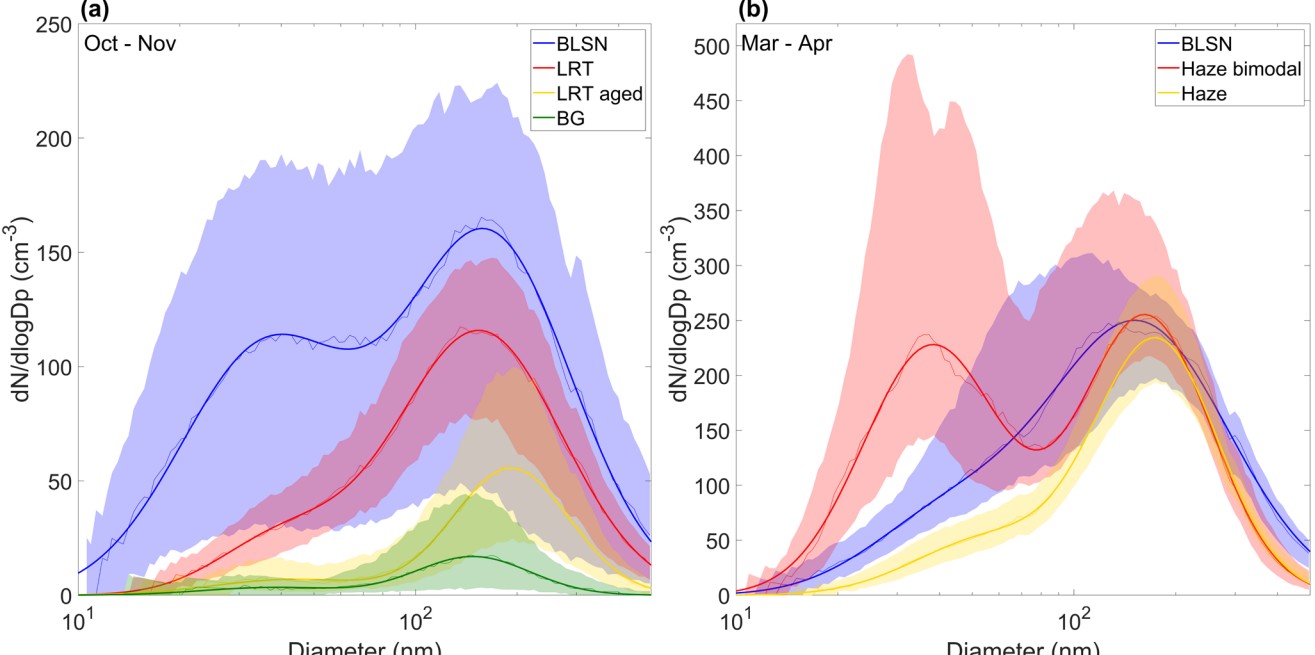

**Figure 7: Clustered PNSD in (a) October-November and (b) March-April.** PNSDs were uniquely attributed to each cluster: blowing snow-related (BLSN), long-range transport-related (LRT), long-range transport-related with larger but fewer particles (LRT aged), low-concentration background conditions (BG), haze-related, or bimodal haze-related (Haze bimodal**)**. Weak solid lines correspond to the medians of the clusters' size distribution while the thick solid lines show the bimodal log-normal distributions fitted to the medians. The lower (upper) boundary of the transparent envelops correspond to the 25th (75th) percentile of the clustered size distributions.

In October-November, the *BG* size distribution was characterized by very low $SO_4^{2-}$ (median = 0.078 µg/m$^3$) and eBC (median = 3 ng/m$^3$) mass concentrations, and low number concentrations of sub- and supermicron particles (median $N_{10-500nm}$ = 15.04 cm$^{-3}$, median $N_{>1000nm}$ = 0.12 cm$^{-3}$) with a weak (fitted) Aitken mode at 38 ± 2 nm and a dominant accumulation mode at 148 ± 1 nm. Overall, a quarter (25.2%) of all available SMPS PNSDs during this period were in the *BG* cluster, mostly in October (see Fig. S8). In contrast, the *BLSN* cluster was associated with high concentrations of submicron particles (median $N_{10-500nm}$ = 179.20 cm$^{-3}$), with strong contributions of Aitken mode particles (mode at 36 ± 1 nm) and, predominantly, accumulation mode particles (mode at 165 ± 2 nm). Spikes in $N_{10-500nm}$, contributing to the high 75th quantile value (255.6 cm$^{-3}$), were observed when blowing/drifting snow was detected (see Fig. S8b), making blowing snow a potential important contribution to the *BLSN* PNSDs, in line with findings from Gong et al. (2023). The accumulation mode particles could also be related to the long-range transported particles, which co-occurred with wind-generated aerosols during the storms (see Sect. 3.2). The Aitken shoulder is also consistent with recent work from Xu et al. (2022) showing Aitken mode sea spray aerosol, and also consistent with ultrafine aerosols observed during spring in Utqiaġvik, Alaska, related with lead-based sea spray aerosol (Myers et al., 2021). This highlights that blowing snow as a source of aerosol is likely not the only process that contributes to the shape of the *BLSN* cluster size distributions and that other locally wind-sourced aerosols (e.g., sea spray from open leads) should be considered. As such, it is important to mention that the *BLSN* cluster refers to periods when blowing snow was



**Table 1: Median (25th quantile, 75th quantile) values of $N_{>1000nm}$, $N_{10-500nm}$, $SO_4^{2-}$, eBC, NaCl, and CCN concentrations associated with the PNSD clusters in Oct-Nov (autumn) and Mar-Apr (spring).** Note that there are no exclusion criteria on the number of datapoints available in each cluster to compute the statistics and that the number of datapoints in each cluster is different for all the variables. An asterisk (*) next to the median value indicate that the distribution of the variable associated with the cluster is statistically different, at the 5% significance level using the Wilcoxon rank sum test, from the one associated with the *BG* cluster (in autumn) and the *Haze* cluster (in spring).

| | Occurrence | $N_{>1000nm}$ (cm$^{-3}$) | $N_{10-500nm}$ (cm$^{-3}$) | $SO_4^{2-}$ (µg/m$^3$) | eBC (ng/m$^3$) | NaCl (a.u.) | CCN SS 0.15% (cm$^{-3}$) | CCN SS 0.3% (cm$^{-3}$) | CCN SS 1% (cm$^{-3}$) |
|---|---|---|---|---|---|---|---|---|---|
| **BLSN (autumn)** | 20.7% | 3.13* (0.64, 6.23) | 179.20* (43.63, 255.60) | 0.185* (0.088, 0.281) | 41* (8, 65) | 0.109* (0.018, 0.196) | 68.5* (15.2, 105.4) | 102.6* (22.5, 137.1) | 138.8* (22.6, 177.0) |
| **LRT (autumn)** | 29% | 1.16* (0.72, 1.82) | 83.97* (58.66, 110.42) | 0.141* (0.094, 0.201) | 17* (6, 29) | 0.045* (0.015, 0.078) | 37.6* (24.4, 48.0) | 45.9* (27.4, 63.7) | 56.9* (41.4, 73.1) |
| **LRT aged (autumn)** | 25.1% | 0.25* (0.13, 0.55) | 29.61* (13.73, 48.73) | 0.150* (0.093, 0.237) | 12* (2, 24) | 0.004 (-0.002, 0.012) | 12.6* (5.5, 24.9) | 14.5* (6.0, 26.1) | 21.6* (10.0, 36.5) |
| **BG (autumn)** | 25.2% | 0.12 (0.05, 0.21) | 15.04 (3.01, 26.60) | 0.078 (0.021, 0.119) | 3 (-5, 9) | 0.003 (-0.004, 0.012) | 3.1 (0.8, 11.2) | 6.7 (2.7, 17.2) | 8.3 (3.9, 17.6) |
| **BLSN (spring)** | 18% | 9.33* (4.99, 13.59) | 198.68* (157.73, 269.17) | 0.442 (0.393, 0.506) | 81* (55, 116) | 0.078* (0.041, 0.114) | 118.7* (65.7, 127.3) | 151.8* (82.4, 176.7) | 170.0* (121.2, 221.6) |
| **Haze bimodal (spring)** | 13.9% | 3.26 (1.81, 5.11) | 234.25* (182.30, 446.89) | 0.509* (0.425, 0.702) | 74* (44, 97) | 0.034* (0.013, 0.052) | 130.6* (90.3, 244.3) | 183.4* (115.4, 326.4) | 322.2* (163.4, 577.0) |
| **Haze (spring)** | 68.1% | 2.91 (1.65, 4.62) | 138.73 (111.68, 165.70) | 0.439 (0.386, 0.611) | 46 (33, 63) | 0.013 (0.004, 0.031) | 85.3 (73.0, 115.1) | 97.9 (85.3, 127.2) | 121.6 (98.3, 152.0) |

observed, but it is not exclusively associated with blowing snow particles. The *BLSN* cluster, representing 20.7% of all available PNSD observations in October-November, was also associated with higher NaCl mass concentrations (a factor ~37), $N_{>1000nm}$ (a factor of ~26), and eBC (a factor of ~14), compared to the *BG* cluster. Furthermore, CCN number concentrations were greatly enhanced within the *BLSN* cluster periods, with medians of 68.5 cm$^{-3}$ at 0.15%, 102.6 cm$^{-3}$ at 0.3% SS, and 138.8 cm$^{-3}$ at 1% SS, ~23, ~15, and ~17 times larger than *BG* median CCN concentrations at these SS levels, respectively. The *LRT* cluster's median size distribution is more monomodal, dominated by an accumulation mode at 155 ± 1 nm, and only a weak Aitken mode at 41 ± 1 nm, associated with a median $N_{10-500nm}$ value of 83.97 cm$^{-3}$ (~6 times higher than the *BG* value). CCN number concentrations and $SO_4^{2-}$ mass concentrations were also enhanced for *LRT* compared to *BG* (Table 1). The second *LRT*





cluster (i.e., *LRT aged*) was characterized by a lower median $N_{10-500nm}$ value of 29.61 cm$^{-3}$ and an accumulation mode at $192 \pm 1$ nm. This could be indicative of longer atmospheric residence times of the particles, yielding lower concentrations through dilution and larger particles through aging and coagulation processes. Similar PNSD clusters were found at VRS, namely "Haze" and "Aged", with strong contributions in November and with main accumulation mode diameters similar to the ones found for our two *LRT* clusters (Lange et al., 2018, 2019; Pernov et al., 2022). Overall, *LRT* and *LRT aged* contained

respectively 29 and 25.1% of all available PNSD measurements in October-November. Since the *BG* median size distribution shape resembles closely that of the *LRT* one (i.e., dominant accumulation mode around 150 nm and weak Aitken mode), the background autumnal aerosol population could be interpreted as diluted long-range transported aerosols. To estimate the contribution of each "main source" (cluster) to the CCN population at 0.3% SS, which is assumed to be a representative SS level in autumn in the Arctic (Motos et al., 2023), we divided the summed CCN number concentrations associated with a given

cluster throughout the October-November period by the total (all clusters) summed CCN number concentrations for the same period. This approach yielded contributions of 80%, 17%, and 3% to the October-November CCN number concentrations (at 0.3% SS) for *BLSN*, *LRT + LRT aged*, and *BG*, respectively.

In March-April, under the high aerosol background concentration characteristic of Arctic haze (see Sect. 3.1.4), we refer to

background conditions with what we call the *Haze* cluster. The size distributions of the *Haze* cluster were strongly stable (i.e., low interquartile range), dominated by an accumulation mode at $176 \pm 1$ nm and a small Aitken shoulder at $57 \pm 1$ nm. Overall, the *Haze* cluster size distribution was characteristic of haze conditions in the Arctic (Boyer et al., 2023; Croft et al., 2016b; Tunved et al., 2013). The association of this cluster with background conditions mostly stems from the fact that this cluster comprised 68.1% of all PNSD observations in March-April. In contrast with the October-November period, the background

*Haze* concentrations in March-April were high (median $N_{10-500nm}$ = 138.73 cm$^{-3}$; median $SO_4^{2-}$ = 0.439 µg/m$^3$, similar to the overall March-April $SO_4^{2-}$ median in Sect. 3.1.4), as a result of more intense pollution long-range transport events and reduced sinks. These more intense long-range transport events are partly the ones that made up the *Haze bimodal* cluster, where higher number concentrations were reached (median $N_{10-500nm}$ = 234.25 cm$^{-3}$) along with high $SO_4^{2-}$ and eBC mass concentrations (median $SO_4^{2-}$ = 0.509 µg/m$^3$, median eBC = 74 ng/m$^3$). The median *Haze bimodal* size distribution had roughly equal

magnitudes of the Aitken mode (at $38 \pm 0.2$ nm) and accumulation mode (at $163 \pm 1$ nm), although with a large interquartile range. Aerosol cloud processing could explain the distinct bimodal distribution shape with a Hoppel minimum (indicative of aerosol cloud processing (Hoppel et al., 1986)) at about 80 nm, which is a similar value to what was found elsewhere in the Arctic (Boyer et al., 2023; Freud et al., 2017; Gramlich et al., 2023; Karlsson et al., 2022). Fast transport, differences in the contributing source regions, as well as contribution from recently newly formed particles are all possible explanations to the

higher contribution of Aitken mode particles to the *Haze bimodal* cluster PNSDs. Freud et al. (2017) demonstrated the importance of cloud processing as a source of accumulation mode particles in the Arctic, suggesting that, overall, our *Haze* cluster could have been associated with more cloud processing than the *Haze bimodal* cluster. *Haze bimodal* comprised 13.9% of the March-April PNSD observations but mainly occurred around the mid-April warm air mass intrusions (Fig. S10), where



the changes in aerosol physicochemical properties associated with these fast-transport events were discussed by Dada et al.
(2022). Similarly to October-November, the *BLSN* cluster in March-April was associated with high submicron PNC (median
$N_{10-500nm}$ = 198.68 cm$^{-3}$), contributing 18% of all available PNSD observations, primarily during times when blowing/drifting
snow was detected (see Fig. S10b). While the accumulation mode (at 156 ± 3 nm) contribution to the median *BLSN* PNSD
was higher in March-April compared to October-November (see Table S2), the Aitken mode (at 43 ± 2 nm) amplitude was
lower. This is also visible when comparing the shapes of the normalized autumn and spring *BLSN* PNSDs (see Fig. S11). Part
of these PNSD differences could be due to differences in surface snow salinity, size of the blowing snow particles, or snow
age, which are all parameters that have been shown through modelling to influence the size of the dry sea salt particles produced
from blowing snow (Yang et al., 2008, 2019). Regarding the relation to the measured CCN concentrations and following the
same approach as for the October-November period, we find that the *BLSN* cluster contributed to 20% of the March-April
CCN number concentrations at 0.3% SS, while *Haze* contributed 61% and *Haze bimodal* 19%. Although the fraction of
blowing snow-related CCN (minding other sources are likely contributing to the *BLSN* cluster) is smaller compared to autumn,
blowing snow episodes remain an important (local) source of CCN, whose importance could increase in the future as the
contribution of anthropogenic emissions to the haze burden decreases.

Using daily averaged PNSDs, Boyer et al. (2023) performed a similar PNSD clustering analysis for the entire MOSAiC year
(October 2019 – October 2020). The authors reported a bimodal cluster, with modal diameters of 46.1 nm and 135.8 nm and
the highest occurrence in November and April. While this cluster was not attributed to any particular emission process in their
study, our results suggest that the bimodal nature of the PNSDs in November and April could have originated from different
processes (i.e., locally-produced blowing snow and SSA particles in November and long-range transported cloud-processed
particles in April). This highlights the importance of concurrent high-time resolution observations of aerosol size distributions
and chemical composition to understand short-term aerosol variability in the central Arctic and the emission processes related
to it.

### 3.3.2 Size-resolved chemical composition measurements

We obtained size-resolved chemical information for sulfate and organics using the AMS, which can provide critical
information, such as inference of aerosol mixing state, to complement the PNSD analysis from Sect. 3.3.1. The monthly median
of the size distributions of sulfate (in autumn and spring) and organics (in spring only) are shown in Fig. 8, and Fig. S12 shows
the same distributions with interquartile range (25$^{th}$ and 75$^{th}$ quantiles). Each mass size distribution was normalized to its
maximum, to compare their shapes and mode diameters (in vacuum aerodynamic diameter). The locations of the fitted modes
for sulfate and organics' monthly mass size distributions are given in Table S3.

Sulfate mass in autumn was characterized by a monomodal distribution with mode diameters at 479 ± 3 and 500 ± 1 nm for
October and November, respectively. Our observations in Fig. 8a and Fig. S13 suggest that autumn was characterized by a



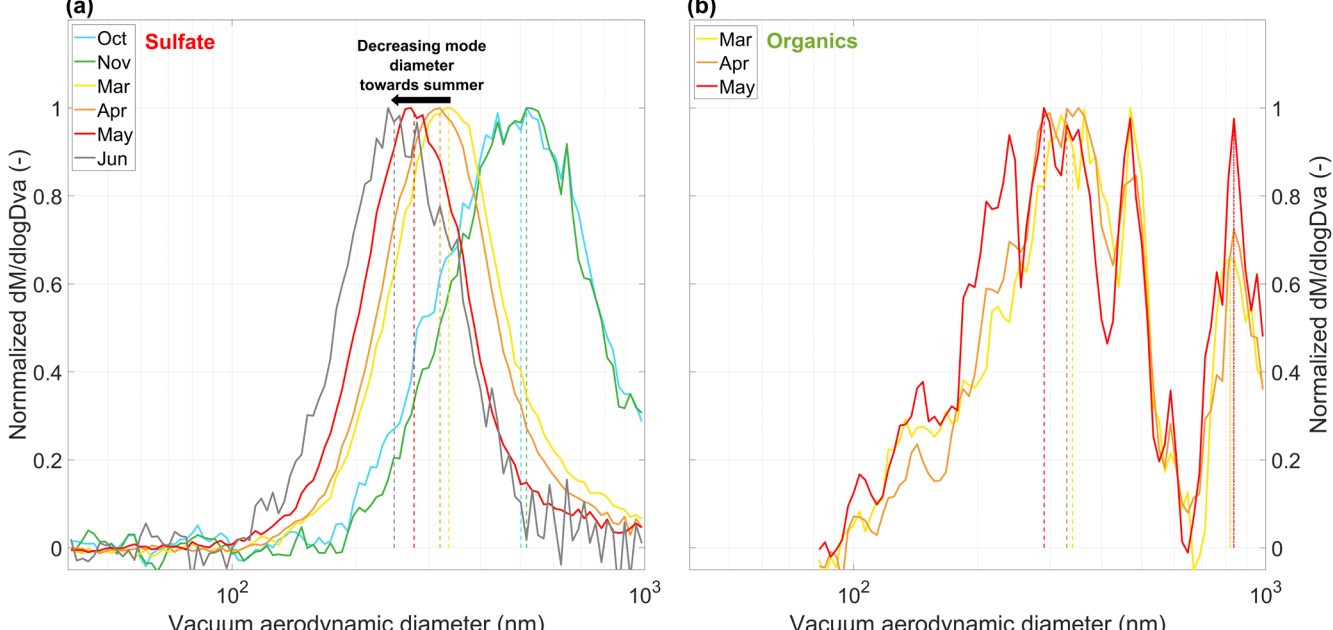

**Figure 8: Species-specific size distributions of sulfate (a) and organics (b) during MOSAiC, presented as monthly median values.** The thick colored lines represent the medians of the species-specific size distributions, which were normalized by the distributions' maximum value. Dotted lines in (a) represent the mode diameter, estimated from fitting a monomodal log-normal distribution through the observations. Dotted lines in (b) represent the mode diameters, manually estimated (i.e., by eye). All the months that are not shown in both panels had a signal-to-noise ratio too low for the PToF data to be analyzed, mainly for organics outside of the spring months. The monthly medians exclude polluted data for both sulfate and organics. Due to gas-phase interactions with the organics' PToF signal, the size distributions of organics in (b) were truncated below 80 nm due to interferences with gas-phase compounds.

smaller number of particles, for which most of the ($SO_4^{2-}$) mass was contained in the larger range of the size distribution (i.e., above ~ 400 nm in vacuum aerodynamic diameter, or above 267 nm in mobility diameter assuming an average particle density of 1.5 g/cm³ (Hegg et al., 1996) and spherically-shaped particles). Possibly the $SO_4^{2-}$ mass could be related to the *LRT aged* cluster in Fig. 7, which was associated with slightly higher (p-value = 0.03) $SO_4^{2-}$ median mass concentration (0.150 µg/m³) compared to the *LRT* cluster (0.141 µg/m³), as reported in Table 1, and larger particles overall. In spring, the monomodal distributions had a mode diameter at around 300 nm (seasonal average March-June = 295 nm), consistent with the characteristics of the *Haze* PNSD cluster in Sect. 3.3.1 which represented more than two thirds of the March-April observations. This mode diameter is in line with past studies that have used stage impactor aerosol collection and chemical characterization with ion chromatography in the Arctic (Hillamo et al., 2001; Leck and Persson, 1996; Mukherjee et al., 2021; Ricard et al., 2002). Overall, we observed a non-negligible decrease in $SO_4^{2-}$ mode diameter from October (479 ± 3 nm) to June (256 ± 1 nm). This decrease in mode diameter was also observed for the submicron particle volume size distribution from the SMPS (see Fig. S13b). Using ion chromatography measurements of non-sea-salt sulfate, Quinn et al. (2002) found smaller submicron sulfate mass scattering efficiency in summer (July-September) compared to spring (March-June) at Utqiaġvik, Alaska. The authors, however, argued that these differences were negligible and concluded that the sulfate size distribution was unchanged throughout the year. Our observations show that this is not the case, at least in the central Arctic, and that



further measurements of size-resolved chemistry in the central Arctic are needed to resolve the influence of sulfate particle size on the climate-relevant scattering efficiency property.

Organics in spring exhibited a multimodal size distribution. The main mode diameter was found at a similar diameter as for sulfate (i.e., around 300 nm, seasonal average March-May = 320 nm), which indicates that, at these sizes, the two species were probably internally mixed. Apart from some variability around the main mode, a second mode was observed around 800-850 nm (estimated at 820 nm in March and at 840 nm in April and May). The second mode was likely missed in the PNSD analysis presented in Sect. 3.3.1 due to the size limitation of the SMPS (10-500 nm). By integrating the monthly median mass size

distributions of organics for March-May, we found reasonable agreements with the monthly medians obtained from the mass spectral quantification (see Fig. 3). That is, we obtained median Org mass concentrations from the integrated size distributions (mass spectra) of 0.425 (0.334), 0.466 (0.357), and 0.359 (0.283) µg/m$^3$ for March, April, and May, respectively. Integrating the second organic mode only (i.e., from 650 nm onward, which corresponds to the approximated location of the minimum between the first and second mode), we found that about 15, 17, and 19% of the submicron organic mass is in this second

mode for March, April, and May, respectively. We hypothesize that this mode corresponded to organic-coated sea salt particles, where only the organics are detected by the AMS and the sea salt core is left (mostly) undetected due to its refractory nature. Sea spray particles with organic coatings have been observed in previous Arctic studies (Hawkins and Russell, 2010; Kirpes et al., 2019), as well as lab-generated nascent SSA (Ault et al., 2013; Kaluarachchi et al., 2022; Mirrielees et al., 2022). The organic coating is obtained during bubble bursting at the sea surface microlayer (Blanchard, 1975). The classes of organic

compounds identified in individual sea spray particles collected in the Arctic include saccharides, fatty acids, and amino acids (Hawkins and Russell, 2010; Kirpes et al., 2019). Such an organic coating can have an impact on the particle CCN activation potential that is two-fold: (1) through the presence of non-soluble surfactants on the outer shell of the particle, surface tension can decrease, increasing the particle's activation potential (Giddings and Baker, 1977; Ovadnevaite et al., 2017); (2) the lower hygroscopicity of organics can, on the other hand, decrease the particles activation potential (Ovadnevaite et al., 2011b). Both

effects would play a more important role in particular for smaller particle sizes. Both effects could also offset each other leading to small changes in CCN activity, as it was shown in controlled laboratory experiments (Moore et al., 2011). It remains to be elucidated what could be the impact of such organic coatings on CCN activation for these large particles in our study.

## 4 Summary and conclusion

In this study, we report the first year-long observations of size-resolved submicron aerosol chemical composition in the central

Arctic, based on high-time resolution measurements from a HR-ToF-AMS. Overall, the yearly cycle of the main non-refractory species mass concentrations (i.e., sulfate, organic, nitrate, ammonium, and chloride) exhibited variations that were typical of the Arctic's aerosol seasonal regimes (Moschos et al., 2022a; Schmale et al., 2022). In June – September, under some of the lowest yearly submicron mass concentrations, the aerosol population was largely dominated by organics in terms of mass (~



63% of $PM_1$). In autumn and spring, under Arctic haze conditions, anthropogenic emissions from lower latitudes constituted

the main source of $PM_1$, with a dominant $SO_4^{2-}$ fraction (47% and 50% for October-November and March-May, respectively) and an important contribution of eBC (6-7%). Due to instrumental failures, statistically representative datasets on aerosol chemistry are unfortunately missing for winter.

Comparing the year-round central Arctic $PM_1$ chemical composition to observations from a set of pan-Arctic land-based

stations (Moschos et al., 2022a), we found comparable results in terms of seasonality and, under certain conditions, absolute mass. Mostly, summer observations over the pack ice in the central Arctic showed lower mass concentrations compared to the coastal land-based sites, likely related to the remoteness of the region away from most open-ocean marine and terrestrial aerosol sources. Ammonium appeared to be far less abundant in the central Arctic throughout the year than at lower latitudes, with potential implications in terms of aerosols' acidity. The relative agreement between central- and pan-Arctic yearly

chemical composition observations suggests that, under current conditions (i.e., the extent of the winter and summer sea ice cover and atmospheric transport pathways), aerosol measurements from land-based monitoring sites can be generally extrapolated to the central Arctic. Whether this statement also applies to the speciation of organic aerosol, will be investigated in a follow up study.

Our real-time observations also allowed for high time-resolution process studies. In autumn when concentrations are generally low, we observed spikes in aerosol mass concentrations, with significant deviation from the background conditions. Such events were observed during the springtime haze period as well, despite the higher background concentration. We attributed these events to cyclonic (storm) activity over, or adjacent to, the central Arctic Ocean. The sensitivity of the central Arctic to the impact of cyclones was found to be two-fold. First, increasing wind speed was related to elevated number and mass of sub-

and supermicron aerosols upon sublimation of blowing snow and/or lead-based sea spray emissions, with sea salt levels up to 80 times larger than in low-wind conditions. Black carbon was found to correlate with sea salt during the blowing snow events ($\rho_{pearson}$ = 0.59-0.74, p-value < 0.001), indicating that these two species likely shared a common source process or controlling factor. Second, the cyclonic conditions were found to be associated with long-range transport of aerosols from Siberia, introducing high levels of, presumably anthropogenic, sulfate and organic aerosols. Overall, both local (wind-generated) and

long-range transported aerosol sources under stormy conditions contributed to enhanced CCN number concentrations in autumn and spring.

The same conclusions were also reached when statistically analyzing seasonal data as opposed to considering case studies. PNSDs were clustered into source-related observations of aerosol physicochemical properties (i.e., number concentration, size

distribution, and chemical composition). Blowing snow, more generally locally wind-generated particles, represented an important source of Aitken and accumulation mode particles in both autumn and spring, associated with high sea salt levels, total submicron PNC, and CCN number concentrations within the *BLSN* cluster. Approximately 20% of all autumn and spring



observations of PNSDs were associated with blowing snow events, as also found by Gong et al. (2023). Long-range transported aerosols were shown to contribute either in the form of diluted and aged accumulation mode (*LRT aged* cluster in autumn and

*Haze* clusters in spring) or as more intense pollution spikes associated with higher $SO_4^{2-}$, eBC, and CCN concentrations (*LRT* cluster in autumn and *Haze bimodal* cluster in spring). Importantly, in autumn, when aerosol number concentrations were low in general, we found that the *BLSN* cluster was associated with about 80% of the total seasonal CCN population at 0.3% SS. In spring, when anthropogenic haze dominated, the *BLSN* cluster was associated with about 20% of the CCN, at the same SS level. While it was suggested that particles locally produced from the sublimation of salty blowing snow particles made an

important contribution to the *BLSN* cluster, we could not fully isolate their contribution with regards to other potential sources (e.g., emissions from nearby open leads or long-range transported aerosols).

Based on size-resolved chemistry measurements, we also showed that organic and sulfate accumulation mode aerosols were internally mixed in autumn and spring. A second size mode > 800 nm $D_{va}$ was observed for the organics in spring, which

represented about 15-19% of the total submicron organic mass. We hypothesize that this second mode was related to organic coating on sea spray particles, obtained during bubble bursting at the sea surface microlayer. The low concentrations during seasons other than spring meant that mass spectrometric data were below the detection limit and did not allow for a more detailed analysis of the particle size of organics. Particulate sulfate however was abundant enough in autumn and spring, and we observe a reduction in diameter from ~480 nm $D_{va}$ in October to ~260 nm $D_{va}$ in June, with potential implications for the

scattering efficiency of these particles.

Our observations demonstrate that understanding aerosol concentrations and their contribution to CCN in the central Arctic requires information on short-timescale processes such as wind-generated particles, e.g., from blowing snow and sea spray, since observations particularly between October and May could not be described by Arctic haze contributions alone. Recent

work by the Arctic Monitoring and Assessment Programme (AMAP, 2021; von Salzen et al., 2022) has shown that reduction in anthropogenic haze, especially sulfate, will lead to significant Arctic warming (+ 0.8°C, range 0.4-1.4°C, from 1995-2014 average to 2050 following SSP1-2.6) due to reduced scattering by long-range transported aerosols. However, the model simulations did not consider local natural Arctic aerosol sources such as from blowing snow or lead-based SSA and aerosol-cloud interactions, which are particularly important for the surface temperature through longwave forcing in the absence of

solar radiation. Our results suggest that wind-generated (including blowing snow-generated) particles could produce CCN number concentrations of comparable or higher magnitude compared to haze particles, particularly in autumn. It is hence essential to conduct further simulations that take these new observations and aerosol-cloud interactions into account, specifically in scenarios with significantly declining anthropogenic haze, to better constrain the aerosol effect on Arctic surface temperature.




**Data availability**

*Datasets collected during MOCCHA:*

Chemical composition from the HR-ToF-AMS: https://bolin.su.se/data/oden-ao-2018-aerosol-ams-1 (Dada et al., 2022b)

Equivalent black carbon mass concentration from the AE33: https://bolin.su.se/data/oden-ao-2018-aerosol-ebc-ae33-1 (Heutte et al., 2024)

Aerosol size distribution (18-660 nm) from the SMPS: https://bolin.su.se/data/oden-ao-2018-aerosol-smps-1 (Baccarini and Schmale, 2020)

*Datasets collected during MOSAiC:*

Chemical composition from the HR-ToF-AMS: https://doi.pangaea.de/10.1594/PANGAEA.961009 (Heutte et al., 2023a)

Equivalent black carbon mass concentration from the AE33: https://doi.pangaea.de/10.1594/PANGAEA.952251 (Heutte et al., 2022)

Aerosol size distribution (10-500 nm) from the SMPS: https://doi.org/10.5439/1476898 (Kuang et al., 2022)

Aerosol number concentration from the APS: https://doi.pangaea.de/10.1594/PANGAEA.960923 (Bergner et al., 2023a)

Cloud condensation nuclei: https://doi.pangaea.de/10.1594/PANGAEA.961131 (Bergner et al., 2023b)

Merged carbon dioxide dry air mole fractions: https://doi.pangaea.de/10.1594/PANGAEA.944272 (Angot et al., 2022a)

Submicron aerosol total light scattering coefficient: https://doi.org/10.5439/1228051 (Koontz et al., 2022)

Atmospheric snow particle flux from the SPCs: https://doi.org/10.5285/7d8e401b-2c75-4ee4-a753-c24b7e91e6e9 (Frey et al., 2023)

Sea ice lead fractions: https://doi.pangaea.de/10.1594/PANGAEA.963671 (von Albedyll et al., 2023)

Continuous meteorological surface measurements: https://doi.pangaea.de/10.1594/PANGAEA.935221; https://doi.pangaea.de/10.1594/PANGAEA.935222; https://doi.pangaea.de/10.1594/PANGAEA.935223; https://doi.pangaea.de/10.1594/PANGAEA.935224; https://doi.pangaea.de/10.1594/PANGAEA.935225 (Schmithüsen, 2021a, b, c, d, e)

**Supplement**

The supplement related to this article is available online at *(insert link here)*.

**Author contribution**

BH wrote the manuscript with input from JS, and all the authors provided comments/revisions. JS, AB, and PZ conducted the MOCCHA field measurements reported in this work. LLJQ, IB, TL, JS, and TJ conducted the MOSAiC field measurements reported in this work. VM and IEH provided the pan-Arctic chemical composition measurements. SH lent and calibrated the



CCNC. BH, LD, NB, HA, MMF and IB provided and interpreted datasets, with help from JBP, JMC, KAP, MB, IEH, KRD, and JS.

JS and TP obtained the funding for the *Swiss* container during MOSAiC, and JS conceived the field measurements.

**Competing interests**

T.P. and P.Z. serve on the Editorial Board of Atmospheric Chemistry and Physics. The other authors declare no competing interests.

**Acknowledgments**

Data reported in this manuscript was produced as part of the international Multidisciplinary drifting Observatory for the Study of Arctic Climate (MOSAiC) expedition with the tag MOSAiC20192020, with activities supported by *Polarstern* expedition
AWI_PS122_00. The authors would like to thank the teams at the Paul Scherrer Institute and INAR for their land-based support during the MOSAiC expedition. We also thank all those who contributed to MOSAiC and made this endeavor possible (Nixdorf et al., 2021). For the expedition AO2018, the Swedish Polar Research Secretariat (SPRS) provided access to the I/B *Oden* and logistical support in collaboration with the U.S. National Science Foundation. We are grateful to the Chief Scientists Caroline Leck and Patricia Matrai, for planning, technical support, and coordination of AO2018, the SPRS logistical staff and
I/B *Oden*'s Captain Mattias Peterson and his crew for expert field support. Funding for the expedition AO2018 was provided to Caroline Leck by the NSF and SPRS's Cost sharing Understanding Arctic Ocean 2018 Reg. no. 2018-113, and by the Swedish Research Council projects nos. 824-2013-222 and 2016-03518. This research was funded by the Swiss National Science Foundation (grant nos. 200021_188478, 200021_169090, PZPGP2_201992, and PZ00P2_216181), the Swiss Polar Institute (grant no. DIRCR-2018-004), the BNP Paribas Swiss Foundation (Polar Access Fund 2018), the Knut-and-Alice-
Wallenberg Foundation within the ACAS project (Arctic Climate Across Scales, project no. 2016.0024), the Bolin Centre for Climate Research (RA2), the Swedish Research Council (project nos. 2018-05045 and 2016-05100) and by the Natural Environment Research Council (grant no. NE/R009686/1). J.S. holds the Ingvar Kamprad chair for extreme environments research, sponsored by Ferring Pharmaceuticals. J.B.P. received support from the Swiss Data Science Center project C20-01 Arctic climate change: exploring the Natural Aerosol baseline for improved model Predictions (ArcticNAP). M.M.F. was
supported by the UK Natural Research Council (NERC) (NE/S00257X/1) and the NERC National Capability International grant SURface FluxEs in AnTarctica (SURFEIT) (NE/X009319/1). This project has received funding from the European Union's Horizon 2020 research and innovation program under grant agreement no. 101003826 via project CRiceS (Climate Relevant interactions and feedbacks: the key role of sea ice and Snow in the polar and global climate system), no. 714621 via project GASPARCON (Molecular steps of gas-to-particle conversion: From oxidation to precursors, clusters and secondary
aerosol particles), and from the Academy of Finland (grants no. 337552, 333397, and 337549). A subset of data was obtained from the Atmospheric Radiation Measurement (ARM) User Facility, a U.S. Department of Energy (DOE) Office of Science



User Facility managed by the Biological and Environmental Research Program. We acknowledge funding from the US DOE grants nos. DE-SC0022046 and DE-SC0019251, and NOAA Cooperative agreement NA22OAR4320151. We thank Silvia Bucci and the FLEXPART group at the University of Vienna for calculating the air mass back trajectories with the FLEXPART model.

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
