# Peer review of "Observations of high time-resolution and size-resolved aerosol chemical composition and microphysics in the central Arctic: implications for climate-relevant particle properties"

_EGUsphere, 2024_

## Author Response (AR1)

**Authors' response to the reviewers**

Reviewer's comments in black, Authors' response in blue
Line numbering in the answers is related to the version of the revised manuscript where the "track changes" mode is not shown.

**Anonymous referee #1:**

Review to "Observations of high time-resolution and size-resolved aerosol chemical composition and microphysics in the central Arctic: implications for climate-relevant particle properties" by Heutte et al.

The manuscript by Heutte et al. presents an annual cycle of aerosol data (number, size, composition) from the central Arctic. Data from two field projects, from August to September 2018 and from October 2019 to September 2020 are combined to obtain an annual cycle. The chemical composition data were obtained using a HR-ToF-AMS, for which a data gaps from January until February 2020 exists. Black carbon was measured using an aethalometer and a MAAP. Size distributions were measured using an SMPS system. Local contamination (exhaust from the research vessel) was identified and removed from the analysis.

Due to the logistical difficulty to obtain such data (such data can only be measured on icebreakers moored on ice floes or frozen in pack ice), such data from the central Arctic a scarce and are therefore very valuable.

The manuscript first presents the chemical composition of the aerosol during the annual cycle. Relative composition for four different time periods as well as the monthly seasonality is shown. Several case study of storm events are discussed in more detail. The influence of long-range transported particles and of blown snow and sea spray on CCN is discussed. The variation of the size distributions between different seasons is analysed by means of clustering of the SMPS size distributions. Size-resolved chemical composition (AMS) is also presented.

Overall, the manuscript is very descriptive. The measured data are presented in great detail, but often in a repetitive way, as for example in the case of the storm events. The main new finding is that wind-generated and blowing snow-generated particles contribute significantly to the CCN number, comparable to haze particles.

However, as said above, the data are very valuable such that I recommend publication.
The authors should consider to shorten the individual case descriptions in 3.2.

We thank the reviewer for the positive and constructive comments. We agree that the original manuscript was quite lengthy. Parts of the redundant information were removed, as described below in the answers to reviewer #1-3. Regarding the case study in Sect. 3.2, we decided to move the analysis of the spring storms (formerly Sect. 3.2.2) in the Supporting Information (now Sect. S3.1) while providing the main conclusions from this analysis in Sect. 3.2.1 and Sect. 3.2.2.

**General comments:**

The fact that the comparison between AMS and SMPS (MOSAiC only) yielded markedly different results is problematic. However, I think the approach of Heutte et al. (2023b) to use scaling factors to match AMS and SMPS volume are justified, but induces an additional uncertainty to the data.

We understand the reviewer's concerns about the AMS scaling procedure. We also thank the reviewer for recognizing that the method employed is justified. Regarding the additional uncertainty that such a scaling procedure induces, we added a statement in the sentence at lines 229-231: "These scaling factors were derived and applied independently for the measurements periods in-between the non-operational periods mentioned above (Heutte et al., 2023b). The scaled concentrations are expected to be upper estimates that add some uncertainty to the data.".

**Specific comments:**

Lines 110 ff: Do Amines (e.g., TMA) play a role as well?
TMA has been observed previously in the Arctic boundary layer (e.g., Köllner et al., 2017). As the sentence at line 110 ff (now 106 ff) was rather a general statement, we decided not to list every class of organic compounds that contribute to the organic aerosol population. Instead, we only mentioned MSA, for which our AMS was calibrated and could be easily identified in the mass spectrum.

Line 193: floe instead of flow?
Thank you for spotting the mistake. The change was made to "floe".

Line 250: PSL were not used for size calibration?
The size calibrations were performed during the campaign with ammonium nitrate and ammonium sulfate, and combined with the ionization efficiency calibrations using the same measurements. Hence, we did not use PSL particles for size calibration of the AMS during the campaign.

Line 259-260: I don't think that estimating the mode of a size distribution "by eye" is a justified method. Looking at the data in Fig 8 b) and Fig S12 b), I would think that a lognormal fit should be able to find the maximum when the starting conditions are set properly.
The reviewer is right that the mode fitting "by eye" is not a justified method. However, following the comment from the reviewer further below, we decided not to fit the second mode of the organics as the decrease in the signal above 900 nm was likely due to the reduced transmission efficiency at the upper cut off of the lens (see answer below). The size distribution for the organics was therefore fitted with a monomodal log-normal distribution, as for sulfate.

Line 322-334: How many data points (what percentage of the measuring period) was excluded due to pollution?
43% of the AMS data points were excluded due to pollution. We added a statement with this information at lines 333-334: "In total, 43% of the available AMS measurements (MOCCHA + MOSAiC) were identified as being influenced by local pollution emissions.". As a side note, we would like to mention that this information was already provided to the reader on a monthly (and instrument) basis in Fig. 3b.

Line 364, Fig 2 and Fig 3: Please mark in Figs 2 and 3 the five distinct periods. Also, please note in caption of Fig 2 that one of the five distinct periods does not include AMS data and therefore is not represented with a pie chart in Fig 2.
We updated Fig. 2 and 3 as suggested by the reviewer. In Fig. 2a, the blank gap between the Oct-Dec and the Mar-May period was enlarged and the mention "no AMS data available" was added. The last sentence in the caption for Fig. 2 was updated to emphasize that there is no pie chart for the Jan-Deb period: "Note that, during MOSAiC, the AMS was not operational between early December 2019 and March 2020, and this period is therefore not represented with a pie chart.". For both Fig. 2 and 3, the five periods are now clearly indicated by text and separations. The new figures are shown hereafter.

Fig.2:

[Figure]

Fig. 3:

[Figure]

Figure 3: I suggest to clearly mark the period Dec 2029-Feb 2020 with "no AMS data available" or "AMS not operational".

The mention "no AMS data available" for the Jan-Feb period was added to Fig. 2 as mentioned in the previous comment. We did not include it in Fig. 3 to avoid overcrowding the figure with text. However, we updated the last sentence of the figure caption to clarify that there are no AMS data available for these months: "Median mass concentrations from the AMS are not reported for December 2019 and July 2020 due to low data availability (<10%), and no AMS data are available for January and February 2020."

Line 400, 435, 522 (and maybe elsewhere): In a sentence without the other AMS species, I would prefer "organics" over "Org"

Thank you for the suggestion. We changed "Org" to "organics" everywhere in the manuscript where no other AMS species were mentioned in the sentence.

Line 572 (and chapter 2): How were the supermicron particles N>1000nm measured? If I didn't miss it, this quantity appears here for the first time.

The reviewer is correct that the description of the supermicron particles measurements was overlooked in the method section. We added a paragraph in Sect. 2.3, at lines 280-282: "During MOSAiC, an Aerodynamic Particle Sizer Spectrometer model 3321 (APS; TSI Inc., USA) was used to measure the coarse mode PNSD between 1.06 and 16.1 µm (Heutte et al., 2023b). The supermicron PNC ($N_{>1000nm}$) reported in this work were averaged to 10 min time resolution.".

Line 632ff: the measurements took place at ground level. Are the supermicron particles transported upwards to cloud level efficiently enough to play a role as CCN?

This is indeed an important question that was not tackled in the manuscript. This question was tackled in another manuscript from our group, recently submitted to the journal *Elementa*. Bergner et al. (In review) found that the blowing snow layer extended to median altitudes between 95 and 145m during three blowing snow events during MOSAiC. The analysis was performed using ceilometer and lidar data. Bergner *et al.* also elaborated that "During MOSAiC, 16 % of the cloud base heights from November to April were below 200 m and 25 % below 300 m (not shown), therefore within the altitude range of possible blowing snow aerosol influence.". Below is a figure from the manuscript submitted by Bergner *et al.*, which shows in panel (d) and (e) the vertical extent of the blowing snow (BLSN) layer, as well as the atmospheric boundary layer (ABL) height. Since the ABL height was similar to the cloud base height, the periods when the vertical extent of the BLSN layer and the ABL height coincide correspond to periods when aerosols from sublimated blowing snow could have potentially had a direct influence on the cloud. We added a sentence at lines 625-626 to mention the relevance for clouds: "As discussed by Bergner et al. (In review), the vertical extent of the blowing snow layer made these particles directly relevant at cloud level during MOSAiC.". We will add the full reference for the Bergner et al. manuscript when available. Our manuscript and the one from Bergner et al. are at the same stage in the peer-review process.

[Figure]

line 853ff + Fig 8: To reduce the noise of the AMS size distribution, one can reduce the number of bins. We thank the reviewer for this suggestion. We reduced the number of bins from 100 to 50, and this effectively reduced the noise of the AMS mass size distributions. We hence updated Fig. 8 (now Fig. 7) and Fig. S12 (now Fig. S13) accordingly.

Fig. 7 (previously Fig. 8):

[Figure]

Fig. S13 (previously Fig. S12):

Table S3: The coarse mode of the organics is likely cut off by the lens transmission. Thus, the estimated mode diameter (Table S3) for the organic coarse mode is not representative for the ambient size distribution.

The reviewer is correct that the decrease in the signal above 900 nm for the organics is likely due to the reduced transmission efficiency at the upper cut off of the lens. Hence, we decided to remove the estimated mode diameter for the organic coarse mode in Table S3. We updated the caption for Table S3 accordingly: "Mode location (in vacuum aerodynamic diameter) for the monthly averaged (median) sulfate and organics mass size distributions. The mode diameters were retrieved by fitting a monomodal log-normal distribution to each monthly mass size distribution. Errors represent the fitting error and are not a measure of the statistical variance.", as well as the caption of Fig. 7 (previously Fig. 8): "Dotted lines represent the mode diameter, estimated from fitting a monomodal log-normal distribution through the observations.". Additionally, we updated the sentence at lines 257-259 in the method section: "A

monomodal log-normal distribution was fitted to each monthly mass size distribution to retrieve the mode diameter, using the "Multipeak fitting" package within IGOR Pro v9.02.".

We also updated the discussion in Sect. 3.3.2 accordingly, at lines 822-824: "Apart from some variability around the main mode, a second mode was observed above 650 nm, with an unknown mode diameter above 1 μm. We could not estimate the location of this mode, as the decrease in the signal above 900 nm was likely related with a decrease in the transmission efficiency at the upper cut off of the lens.", and at lines 831-832: "These percentage contributions are, however, lower estimates, due to the decreased lens transmission efficiency near 1 μm.". Finally, we reformulate the sentence at lines 896-897 in the conclusion section: "A second size mode above 650 nm $D_{va}$ was observed for the organics in spring, which represented at least 15-19% of the total submicron organic mass.".

**Anonymous referee #2:**

Based on year-long, chemically-speciated, and size-resolved $PM_1$ measurements during two expeditions using HR-ToF-AMS and collocated instruments, the manuscript titled "Observations of high time-resolution and size-resolved aerosol chemical composition and microphysics in the central Arctic: implications for climate-relevant particle properties" by Heutte et al. presents detailed measurements of aerosol properties in the central Arctic. The comprehensive analysis reveals clear seasonality in $PM_1$ mass concentration, chemical composition, microphysical properties, and sources, driven by typical Arctic seasonal regimes. Specifically, $PM_1$ concentrations were lowest in summer, dominated by organic aerosols, while autumn and winter saw elevated concentrations with anthropogenic sulfate as the dominant species. The study highlights the significant role of cyclonic activity in influencing aerosol variability, where wind-generated particles contribute substantially to the CCN population, particularly in autumn. The long-range transport of anthropogenic emissions from lower latitudes also plays an important but smaller role in elevating the $PM_1$ level in central Arctic. By comparing with observations from a set of pan-Arctic land-based stations, this study pointed out that ammonium in central Arctic appeared to be far less abundant than at lower latitudes, with potential implications in terms of aerosols' acidity. The findings provide a baseline for central Arctic aerosol characteristics, which are crucial for improving climate model predictions.

**General comments:**

1. This study presents a comprehensive analysis of aerosol measurements in the central Arctic, overcoming significant technical challenges associated with conducting such research in remote environments. I recommend publishing this manuscript. However, given its primary focus on measurements, it may be more appropriate to publish it as a measurement report.
   We thank the reviewer for the positive comments and for acknowledging that obtaining year-long measurements from the central Arctic is a true technical challenge. However, we still believe that this manuscript should be considered by the journal as a research article and not a measurement report. Indeed, we used unique high-resolution measurements which allowed us to obtain an unprecedented level of process-understanding. In particular, the influence of synoptic scale events (storms) on the aerosol loading, chemical composition, and CCN population is presented and discussed. This represents a substantial advance in our understanding of short-timescale processes in the central Arctic.

2. While the detailed and informative results are valuable, the manuscript is somewhat lengthy and contains some repetitive information. I suggest streamlining the content to enhance clarity and conciseness. Below are a few examples of sections that could be shortened:
   We agree with the reviewer that the level of detail and quantity of information made the original manuscript lengthy. We also thank the reviewer for providing suggestions on how to shorten the manuscript. We implemented changes as suggested by the reviewer, as discussed in the following points:

P12, L351-363: The caption of Figure 2 already provides detailed information, making this paragraph somewhat repetitive and overly lengthy.
Indeed, some of the information was redundant between the figure caption and the first paragraph of Sect. 3.1. We therefore removed from the first paragraph in Sect. 3.1 all the information that was already provided in the captions of Fig. 2 and 3.

P18, L481-497: Is there a compelling reason to further divide summer into "summer" and "late summer"? Given that the PM$_1$ chemical composition remains similar, the same HR-ToF-AMS was used, and there are no significant variations in chemical composition over years, this separation seems unnecessary. Additionally, these two periods are combined in the conclusions, so it may be more consistent not to separate them initially.

The reason for separating the discussion of these two periods was primarily because they were associated with two different measurement campaigns, taking place at different locations of the central Arctic and during a different year. However, as said by the reviewer, the chemical composition remains similar, the instrument used was the same, and there are no significant variations in chemical composition expected over the course of the two years separating the two expeditions. Hence, we decided to merge Sect. 3.1.1 and Sect. 3.1.5 into one section (now Sect. 3.1.1). The key information from both sections was kept and redundancies common to the two sections were removed. We added a short statement at lines 370-372 to justify the joint discussion of these two distinct periods: "Due to the similarities in aerosol chemical composition between the MOSAiC June – July and the MOCCHA August – September data, the discussion for these two periods is provided jointly (see Sect. 3.1.1).".

The title for Sect. 3.1.1 was renamed accordingly "Jun-Sep: summer (MOCCHA and MOSAiC)".

P26-P28, L697-753: While it's important to assess whether storm-induced high concentration events observed in autumn were also present in spring (when background particle concentrations were higher during the haze) and whether their implications are comparable, instead of giving detailed case studies in spring, it might be more effective to summarize the main conclusions for spring and highlight the most significant differences, and maybe move the details to the supplementary.

Thank you for this suggestion. We decided to move what was formerly Sect. 3.2.2 into the supplement (now Sect. S3.1). Section S3.1 is now introduced at lines 546-548: "Here, we focus our analysis on two major storms which happened in November 2019 (Fig. 5). For comparability, the same case study analysis was performed for spring storms in March 2020, when Arctic haze is present, and the detailed discussion can be found in Sect. S3.1.". The main conclusions on the similarities and differences between spring and autumn storms are presented at lines 626-627 "In spring, we observed similar relative wind-dependent increases in the variables discussed above, although with a smaller magnitude (Sect. S3.1.1).", at lines 649-651 "In spring, we also found a storm-peak increase in eBC by a factor of ~2, but the data availability was insufficient to draw robust conclusions on the source of eBC during the storm (see Sect. S3.1.1).", and at lines 671-673 "In spring, we observed an increase in SO$_4^{2-}$ and Org during a storm on March 15[th], associated with air masses travelling from eastern Siberia (Sect. S3.1.2). The influence from long-range transport during spring storms was, however, partly masked by the high haze background concentrations.".

**Some other minor comments:**

1. P11, L329-331: It is unclear here if any quantitative criteria were used to assess the similarity between the AMS-measured chemical spectrum and fresh hydrocarbon emissions. Additionally, the specific fresh hydrocarbon emission profile being referenced is not clearly defined. While comparing with fresh hydrocarbon emission profiles can help identify pollution periods influenced solely by fossil fuel combustion, how are periods where fossil fuel combustion is mixed with other sources addressed if PMF analysis were not performed yet? The bulk OA profile may differ from pure fossil fuel emissions during these mixed periods, potentially making this approach less effective in excluding all local emissions from research activities. In this context, is it appropriate to label the filtered data as "unpolluted"? Besides

fossil fuel combustion, are there other potential impacts from research activities, such as cooking emissions?

The reviewer is correct that the quantitative criteria to assess the similarity between the AMS spectrum and the fresh hydrocarbon spectra was not explicitly explained in the methods. However, a reference was given to Heutte et al. (2023), where this information can be found. In short, we used a cosine similarity approach, where the data were defined as "polluted" if the cosine similarity (Eq. 1 in Heutte et al. (2023)) value exceeded a certain threshold. This threshold was determined as the cosine similarity value where 80% of the data points were outside of the 120-240° wind direction polluted window, as also explained by Dada et al. (2022). We added the information that the resemblance was defined based on the cosine similarity in the revised manuscript (lines 331-333): "In short, AMS measurements were cleaned from local pollution influence by identifying periods where the measured chemical spectrum resembled (cosine similarity) that of a chosen spectrum of fresh hydrocarbon emissions."

Regarding the periods when fossil fuel combustion pollution is mixed with other sources, the reviewer raises an important point. This question is a central one in an organics source apportionment (on the MOSAiC AMS data) manuscript currently in preparation. There, we were able to extract the fossil fuel combustion pollution mixed with other sources and found that it represented between 1 and 19% of the total organic mass between March and July (this analysis was not performed for other months). In addition, the PMF analysis revealed that all of the major spikes in pollution with concentration >1 µg/m$^3$ were effectively removed by the cosine similarity approach. While the reviewer is right that not all pollution can be removed by our cosine similarity approach, especially the part that is mixed with other sources, it remains the most effective way of excluding time periods directly influenced by local pollution from research activities. As mentioned in Heutte et al. (2023), we also cross-evaluated the results from the cosine similarity approach against other atmospheric variables: wind direction, black carbon concentration, and particle number concentration. Therefore, we believe that labelling the filtered data as "unpolluted" is appropriate in this context. Finally, we did not observe any influence from cooking emissions in the organic mass spectrum, as no such source emerged in our source apportionment study. Other sources of pollution from research activities, including snowmobiles, diesel generators, or helicopters, were also likely removed with our cosine similarity approach.

2. P13, L367-368: Given that the MOCCHA data covers only August to September and is defined as late summer, would it be more accurate to state, "Furthermore, we argue that the MOCCHA data from summer 2018 can be considered representative of the central Arctic Ocean late summer conditions"?

The reviewer's suggestion is indeed more appropriate. The sentence at lines 364-366 was changed accordingly and now reads as follow: "Furthermore, we argue that the MOCCHA data from summer 2018 can be considered representative of the central Arctic Ocean late summer conditions, and hence used to replace the missing MOSAiC late summer (2020) data, for the following reasons.".

3. P13, Figure 2: Would it be clearer to indicate the absence of AMS data between December 2019 and March 2020 by creating a larger gap between the pie charts for Oct-Dec and Mar-May, accompanied by a note such as 'Data not available'?

Thank you for this suggestion. In conjunction with the suggestion from reviewer #1, we enlarged the blank space between the Oct-Dec and Mar-May period in Fig. 2a, and added the following note in-between: "Jan-Feb: no AMS data availability". We also added more space between the pie charts for these two periods (i.e., between Fig. 2c and Fig. 2d).

4. P13, L371: There is inconsistency on the abbreviation of black carbon, alternating between eBC and BC.

Thank you for spotting the inconsistency. We changed "BC" to "eBC" in the manuscript in places where we were referring to the quantity measured by the aethalometer.

5. P15, L395: Are there any quantitative results showing the significance of the temperature drop in October?

Shupe et al. (2022) reported a decrease of the surface temperature from ~-10°C at the beginning of October to ~-20°C at the end of the month. We believe that reporting the values for this change in temperature is not relevant for the information provided in Sect. 3.1.2, hence we changed the wording of the sentence at lines 404-405 accordingly: "October marked the beginning of the dark season in the central Arctic, associated with a decrease in surface temperatures (Shupe et a., 2022).".

6. P17, L449: It's unclear how the background periods/concentrations were defined. Were clean periods, in the absence of pollution, identified as background periods? What are the background concentration levels in other seasons? Or, does the term "background PM$_1$ concentration" refer to the monthly median PM$_1$ concentration? If so, I question whether it's appropriate to refer to this as the background concentration.

The reviewer is right that the term "background" was ambiguous and not clearly defined. We were actually referring to the concentrations outside of the "high concentration events" introduced at line 461. Following the suggestion, we removed the term "background", and lines 459-460 now read as follow: "The spring season (March - May) was characterized by elevated PM$_1$ concentrations, where SO$_4^{2-}$ contributed by 50% to the measured mass, followed by Org (36%), eBC (6%), NH$_4^+$ (3%), NO$_3^-$ (3%), and Chl (2%)."

7. P17, L461-463: Is there any trajectory analysis that supports this conclusion?

The study from Boyer et al. (2023) referenced in the previous sentence at lines 468-470 made use of trajectory analysis to support their finding that "the surface aerosol population was largely influenced by transport from Siberia in spring during MOSAiC". The authors used back-trajectories from the FLEXPART v10.4 model and inverse modelling to show that air masses were predominantly originating from Siberia in spring during MOSAiC, representing the dominant source of haze particles in the particle number size distribution and equivalent black carbon observations.

8. P19, L523-524: Why was only the SO$_4$ sink enhanced, and not Org, if they were transported together? Could it be that Org has stronger local emissions or formation?

The original hypothesis was that the sink in SO$_4^{2-}$ could have been larger due to selective activation in clouds (SO$_4^{2-}$ being more hygroscopic than organic material), provided the two species were externally mixed. However, the reviewer is correct that different emission intensities or different emission sources could represent a simpler and coherent hypothesis. We changed lines 510-511 accordingly: "This could possibly suggest that SO$_4^{2-}$ and Org had different emission intensity or different sources."

9. P23, L626-627: What could be the reason for the elevated background before the second storm case? Is it still valid to treat this elevated period as a background period?

We defined this period as "background" for consistency with the other storms, as a way to directly compare what are the increases in mass and number concentrations between the storm period and the period just preceding it. The elevated background before the storm is also the reason why we introduced (at lines 618-621) a new background period, which corresponded to the low concentration period following the storm. It is possible that the elevated NaCl signal, scattering coefficient, and CCN concentrations before the storm (i.e., between 1:00 and 10:00 UTC on the 23$^{rd}$ of November) were associated with long-range transported sea spray aerosols from the open ocean and/or advected aerosols from sublimated blowing snow particles. This hypothesis is supported by the 10-days back-trajectories (figure below) during this period, showing that the air mass spent most of the time above the Arctic Ocean and, to a lower extent, above the Siberian coastline (including over Novaya Zemlya and the Yamal peninsula). However, it is unlikely that transported haze particles contributed to the high scattering coefficient and CCN concentrations during the "background" period before the storm. Indeed, the NaCl signal was ~10 times higher during the "background" period before the storm compared to the "background" period after the storm, while sulfate mass concentrations were ~1.4 lower before the storm compared to after, as seen in Fig. 5d of the manuscript.

[Figure]

10. P28, L758: It is unclear why these two periods were selected. Could you provide brief context or justification?

Thank you for pointing out that no justification on the choice of the case study periods was provided in the original manuscript. We added a statement at lines 548-549 to justify our choice: "These storms were chosen based on the data availability (i.e., low influence from local pollution emissions) and the condition that the maximum wind speed during the storm exceeded 15 m/s.".

**Anonymous referee #3:**

Heutte et al. present results from the MOCCA and MOSAiC ship-cruises, aiming a year-long measurement of the physicochemical properties of ambient aerosol in the central Arctic. The authors deployed high-resolution time-of-flight aerosol mass spectrometry (HR-ToF-AMS) to analyze the bulk aerosol chemical composition. Further, particle size distribution measurements were conducted and discussed together with the chemical analysis. The authors showed that the seasonal evolution of the aerosol composition is in line with earlier findings from pan-Arctic land-based measurements. Interestingly, ammonium concentrations in the central Arctic were far below measured concentrations at lower latitude land-based stations with potential implication on particle acidity. Individual events were selected to demonstrate the influence of long-range transport, blowing snow etc. on aerosol physicochemical characteristics in autumn and summer. Indeed, spring and autumn composition are largely influenced by such short-term events like wind-driven blowing snow events and synoptically-driven long-range transport.

This work is important for our better understanding of the seasonality in Arctic aerosol. And I highly appreciate the effort of obtaining this data set with all known challenges and technical complexities, especially in the harsh remote environment. The study is in line with the aims and scopes of the journal. Concerns and suggestions (including major issues) are described below.

We thank the reviewer for the positive comments and for providing numerous suggestions to clarify and shorten the manuscript. We address below the concerns from the reviewer.

During revision, the authors should work to improve the writing. There are a few "paragraphs" that consist of 1-3 sentences each and as such do not represent full paragraphs with fully developed thoughts. Please consider to incorporate these sentences into longer paragraphs. A few examples:

- Lines 71-77

Thank you for the suggestion. Since this paragraph was in line with the ideas from the previous one, we move the text at the end of the previous paragraph (now at lines 70-75).

- Lines 116-119

The reviewer is correct that this paragraph is short. However, we decided to keep it as a separate paragraph as the topic of the paragraph (aerosols in the Arctic in autumn) is different from the one of the neighboring paragraphs.

- Lines 186-190

Thank you for the suggestion. The short paragraph describing the MOCCHA campaign was merged with the following paragraph describing the MOSAiC expedition (now at lines 188-191).

- Lines 202-205

The paragraph describing the heating of the sampling lines was merged with the paragraph describing the MOCCHA campaign and MOSAiC expedition. The new paragraph can be found at lines 200-204.

- Lines 476-480

We decided to remove this short paragraph as it contained redundant information (about the collapse of the polar dome, already discussed before at lines 478-480) and mostly served as a transition paragraph to introduce Sect. 3.1.5 which has now been merged with Sect. 3.1.1.

- Lines 693-696

This paragraph is indeed short but here serves as a concluding statement to transition to Sect. 3.2. We decided to keep it as it was.

There are many sentences that are far too long and as such reduce readability. Please shorten here and/or separate in two or more sentences. A few examples:

- Lines 38-41

The original sentence was split into two sentences as follow (lines 39-42): "Cyclonic (storm) activity was found to have a significant influence on aerosol variability by enhancing both emissions from local sources and transport of remote aerosol. Locally wind-generated particles contributed up to 80% (20%) of the cloud condensation nuclei population in autumn (spring).".

- Lines 41-45

We decided to separate the sentences in two, which now reads as follow at lines 42-45: "While the analysis presented herein provides the current central Arctic aerosol baseline, which will serve to improve climate model predictions in the region, it also underscores the importance of integrating short-timescale processes, such as seasonal wind-driven aerosol sources from blowing snow and open leads/ocean in model simulations. This is particularly important given the decline in mid-latitude anthropogenic emissions and the increase in local ones.".

- Lines 156-160

Same here, the original sentence was split into two for better readability. Lines 157-161 now read as follow: "The summer melt season will likely further lengthen at the expense of a shortened winter sea ice growth (Markus et al., 2009; Stroeve et al., 2014), having direct consequences on the coupled ocean-sea ice-atmosphere processes (Willis et al., 2023). The role that transition seasons (i.e., spring and autumn) will play for this changing seasonality is yet to be elucidated with present-day measurements.".

- Lines 252-255

We agree that the original sentence was hard to follow. We decided to split it in two as follow (lines 252-255): "This relation is linearly dependent on the particle density ($\rho$), if we assume that particles are spherical ($D_{va} = D_m * \rho$). Hence, an uncertainty in the density estimated from the particle chemical composition (see calculation details in Heutte et al. (2023b)) would propagate into an uncertainty of the same magnitude for the mass size distribution.".

- Lines 361-364

We separated this sentence into two and removed some information that was already given in the captions of Fig. 2 and 3. Lines 358-360 now read as follow: "For completeness, we show the seasonality of the total aerosol volume ($V_{tot}$) for particles between 18 and 500 nm in mobility diameter. $V_{tot}$ is used as a proxy for $PM_1$ for months when AMS data are missing.".

- Lines 437-440

The original sentence was reformulated and separated in two to improve readability. Lines 447-451 now read as follow: "Wintertime oxidation pathways could have, however, still resulted in $SO_4^{2-}$ being a dominant species in the dark January and February months, in the context of anomalously high positive AO. Such pathways involve, for instance, the metal-catalyzed in-cloud oxidation of $SO_2$ by $O_2$ (Alexander et al., 2009; McCabe et al., 2006) and the poleward transport of $SO_4^{2-}$ formed at lower latitudes where sunlight is available for photo-oxidation.".

- Lines 443-446

We agree with the reviewer that the sentence is rather long. However, the sentence includes necessary clarifications in parenthesis that increase its length. We believe that this sentence was well punctuated and readable. We decided to keep the sentence in its original form (now at lines 453-456).

- Lines 451-455

The sentence was slightly reformulated to reduce its length (lines 461-464): "A number of high mass concentration events were also observed, such as on March 15th when $PM_1$ mass concentration neared 2 $\mu g/m^3$, and during two intense episodes of warm and moist air mass intrusions from northern Eurasia on April 15th and 16th, when pollution levels ($[PM_1] \geq 4$ $\mu g/m^3$) became comparable to central-European urban pollution levels (Dada et al., 2022a).".

- Lines 476-480

As mentioned above, this short paragraph was removed since it was redundant and no longer served its role of transition from Sect. 3.1.4 to 3.1.5, the latter having been merged to Sect. 3.1.1.

- Lines 535-538

This sentence is indeed lengthy. We feel that this sentence shouldn't be split as it contains a statement and an example to illustrate that statement. We replaced a comma by a semicolon to "separate" the statement from the example, and reformulated slightly the sentence (now at lines 523-526): "In light of the decreasing sulfate concentrations in the Arctic (Schmale et al., 2022), efforts should be maintained to rigorously monitor aerosol chemical composition in the future, as a range of aerosol physicochemical processes depend on the particles' acidity (Pye et al., 2020); for example, the partitioning of nitrate into the particle phase tends to increase as the sulfate-to-ammonium ratio decreases (Sharma et al., 2019)."

- Lines 802-806

We reformulated and split this sentence to improve the overall readability. The new sentences at lines 742-745 read as follow: "To estimate the contribution of each "main source" (cluster) to the CCN population, we divided the summed CCN number concentrations associated with a given cluster throughout the October-November period by the total (all clusters) summed CCN number concentrations for the same period. We used CCN concentrations at 0.3% SS, which is assumed to be a representative SS level in autumn in the Arctic (Motos et al., 2023).".

Further, the manuscript is very lengthy and partly repetitive. Please re-consider if you can shorten or summarize some paragraphs (some suggestions are listed below). The readability of the manuscript would further benefit from removing redundant sub-clauses, fillers, details in parentheses etc. (some of which are listed below).

According to the reviewer's suggestions, we shortened the manuscript by removing some redundant information, merging some sections together, and moving material to the supplement. Details on the

changes made to the manuscript are found below in the answer to the reviewer's comments. However, for most cases, we decided to keep the original details given in parenthesis as these were important components for the comprehension of the manuscript and for a rigorous treatment of the statistics (e.g., correlation coefficient and p-values).

**Major comments:**

- 3.1.1, Sect. 3.1.5, and lines 880 ff: What about methanesulfonic acid (MSA)? As MSA plays a major role for the sulfur content in Arctic summer, you should consider looking into the detection of MSA with the HR-ToF-AMS. The publication by Zorn et al. (2008) provides information on how to extract mass concentration for MSA from HR-ToF-AMS field measurements. Further, you might can use the MSA-to-sulfate as well as the Org-to-sulfate ratios to better discriminate the contribution from anthropogenic and marine sources to particulate sulfur and organics (see Willis et al. (2017)). Along with your large distance to the open ocean, it is highly interesting to see if MSA and marine-biogenic organics still play a role.

Indeed, dimethyl sulfide (DMS) and its oxidation products are important for the organics and sulfate aerosol populations in the Arctic in summer, and merits proper discussion. Our HR-ToF-AMS was actually calibrated for MSA, so that we could quantify it. There is a lot of information related to MSA and marine organics in our dataset and we are working on a dedicate manuscript to have te space to properly discuss this. This follow-up manuscript was mentioned at the end of Sect. 3.1.5 (now part of Sect. 3.1.1). We reformulated the sentence slightly to incorporate the information that MSA will also be discussed in that follow-up manuscript (at lines 400-402): "A follow-up source apportionment study will elucidate the sources associated with organic aerosols (including MSA) in the summertime central Arctic during MOSAiC.".

- 3.1.1: Can you explain the enhanced nitrate signal in August and September compared to June and July (Sect. 3.1.5)? In particular, which form of nitrates was observed here – organic nitrates or ammonium nitrate? The latter one plays obviously a minor role with regard to the very low ammonium mass concentration. It would be worth to check the presence of organic nitrates with the HR-ToF-AMS data.

The reviewer is right that ammonium nitrate is likely not contributing much to the enhanced nitrate signal, given the very low ammonium concentrations. However, the $NO_3^-$ timeseries (and fractional contribution) shown in Fig.2 corresponds to inorganic nitrates, derived from the signals of the $NO^+$ and $NO_2^+$ ion fragments (and isotopes) in the spectrum. While it is possible to get $NO^+$ and $NO_2^+$ fragments from the fragmentation of organic nitrates in the AMS (e.g., Huang et al., 2021), we found no correlation between the concentrations in organic nitrate fragments ($C_xH_yO_zN^+$) and inorganic nitrate ($NO_3^- = NO^+ + NO_2^+$). Hence, it is unlikely that organic nitrates would have contributed to the $NO_3^-$ signal. We believe that the $NO_3^-$ signal was likely overestimated due to potential interference between the $NO^+$ and $C^{18}O^+$ fragments ($mz$ 30) in the AMS, rendering the analysis of $NO_3^-$ difficult for August and September. The two fragments are only 0.001 mz apart, which is less than the mass resolving power of the instrument.

We added this information at lines 377-379 "However, it should be noted that $NO_3^-$ measurements during MOCCHA were likely overestimated due to interferences between the $NO^+$ and $C^{18}O^+$ fragments at $mz$ 30 in the AMS. Hence, $NO_3^-$ will not be further discussed in this section.".

- Sect 3.1.2: Similar to the comment above - Which form of nitrates do you observed here in October to December? Ammonium is not available (below DL)– please consider the presence of organic nitrates.

For the October-December period, we also found no correlation between organic nitrate fragments and $NO_3^-$, suggesting that organic nitrates were not contributing to the nitrate signal. Here, we believe that interferences with $C^{18}O^+$ at *mz* 30 are also likely to have influenced the nitrate signal. We added a statement at lines 407-408: "As for the MOCCHA measurements, $NO_3^-$ likely suffered from interferences with the $C^{18}O^+$ fragment at *mz* 30, which could explain its relatively high fractional contribution to the $PM_1$ mass.".

- 3.2 and Lines 880 ff: Is it correct that aerosol particles were also collected and subsequently analyzed by offline-techniques (by Pratt et al.)? As your results on the abundance of sea spray in spring and autumn provide an important part of your manuscript, your analysis could benefit from such additional evidence by aerosol off-line techniques. You can obtain more information on mixing states (internal mixing with organics?), size and maybe even mass.

The reviewer is correct that aerosols were also collected on filters for offline analysis during MOSAiC. It is also true that such offline-techniques provide important additional information on the morphology, chemical composition, and mixing state of the particles. We have been working in tight collaboration with Pratt et al., who have also observed sub and supermicron-sized organic-coated sea salt particles using scanning electron microscopy with energy dispersive X-ray spectroscopy (SEM-EDX). Given the abundance of information contained in the offline-analyzed samples, Pratt et al. will publish a separate manuscript with details regarding the sea salt aerosols measured during MOSAiC. Their findings should complement the findings reported in the current manuscript.

**Minor comments:**

- Your abstract would generally benefit from a few more sentences elucidating your main results. For example, I suggest to add a sentence on the comparison between your central Arctic ship-based measurements and the pan-Arctic land-based observations like it was discussed in Sect. 3.1.6 and in lines 914 ff. Another example, I think your results on enhanced BC from blowing snow and sublimation is important and should be mentioned in the abstract.

We agree with the reviewer that the comparison with the pan-Arctic land-based observations deserved to be mentioned in the abstract. We reformulated the abstract to add this information and stay within the 250 words limit. In particular, the sentence at lines 35-37 now highlights the comparison with the pan-Arctic observations: "Seasonal variations in aerosol mass concentrations and chemical composition in the central Arctic were found to be driven by typical Arctic seasonal regimes, and resemble those of pan-Arctic land-based stations.". Regarding the enhanced eBC mass concentrations during blowing snow, we decided to keep it out of the abstract due to the lack of strong observational evidences (e.g., continuous in situ observations of eBC in snow) to validate the hypothesis.

- Line 39: "emissions" instead of "emission"

Done.

- Line 86: Do you mean "aged organics" instead of "aged sulfate"?

"Aged" here didn't refer specifically to sulfate but to all of the compounds listed in the sentence. For clarity, we moved "aged" before in the sentence which now reads as follow (at lines 84-87): "Haze is primarily composed of aged accumulation mode particles, comprising a mixture of sulfate, organics, black carbon, ammonium, and nitrate […].".

- Lines 91-94: Please re-formulate this sentence. Suggestion: "There are a few key mechanisms that control particle activation potential, for example, atmospheric aging […]"

We reformulated the sentence by splitting it into two, for better readability. Lines 89-92 now read as follow: "Atmospheric aging during air mass transport is a key mechanism that controls particle activation potential, especially for black carbon and organic species (Ervens et al., 2010; Jimenez et al., 2009; Liu et al., 2011). Aging can occur through condensation of low-volatility gases on existing particles, coagulation processes, cloud processing, and/or photooxidation reactions.".

- Lines 103-107: Please re-formulate this sentence. Suggestion: "This is associated with […]. As a result, the summertime Arctic (June-August) is characterized […]."

We thank the reviewer for the suggestion. We reformulated lines 101-105 as follow: "This is associated with more frequent precipitation and a weaker atmospheric stratification, both locally and along the trajectory of transported air masses. As a result, the summertime Arctic (June-August) is characterized by relatively low aerosol mass concentrations from more local/regional emissions (Stohl, 2006). The aerosol population in summer is dominated by Aitken mode and nucleation mode particles originating from local biogenic sources (Boyer et al., 2023; Freud et al., 2017; Pernov et al., 2022; Tunved et al., 2013; Willis et al., 2017).".

- Lines 106-109: Please re-formulate as there are too many "and", "or" and commas that reduce readability.

The sentence at lines 104-108 was split in two for better readability: "The aerosol population in summer is dominated by Aitken mode and nucleation mode particles originating from local biogenic sources (Boyer et al., 2023; Freud et al., 2017; Pernov et al., 2022; Tunved et al., 2013; Willis et al., 2017). These include primary marine and terrestrial aerosols, or secondary particles formed via new particle formation (Baccarini et al., 2020; Beck et al., 2020; Brean et al., 2023; Schmale and Baccarini, 2021) or condensation of precursor gases onto pre-existing particles (Willis et al., 2016).".

- Lines 106-114: Please also check references from the aircraft-based mission NETCARE in summer 2014. As for example: Willis et al. (2016), Willis et al. (2017), and Koellner et al. (2021).

Thank you for suggesting to add these additional references, which fit perfectly to the statements of the paragraph. We added all the three references suggested by the reviewer at lines 104-108 (see text in the two previous comments from the reviewer) and at lines 108-110: "Organic aerosols from different sources contribute significantly to the submicron aerosol mass concentrations in summer (e.g., Chang et al., 2011; Croft et al., 2019; Fu et al., 2009, 2013; Köllner et al., 2021; Leaitch et al., 2018; Moschos et al., 2022b; Nielsen et al., 2019; Siegel et al., 2021).".

- Line 134: Please re-formulate: "[…] and an observation bias in the central Arctic summer."
Done.

- Your introduction could benefit from a few sentences elaborating the representativeness of your ship-based measurements (or more general ground-based measurements in the boundary layer) compared to the aerosol vertical column with respect to low inversions, influence of low-level clouds etc. (see references Willis et al. (2019), Schulz et al. (2019), and Köllner et al. (2021)).

While this is not a central focus of the manuscript, the reviewer is right that the representativeness of surface-based measurements with respect to the tropospheric vertical column should be mentioned in the introduction. We added a short statement in the instruction at lines 131-136:

"Despite the year-round observations […], as well as a lack of vertical profiles. The Arctic boundary layer is highly stratified for most of the year due to strong temperature inversions (Jozef et al., 2024), which means that surface and ship-based observations are generally only representative of the Arctic boundary layer (Köllner et al., 2021; Willis et al., 2019).".

- Line 182: Do you mean "These two expeditions were set up to […]" instead of "[…] set out to […]"?

No, we meant "[…] set out to […]". In this context "set out" means "to start something with the aim to…".

- Line 183: Remove "driving" – redundant.

We understand the reviewer's point to remove the redundant information. However, we don't feel that "driving" in this sentence is redundant. Here we want to express the idea that the coupled atmospheric-ice-ocean-ecosystem processes can either be drivers of the changes occurring in the Arctic, or be driven ("influenced") by these changes. These are two different information that, we believe, are not redundant.

- Lines 193-194: Remove "To provide context into the sea ice extent during that year," – redundant.

Thank you for the suggestion, we removed that part of the sentence and lines 192-193 now read as follow: "We show in Fig.1 the minimum and maximum sea ice extent, respectively reached on September 15th and March 5th, 2020.".

- Lines 195-196: Please re-formulate: "Polarstern was in general […], except for the drift […]."

Thank you for the suggestion. The sentence at lines 194-195 has been reformulated as follow: "*Polarstern* was in general far away from the marginal ice zone and the open ocean, except for the drift period during leg 4 in mid-summer (i.e., between June 19th and July 31st)."

- Line 240: Remove "(A.M.)" – redundant as it is mentioned above. Please check throughout the manuscript.

Following the reviewer's suggestion, we mentioned once at line 219 that the averaging was based on the arithmetic mean and removed the "(A.M.)" abbreviation from the rest of the manuscript.

- Lines 257-258: Remove "Therefore, this does not […] for example." – redundant.

We think that the reviewer meant lines 357-358 here. Indeed, this information was already provided in the method section (Sect. 2.2), so we removed it from the first paragraph of Sect. 3.1.

- Lines 358-359: Please re-formulate. Suggestion: "Figure 3 shows the annual cycle of each species with monthly statistics."

We re-formulated the sentence at line 358, as suggested by the reviewer: "Figure 3 shows the annual cycle of each species with monthly statistics (median and interquartile range)."

- Your discussion would generally benefit from a few sentences describing how representative was your measurement period (MOCCHA and MOSAiC) compared to the climatological mean.

It is not entirely clear what the reviewer is referring to (climatology of temperature, wind speed, or precipitation?) and where in the manuscript. When appropriate, we discussed what was specific to the MOSAiC year in terms of climatology. For example, the positive anomaly of the Arctic Oscillation between January and March 2020, affecting the occurrence and intensity of haze, was

mentioned and discussed in Sect. 3.1.3. In Sect. 3.2, we also discussed the anomalous nature of the cyclonic conditions during MOSAiC, leading to intense storms (Rinke et al., 2021).

- Figure 3 caption last line: Do you mean "[…] not reported for December 2019 until February 2020 […]"?

There are a few data points available for December 2019 and July 2020 (as shown in Fig. 2a). However, we did not compute monthly statistics for these months as the data coverage was too low. The situation is different for January and February 2020, as there are simply no AMS data available. This is why we made the distinction for these months in the caption of Fig. 3.

- Line 410: Please change "contribution of sea salt in the autumn aerosol budget" to "contribution of sea salt to the aerosol budget in autumn".

Done. Lines 419-420 now read as follow: "Using instead the NaCl$^+$ fragment at $m/z$ 58 (Ovadnevaite et al., 2012), we discuss in Sect 3.2 and 3.3 the contribution of sea salt in the aerosol budget in autumn."

- 3.1.5: I suggest to discuss Sect. 3.1.5 and 3.1.1 together. I understand the separation between MOSAiC and MOCCHA data. However, the results and discussion are comparable. This would largely shorten the manuscript.

We thank the reviewer for this suggestion. As described in the answer to reviewer #2, we decided to merge Sect. 3.1.1 and Sect. 3.1.5 into one section (now Sect. 3.1.1, renamed "Jun-Sep: summer (MOCCHA and MOSAiC)").

- Lines 489-490: Please remove "(i.e. change […])"- redundant.

This sentence was part of Sect. 3.1.5, which has been merged with Sect. 3.1.1 now. This sentence is no longer there, as it was redundant information with the original section on the Aug-Sep period.

- Line 416: Remove "also".

Done.

- 3.1.2- 3.1.4: I miss a discussion on the abundance and sources of BC. Especially, for spring, I think BC is an important player for the Arctic haze period. I suggest to add a few more sentences on the abundance and sources of BC during your measurements.

BC is indeed an important component of the haze aerosol population. We made the deliberate choice to provide a succinct discussion on eBC as this was done with much more details for the MOSAiC year by Boyer et al. (2023), as referenced in Sect. 3.1.3 (lines 435-437).

- Line 465: Do you really mean "lower stratosphere"? To my knowledge, both references (Fisher et al. (2011) and Willis et al. (2019)) do not focus on the stratosphere. The authors show the contrast in acidity between the boundary layer and the free troposphere.

The reviewer is correct that both studies cited were comparing concentrations in the free troposphere to the boundary layer. We reformulated the sentence at lines 473-476 accordingly: "Observational and modelling studies have shown strong vertical gradient of NH$_4^+$/SO$_4^{2-}$ ratio in the springtime Arctic, with higher concentration of NH$_4^+$ in the upper (free) troposphere than in the boundary layer, resulting from a stronger contribution of East Asian anthropogenic (agricultural) NH$_4^+$ emissions at higher altitudes (Fisher et al., 2011; Willis et al., 2019).".

- 3.1.6 and Fig. 4: I do not understand why the authors have chosen variable time periods for averaging across the stations? And isn't it more appropriate to select pan-Arctic measurements only for the sampling period of MOCCA and MOSAiC for comparison reasons?

We decided to use the same dataset as in Moschos et al. (2022), for which the pan-Arctic measurements cover variable time periods, according to data availability. The time periods covered by the pan-Arctic measurements (i.e., between 2015 and 2019) are only partially overlapping the MOCCHA-MOSAiC period (i.e., 2018-2020), as there are no published chemical composition data available yet for the stations between 2019 and 2020. However, the interquartile range for the pan-Arctic measurements expresses a large variability and we considered this information as an advantage to place our new central Arctic data into context.

- 4a bottom (nitrate): It is difficult to identify any trend or comparison with the station as the range of y-axis is too large. Please adjust it.

We adjusted the y-axis range for nitrate in Fig. 4a (bottom), which now goes up to 0.25 µg/ m³. The original y-axis was set as such to show the outlier value for PAL in April. Because this data point is now invisible with the new y-axis range, we added a sentence in the caption of Fig. 4: "The y-axis for nitrate in (a) was cropped for readability (the value for PAL in April is equal to 0.654 µg/m³).".

- Lines 523-524: I do not understand your hypothesis. Why should have sulfate and organics different sink processes if they have the same origin and transport way? Isn't it more feasible that different sources can explain the discrepancy? Anyhow, this is very speculative if you have no evidence (trajectories, modelling etc.).

As formulated in the answer to reviewer #2, we now changed lines 510-511 to "This could possibly suggest that $SO_4^{2-}$ and Org had different emission intensity or different sources.". The reviewer is correct that this new, suggested statement remains speculative. However, we believe that we worded the sentence in a cautiously enough manner to express that it is only a hypothesis ("could possibly suggest [...]").

- Line 530: This paragraph is in general very long. I suggest to start here a new paragraph with the topic of "ammonium in the central Arctic".

Thank you for the suggestion, we started a new paragraph (at line 518) on the topic on "ammonium in the central Arctic" as suggested.

- Line 532: This is a very important result!

We thank the reviewer for pointing out the importance of this result, which was also highlighted in the abstract and in the conclusion.

- Lines 538-540: Please consider if it is really necessary to show Fig. 4b if you discuss this subfigure with only one sentence.

We understand the reviewer's concerns about the necessity of Fig. 4b. However, we believe that this figure is a strong additional evidence for the concluding argument that "long-term observations at Arctic land-based stations are relevant to the central Arctic seasonal cycle of chemical composition and mass loading." (lines 530-531). We show with this figure that, not only are the measured concentrations in the central Arctic in the same range as at the pan-Arctic stations, but the timing at which min/max concentrations are observed is also similar.

- 3.2: Can you explain the reasons for selecting particularly these four events/case studies?

As mentioned in our answer to a comment from reviewer #2, we added a statement at lines 548-549 to justify our choice: "These storms were chosen based on the data availability (i.e., low influence from local pollution emissions) and the condition that the maximum wind speed during the storm exceeded 15 m/s.".

- Lines 522-523: I cannot find the answer to the question "How often do we observe significant …?" in Sect. 3.2. I guess the answer is given in Sect. 3.3. Please re-structure or re-formulate this.

The reviewer is correct that no answer to the question "How often do we observe significant […]" was provided in Sect. 3.2. We reformulated the sentence at lines 539-542 to remove that part of the question: "Are there any significant changes in aerosol chemical composition on shorter timescales over the central Arctic Ocean? If so, by how much do the aerosol mass and number concentrations deviate from the background conditions or monthly medians/means, what are the sources of the particles, and what are their contributions to the CCN population and direct radiative budget?".

- Sects 3.2.1 and 3.2.2: Please re-consider if it necessary to discuss "spring" and "autumn" events separately as results and implications are comparable. It would probably shorten the paper if you discuss it in a more compact way and leave details for the reader in the Supplement.

Thank you for the suggestion. As described in our answer to a similar comment from reviewer #2, we decided to move what was formerly Sect. 3.2.2 into the supplement (now Sect. S3.1). The main conclusions on the similarities and differences between spring and autumn storms were added to what are now Sect. 3.2.1 and 3.2.2.

- 5/6: Change "expect" to "except". The blue tones for nitrate and NaCl are difficult to differentiate. Please consider if it is necessary to show both "LF" parameters in (a) and (b) – redundant?

We changed "expect" to "except" in the caption of Fig. 5 and Fig. 6 (now Fig. S3). We also changed the color for NaCl in both figures, the light blue has been replaced by a purple color. Regarding the "LF" parameters, we believe that it is important to show both. Indeed, as mentioned in Sect. 2.4.5, $LF_{5x\ accu,\ div}$ considers leads that have been opening, closing, or staying opened over a period of 5 days. In our argumentation, it is important to show that, even when considering lead opening within this larger time window, the lead fraction remains low compared to the area covered by sea-ice.

- Lines 591 ff: This paragraph would benefit from a topic sentence stating the main message of this paragraph.

We agree with the reviewer that this paragraph was lacking context, and was actually mixing information on the relation between NaCl and $N_{>1000nm}$ with correlations to climate-relevant variables. The information on the relation between NaCl and $N_{>1000nm}$ was partly redundant with the previous paragraph and was hence merged with the previous paragraph. The paragraph at lines 589-609 now starts directly with the topic of "effect on climate-relevant variables".

- Line 591: In general, how did you define you background periods?

The background periods were chosen as the periods just preceding the start of the storm, when we had data available (i.e., not polluted). In theory, the background period preceding the storm was chosen to be of 24 hours, but data availability constrained this period to variable durations for the different storms.

- Lines 601-603: Please shorten this sentence. Suggestion: "[…] by a factor of 4 to 7 depending on the SS levels."

We thank the reviewer for the suggestion, and agree that the original sentence was lengthy and hard to follow. Instead of removing the details in parenthesis, which are essential to understand the particle activation behavior, we split the original sentence in two. Lines 590-594 now read as follow: "We found correlations ($\rho_{pearson}$) between NaCl signal and CCN number concentrations between 0.84 and 0.88 depending on the SS level (all p-values < 0.001) during blowing snow. Compared to the background period, CCN number concentrations during the storm peak increased by factors of ~4 (from 27.0 to 119.3 cm$^{-3}$), ~5 (from 30.2 to 144.5 cm$^{-3}$), ~5 (from 32.1 to 161.3 cm$^{-3}$), ~6 (from 32.4 to 186.0 cm$^{-3}$), and ~7 (from 33.1 to 228.3 cm$^{-3}$), at SS levels of 0.15, 0.2, 0.3, 0.5, and 1% respectively."

- Line 693-694: Please re-formulate. Suggestion: "Overall, CCN number concentrations are influenced by both wind-driven local aerosol production from blowing snow and SSA as well as from long-range transported aerosol under cyclonic conditions. Yet, the latter process plays a minor role."

Thank you for the suggestion. We reformulated the sentences at lines 689-691 accordingly: "Overall, under these cyclonic conditions, CCN number concentrations are influenced by both wind-driven local aerosol production from blowing snow and SSA as well as from long-range transported aerosols. Yet, the latter process plays a smaller role."

- Line 700: Remove "logically" – redundant.

Done. The section in which the sentence originally was is now in the supplement (Sect. S3.1, lines 82-84).

- Line 711 and others: Is it really necessary to indicate the coefficients in parenthesis? I suggest to remove these details from the main text as it limits readability.

We understand the reviewer's question on the necessity of presenting the correlation coefficients in the text. However, we decided to keep this information in the main text, as we want to provide a full and rigorous treatment of the statistics presented in the manuscript.

- Line 754: I suggest to state "[…] during spring and autumn" instead of "transitions seasons".

Indeed, "transition seasons" might not be a clear and objective formulation for the reader. We hence renamed the title of Sect. 3.3 to "Aerosol size distributions during autumn and spring".

- Line 791: I suggest to start here a new paragraph for better readability.

Thank you for the suggestion. We decided to keep the original paragraph as it was, i.e., one paragraph discussing the PNSD clusters in October-November and one for the March-April period. We believe that further splitting of these paragraphs would likely disrupt the flow and logic of the section.

- Line 792: Remove "more" -redundant.

Done.

- Line 795-797: The comparisons are incomplete as written.

It is unclear what the reviewer meant by "incomplete". The comparison here relies on the fact that the *LRT aged* cluster is characterized by lower particle number concentrations and an accumulation mode shifted to larger sizes. Furthermore, a comparison to published literature was provided (Lange

et al., 2018, 2019; Pernov et al., 2022), where the authors provided further explanations on the nature of these two clusters. We decided to add the study from Dall'Osto et al. (2019) in the references cited at lines 738-739. The authors also found two distinct accumulation mode clusters (with main modes at 150 and 220 nm) at three different Arctic monitoring sites. The cluster for the accumulation mode at 220 nm was found to occur mostly in October-December, while the cluster with the accumulation mode at 150 nm peaked during the February-April period.

- Line 797: I guess it is possible to check the transport time scales with your trajectory analysis. Any indications?

The reviewer's suggestion to look at the back-trajectories associated with the two *LRT* clusters is very pertinent. We tried such an analysis and the main caveat was that to estimate the transport time we would need to know the exact source of the particles. Indeed, the back-trajectories are simulated for a given period back in time (0.25, 1, 7, 10, or 30 days), and each grid cell is associated with a residence time. Hence, summing the residence time in each grid cell along the trajectory would require that the exact point source is known.

- Line 829: Remove "were".

We thank the reviewer for the suggestion. However, the sentence would be grammatically incorrect without "were" for the "were discussed" part. We hence decided to keep the sentence at it is.

- Line 830: I suggest to start here a new paragraph for better readability.

Similarly to the comment above, we feel that splitting this paragraph would disrupt the comparison between the October-November period and the March-April period. Hence, we decided to keep this paragraph whole.

- Line 886: What is meant by "That is,"?

Here we meant "indeed" by "that is". For clarity, we reformulated the sentence at lines 827-829 accordingly: "Indeed, we obtained median organics mass concentrations from the integrated size distributions […].".

- Line 932: Remove the details in parenthesis.

As these statistical details were already given in Sect. 3.2, we removed the details in parenthesis at lines 874-875, as suggested by the reviewer.

References (from anonymous referee #3):

Zorn, S. R., et al.: Characterization of the South Atlantic marine boundary layer aerosol using an aerodyne aerosol mass spectrometer, Atmos. Chem. Phys., 8, 4711–4728, https://doi.org/10.5194/acp-8-4711-2008, 2008.

Willis, M. D., et al.: Evidence for marine biogenic influence on summertime Arctic aerosol, Geophys. Res. Lett., 44, 6460–6470, doi:10.1002/2017GL073359, 2017.

Willis, M. D., et al.: Growth of nucleation mode particles in the summertime Arctic: a case study, Atmos. Chem. Phys., 16, 7663–7679, https://doi.org/10.5194/acp-16-7663-2016, 2016.

Köllner, F., et al.: Chemical composition and source attribution of sub-micrometre aerosol particles in the summertime Arctic lower troposphere, Atmos. Chem. Phys., 21, 6509–6539, https://doi.org/10.5194/acp-21-6509-2021, 2021.

Willis, M. D., et al.: Aircraft-based measurements of High Arctic springtime aerosol show evidence for vertically varying sources, transport and composition, Atmos. Chem. Phys., 19, 57–76, https://doi.org/10.5194/acp-19-57-2019, 2019.

Schulz, H., et al.: High Arctic aircraft measurements characterising black carbon vertical variability in spring and summer, Atmos. Chem. Phys., 19, 2361–2384, https://doi.org/10.5194/acp-19-2361-2019, 2019.

**Additional changes:**

- A new affiliation was added for the author Jakob B. Pernov: "ᶜ Now at: School of Earth and Atmospheric Sciences, Queensland University of Technology, Brisbane, Australia"

**References**

Bergner, N., Heutte, B., Beck, I., Pernov, J. B., Angot, H., Arnold, S. R., Boyer, M., Creamean, J. M., Engelmann, R., Frey, M. M., Gong, X., Henning, S., James, T., Matrosov, S. Y., Mirrielees, J. A., Petaja, T., Pratt, K. A., Quéléver, L. L. J., Schneebeli, M., Uin, J., Wang, J., and Schmale, J.: Characterization and effects of aerosols during blowing snow events in the central Arctic, In review.

Boyer, M., Aliaga, D., Pernov, J. B., Angot, H., Quéléver, L. L. J., Dada, L., Heutte, B., Dall'Osto, M., Beddows, D. C. S., Brasseur, Z., Beck, I., Bucci, S., Duetsch, M., Stohl, A., Laurila, T., Asmi, E., Massling, A., Thomas, D. C., Nøjgaard, J. K., Chan, T., Sharma, S., Tunved, P., Krejci, R., Hansson, H. C., Bianchi, F., Lehtipalo, K., Wiedensohler, A., Weinhold, K., Kulmala, M., Petäjä, T., Sipilä, M., Schmale, J., and Jokinen, T.: A full year of aerosol size distribution data from the central Arctic under an extreme positive Arctic Oscillation: insights from the Multidisciplinary drifting Observatory for the Study of Arctic Climate (MOSAiC) expedition, Atmospheric Chemistry and Physics, 23, 389–415, https://doi.org/10.5194/acp-23-389-2023, 2023.

Dada, L., Angot, H., Beck, I., Baccarini, A., Quéléver, L. L. J., Boyer, M., Laurila, T., Brasseur, Z., Jozef, G., de Boer, G., Shupe, M. D., Henning, S., Bucci, S., Dütsch, M., Stohl, A., Petäjä, T., Daellenbach, K. R., Jokinen, T., and Schmale, J.: A central arctic extreme aerosol event triggered by a warm air-mass intrusion, Nat Commun, 13, 5290, https://doi.org/10.1038/s41467-022-32872-2, 2022.

Dall'Osto, M., Beddows, D. C. S., Tunved, P., Harrison, R. M., Lupi, A., Vitale, V., Becagli, S., Traversi, R., Park, K.-T., Yoon, Y. J., Massling, A., Skov, H., Lange, R., Strom, J., and Krejci, R.: Simultaneous measurements of aerosol size distributions at three sites in the European high Arctic, Atmospheric Chemistry and Physics, 19, 7377–7395, https://doi.org/10.5194/acp-19-7377-2019, 2019.

Heutte, B., Bergner, N., Beck, I., Angot, H., Dada, L., Quéléver, L. L. J., Laurila, T., Boyer, M., Brasseur, Z., Daellenbach, K. R., Henning, S., Kuang, C., Kulmala, M., Lampilahti, J., Lampimäki, M., Petäjä, T., Shupe, M. D., Sipilä, M., Uin, J., Jokinen, T., and Schmale, J.: Measurements of aerosol microphysical and chemical properties in the central Arctic atmosphere during MOSAiC, Sci Data, 10, 1–16, https://doi.org/10.1038/s41597-023-02586-1, 2023.

Huang, W., Yang, Y., Wang, Y., Gao, W., Li, H., Zhang, Y., Li, J., Zhao, S., Yan, Y., Ji, D., Tang, G., Liu, Z., Wang, L., Zhang, R., and Wang, Y.: Exploring the inorganic and organic nitrate aerosol formation regimes at a suburban site on the North China Plain, Science of The Total Environment, 768, 144538, https://doi.org/10.1016/j.scitotenv.2020.144538, 2021.

Köllner, F., Schneider, J., Willis, M. D., Klimach, T., Helleis, F., Bozem, H., Kunkel, D., Hoor, P., Burkart, J., Leaitch, W. R., Aliabadi, A. A., Abbatt, J. P. D., Herber, A. B., and Borrmann, S.: Particulate trimethylamine in the summertime Canadian high Arctic lower troposphere, Atmospheric Chemistry and Physics, 17, 13747–13766, https://doi.org/10.5194/acp-17-13747-2017, 2017.

Lange, R., Dall'Osto, M., Skov, H., Nøjgaard, J. K., Nielsen, I. E., Beddows, D. C. S., Simo, R., Harrison, R. M., and Massling, A.: Characterization of distinct Arctic aerosol accumulation modes and their sources, Atmospheric Environment, 183, 1–10, https://doi.org/10.1016/j.atmosenv.2018.03.060, 2018.

Lange, R., Dall'Osto, M., Wex, H., Skov, H., and Massling, A.: Large Summer Contribution of Organic Biogenic Aerosols to Arctic Cloud Condensation Nuclei, Geophysical Research Letters, 46, 11500–11509, https://doi.org/10.1029/2019GL084142, 2019.

Moschos, V., Schmale, J., Aas, W., Becagli, S., Calzolai, G., Eleftheriadis, K., Moffett, C. E., Schnelle-Kreis, J., Severi, M., Sharma, S., Skov, H., Vestenius, M., Zhang, W., Hakola, H., Hellén, H., Huang, L., Jaffrezo, J.-L., Massling, A., Nøjgaard, J. K., Petäjä, T., Popovicheva, O., Sheesley, R. J., Traversi, R., Yttri, K. E., Prévôt, A. S. H., Baltensperger, U., and El Haddad, I.: Elucidating the present-day chemical composition, seasonality and source regions of climate-relevant aerosols across the Arctic land surface, Environ. Res. Lett., 17, 034032, https://doi.org/10.1088/1748-9326/ac444b, 2022.

Pernov, J. B., Beddows, D., Thomas, D. C., Dall´Osto, M., Harrison, R. M., Schmale, J., Skov, H., and Massling, A.: Increased aerosol concentrations in the High Arctic attributable to changing atmospheric transport patterns, npj Clim Atmos Sci, 5, 1–13, https://doi.org/10.1038/s41612-022-00286-y, 2022.

Rinke, A., Cassano, J. J., Cassano, E. N., Jaiser, R., and Handorf, D.: Meteorological conditions during the MOSAiC expedition: Normal or anomalous?, Elementa: Science of the Anthropocene, 9, 00023, https://doi.org/10.1525/elementa.2021.00023, 2021.

Shupe, M. D., Rex, M., Blomquist, B., Persson, P. O. G., Schmale, J., Uttal, T., Althausen, D., Angot, H., Archer, S., Bariteau, L., Beck, I., Bilberry, J., Bucci, S., Buck, C., Boyer, M., Brasseur, Z., Brooks, I. M., Calmer, R., Cassano, J., Castro, V., Chu, D., Costa, D., Cox, C. J., Creamean, J., Crewell, S., Dahlke, S., Damm, E., de Boer, G., Deckelmann, H., Dethloff, K., Dütsch, M., Ebell, K., Ehrlich, A., Ellis, J., Engelmann, R., Fong, A. A., Frey, M. M., Gallagher, M. R., Ganzeveld, L., Gradinger, R., Graeser, J., Greenamyer, V., Griesche, H., Griffiths, S., Hamilton, J., Heinemann, G., Helmig, D., Herber, A., Heuzé, C., Hofer, J., Houchens, T., Howard, D., Inoue, J., Jacobi, H.-W., Jaiser, R., Jokinen, T., Jourdan, O., Jozef, G., King, W., Kirchgaessner, A., Klingebiel, M., Krassovski, M., Krumpen, T., Lampert, A., Landing, W., Laurila, T., Lawrence, D., Lonardi, M., Loose, B., Lüpkes, C., Maahn, M., Macke, A., Maslowski, W., Marsay, C., Maturilli, M., Mech, M., Morris, S., Moser, M., Nicolaus, M., Ortega, P., Osborn, J., Pätzold, F., Perovich, D. K., Petäjä, T., Pilz, C., Pirazzini, R., Posman, K., Powers, H., Pratt, K. A., Preußer, A., Quéléver, L., Radenz, M., Rabe, B., Rinke, A., Sachs, T., Schulz, A., Siebert, H., Silva, T., Solomon, A., et al.: Overview of the MOSAiC expedition: Atmosphere, Elementa: Science of the Anthropocene, 10, 00060, https://doi.org/10.1525/elementa.2021.00060, 2022.

---

## Referee Report (RR1)

I am providing a follow-up review to Heutte et al. "Observations of high time-resolution and size-resolved aerosol chemical composition and microphysics in the central Arctic implications for climate-relevant particle properties". Heutte et al. have made useful revisions to their manuscript. Notably, the manuscript had been significantly shortened by merging sections and moving sections to the Supplement. Also, the readability had been improved by removing redundant information and shorten a few long sentences. While the manuscript could still be more concise, it is comprehensive, which is also important. I have just two further minor comments provided below.

- Thanks for clarification with the $NO_3^+$ signal and the interferences at m/z 30 with $C^{18}O$. However, can you please clarify if this is an instrument-specific issue or if it was something specific for this measurement period.
- Besides that, I just recognized that the nitrate signal is partly correlated with the NaCl signal from the AMS (Fig. 5). Is it feasible that some measured nitrate was incorporated in sea salt particles? This might by indicative for aged sea spray particles as for example discussed in Gard et al. (1998).

Gard *et al.*, Direct Observation of Heterogeneous Chemistry in the Atmosphere. *Science* **279**, 1184-1187 (1998). DOI:10.1126/science.279.5354.1184

---

## Author Response (AR2)

Reviewer's comments in black, Authors' response in blue
Line numbering in the answers is related to the version of the revised manuscript where the "track changes" mode is not shown.

**Authors' response to referee #3**

I am providing a follow-up review to Heutte et al. "Observations of high time-resolution and size-resolved aerosol chemical composition and microphysics in the central Arctic implications for climate-relevant particle properties". Heutte et al. have made useful revisions to their manuscript. Notably, the manuscript had been significantly shortened by merging sections and moving sections to the Supplement. Also, the readability had been improved by removing redundant information and shorten a few long sentences. While the manuscript could still be more concise, it is comprehensive, which is also important. I have just two further minor comments provided below.

• Thanks for clarification with the NO3+ signal and the interferences at m/z 30 with C18O. However, can you please clarify if this is an instrument-specific issue or if it was something specific for this measurement period.

The interference is actually related to the processing of the AMS data. For the MOCCHA data (August-September 2018), processed by Karlsson et al. (2022) , the $CO^+$ fragment at $m/z$ 28 was fitted and by extension the $C^{18}O^+$ fragment at $m/z$ 30 was defined based on the intensity of the signal for the $CO^+$ fragment. This led to the interference between the $NO^+$ and $C^{18}O^+$ fragments at $m/z$ 28, for the MOCCHA period.

However, for the MOSAiC AMS dataset, the $CO^+$ fragment was not fitted but calculated based on the intensity of the $CO_2^+$ fragment at $m/z$ 44. Hence, interferences between the $NO^+$ and $C^{18}O^+$ fragments were not possible. We therefore deleted the statement at lines 407-408 of the manuscript which stated that "As for the MOCCHA measurements, $NO_3^-$ likely suffered from interferences with the $C_{18}O^+$ fragment at $m/z$ 30, which could explain its relatively high fractional contribution to the $PM_1$ mass.".
The increased nitrate signal in October-November during MOSAiC could have, as pointed out below by the reviewer, originated from sodium nitrate following chloride displacement in aged sea-salt particles. Nitrate has also been previously found in long-range transported haze particles, associated with combustion processes (Quinn et al., 2007). We added a statement regarding the potential sources of nitrate in October-November at lines 419-421: "Similarly, for $NO_3^-$, part of the signal could be associated with sodium nitrate following chloride displacement in aged sea-salt particles (Gard et al., 1998), while long-range transport of $NO_3^-$ in haze particles is also likely playing a role (Quinn et al., 2007).".

• Besides that, I just recognized that the nitrate signal is partly correlated with the NaCl signal from the AMS (Fig. 5). Is it feasible that some measured nitrate was incorporated in sea salt particles? This might by indicative for aged sea spray particles as for example discussed in Gard et al. (1998).

The reviewer made a valid point that the nitrate signal seemed to have a partial correlation with the NaCl signal in Fig. 5. We quantified this potential correlation taking all data between November 8th and December 3rd during MOSAiC (see Fig. A1 below), the period when nitrate and sea salt concentrations were the highest. The linear Pearson correlation between $NO_3^-$ and NaCl was low overall ($\rho_{Pearson} = 0.28$) as shown in Fig. A2, which suggest that if a relation exists between the two variables, it is not necessarily linear. Periods when $NO_3^-$ and NaCl covaried in time (such as on December 3rd, see Fig. A1) could indicate that $NO_3^-$ could have been incorporated in aged sea salt particles as sodium nitrate. However,

periods when the two variables did not covary in time (such as between November 12th and 15th) suggest that $NO_3^-$ and NaCl were externally mixed. Furthermore, a low correlation was found between $Na^+$ and $NO_3^-$ ($\rho_{Pearson} = 0.24$), and a high correlation was found between NaCl and $Na^+$ ($\rho_{Pearson} = 0.70$), suggesting that the signal of the sodium fragment in the AMS was related with NaCl and not with sodium nitrate ($NaNO_3$).

Since the relation between $NO_3^-$ and NaCl is not entirely clear, we decided not to discuss it in the storm case study in Sect. 3.2 and only keep the statement added at lines 419-421 (see the answer to the previous reviewer comment).

[Figure]

**Figure A1: Timeseries of nitrate mass concentrations ($NO_3^-$), sea salt signal (NaCl * 51), and of the sodium fragment ($Na^+$) in November and early December during MOSAiC.**

[Figure]

**Figure A2: Scatter plot and Pearson correlation between $NO_3^-$ and NaCl (left), $NO_3^-$ and $Na^+$ (middle), and $Na^+$ and NaCl (right) for the month of November and early December during MOSAiC.**

**References (from anonymous referee #3):**

Gard et al., Direct Observation of Heterogeneous Chemistry in the Atmosphere.Science 279,1184-1187(1998). DOI:10.1126/science.279.5354.1184

**References (from the authors):**

Karlsson, L., Baccarini, A., Duplessis, P., Baumgardner, D., Brooks, I. M., Chang, R. Y.-W., Dada, L., Dällenbach, K. R., Heikkinen, L., Krejci, R., Leaitch, W. R., Leck, C., Partridge, D. G., Salter, M. E., Wernli, H., Wheeler, M. J., Schmale, J., and Zieger, P.: Physical and Chemical Properties of Cloud Droplet Residuals and Aerosol Particles During the Arctic Ocean 2018 Expedition, Journal of Geophysical Research: Atmospheres, 127, e2021JD036383, https://doi.org/10.1029/2021JD036383, 2022.

Quinn, P. K., Shaw, G., Andrews, E., Dutton, E. G., Ruoho-Airola, T., and Gong, S. L.: Arctic haze: current trends and knowledge gaps, Tellus B, 59, 99–114, https://doi.org/10.1111/j.1600-0889.2006.00238.x, 2007.